# The Half-order Energy Balance Equation, Part 1: The homogeneous HEBE and long memories

Shaun Lovejoy

Physics dept., McGill University, Montreal, Que. H3A 2T8, Canada

Correspondence to: Shaun Lovejoy (lovejoy@physics.mcgill.ca)

**Abstract:** The original Budyko-Sellers type 1-D energy balance models (EBMs) consider the Earth system averaged over long times and applies the continuum mechanics heat equation. When these and the more phenomenological box models are extended to include time varying anomalies, they have a key weakness: neither model explicitly nor realistically treats the conductive - radiative surface boundary condition that is necessary for a correct treatment of energy storage.

In this first of a two part series, we apply standard Laplace and Fourier techniques to the continuum mechanics heat equation, solving it with the correct radiative - conductive boundary conditions obtaining an equation directly for the surface temperature anomalies in terms of the anomalous forcing. Although classical, this equation is half – not integer – ordered: the "Half - ordered Energy Balance Equation" (HEBE). A quite general consequence is that although Newton's law of cooling holds, that the heat flux across surfaces is proportional to a half (not first) ordered time derivative of the surface temperature. This implies that the surface heat flux has a long memory, that it depends on the entire previous history of the forcing, the temperature- heat flux relationship is no longer instantaneous.

We then consider the case where the Earth is periodically forced. The classical case is diurnal heat forcing; we extend this to annual conductive – radiative forcing and show that the surface thermal impedance is a complex valued quantity equal to the (complex) climate sensitivity. Using a simple semi-empirical model of the forcing, we show how the HEBE can account for the phase lag between the summer maximum forcing and maximum surface temperature Earth response.

In part II, we extend all these results to spatially inhomogeneous forcing and to the full horizontally inhomogeneous problem with spatially varying specific heats, diffusivities, advection velocities, climate sensitivities. We consider the consequences for macroweather (monthly, seasonal, interannual) forecasting and climate projections.

## 1 Introduction

Ever since [*Budyko*, 1969] and [*Sellers*, 1969] proposed a simple model describing the exchange of energy between the earth and outer space, energy balance models (EBMs) have provided a straightforward way of understanding past, present and possible future climates. The models usually have either zero or one spatial dimension representing respectively the globally or latitudinally averaged meridional temperature distribution (for a review, see [*McGuffie and Henderson-Sellers*, 2005 ], and [*North and Kim*, 2017]).

The fundamental EBM challenge is to model the way that imbalances in incoming short wave and outgoing long wave radiation are transformed into changes in surface temperatures. In an energy balanced climate state, the vertical flux imbalances are transported horizontally. Here we are primarily interested in the anomalies with respect to this state. When an external flux (forcing) is added, some of this anomalous imbalance is radiated to outer space while some is converted into sensible heat and conducted into (or out of) the subsurface. This latter flux accounts for both energy storage as well as for surface temperature changes and attendant changes in long wave emissions. EBMs avoid explicit treatment of this critical surface boundary condition, treating it phenomenologically in ways that are flawed; in this two part paper, we show how they

can easily be improved with significant benefits: first, the (idealized) homogeneous case (part I), and then the general horizontally inhomogeneous (2D) case (part II).

First consider box EBMs with zero horizontal dimensions, a model of the mean Earth temperature. These are based on two distinct assumptions: a) that the rate that heat ($S$) is exchanged between the earth and outer space ($dS/dt$) is proportional to the difference between the surface temperature ($T$) and its long term equilibrium value ($T_{eq}$): $dS/dt \propto (T_{eq}-T)$ (Newton's

Law of Cooling, NLC) and b) that this rate is also proportional to the rate of change of surface temperature: $dS/dt \propto dT/dt$. Budyko-Sellers models are on firmer ground: they start with the basic continuum mechanics heat equation with advective and diffusive heat transport. Yet they have no vertical coordinate, and so are unable to correctly treat the surface conduction – radiation - energy storage issue. By restricting explicit treatment of energy transport to the horizontal, they resort to the ad hoc assumption that the vertical flux imbalances are redirected horizontally and meridionally. The original

Budyko-Sellers models were of time independent climate states, there was no energy storage at all: the radiative imbalances were completely redirected. While this approximation may be reasonable for these long term states, they become problematic as soon the original models were extended to include temporal variations ([*Dwyers and Petersen*, 1975]). While these time varying extensions implicitly allow for subsurface energy storage, this implicit treatment is both unnecessary and unsatisfactory.

The basic physical problem is that anomalous radiative flux imbalances partly lead to heat conduction fluxes into the subsurface and partly to changes in longwave radiative fluxes. The part conducted into the subsurface is stored and may re-emerge, possibly much later. Starting with the heat equation, realistic and mathematically correct treatments, involve the introduction of a vertical coordinate and the use of conductive - radiative surface boundary conditions (BCs). If one considers the horizontally homogeneous 3-D problem in a semi-infinite medium with these mixed BCs and linearized long wave

emissions, the problem is classical and can be straightforwardly solved using Laplace and Fourier techniques. Mathematically it turns out that the key is the surface layer that defines the surface vertical temperature gradient. The influence of the subsurface is only over a thin layer of the order of a few diffusion depths where most of the energy storage occurs. This depth depends on the specific heat per volume as well as the diffusivity and is estimated to be typically of the order of 100m for the ocean (depending its turbulent diffusivity), and less over land (see appendix A, part 2).

The exact treatment of this homogeneous problem confirms that Newton's law of cooling holds, but shows that the classical box model relation between heat flux and the surface temperature is wrong: symbolically the correct relation is $dS/dt \propto d^{h}T/dt^{h}$ with $h = 1/2$ - not the phenomenological value $h = 1$. Physically, these fractional derivatives are simply convolutions and in the Fourier domain, they are power law filters, in this case involving power law storage (hence "memories"). The corresponding half-order energy balance equation (HEBE) has qualitatively much stronger storage than the

short exponential memories associated with the standard integer ordered ($h = 1$) box model derivatives.

Half-order derivatives have appeared in heat and diffusion problems since at least [*Meyer*, 1960], [*Oldham and Spanier*, 1972], [*Oldham*, 1973], and [*Oldham and Spanier*, 1974]. An equation mathematically identical to the

homogeneous $h = 1/2$ special case of the FEBE was derived by [*Oldham*, 1973] as a short time approximation to electrolyte diffusion in a spherical geometry, and [*Oldham and Spanier*, 1974] anticipate our present application by noting that half-order derivatives can be applied to "not one but an entire class of boundary value problems...". Later, half-order derivatives were developed by [*Babenko*, 1986], and have been regularly exploited in engineering heat transfer problems, see e.g. [*Sierociuk et al.*, 2013], [*Sierociuk et al.*, 2015] and references therein. The method is probably not more generally known since most applications are with fairly standard heat flux boundary conditions and other more familiar techniques can also be used.

More generally, fractional derivatives and their equations [*Podlubny*, 1999], have a history going back to Leibniz in the 17th century and their development has exploded in the last decades (for books on the subject, see e.g. [*Miller and Ross*, 1993], [*Podlubny*, 1999], [*Hilfer*, 2000], [*West et al.*, 2003], [*Tarasov*, 2010], [*Klafter et al.*, 2012], [*Klafter et al.*, 2012], [*Baleanu et al.*, 2012], [*Atanackovic et al.*, 2014]).

Interestingly, the explicit or implicit application of fractional derivatives to model the Earth's temperature - and more recently the energy budget - has several antecedents arising from the wide range spatial scaling symmetries of atmospheric fields respected by the fluid equations, models and (empirically) by the atmospheric fields themselves (see the reviews [*Lovejoy and Schertzer*, 2013], [*Lovejoy*, 2019a]). Since this includes the velocity field - whose spatial scaling implies scaling in time - it implies that power laws should be more realistic than exponentials. At first, this led to power law Climate Response Functions (CRFs), [Rypdal, 2012; van Hateren, 2013], [*Rypdal and Rypdal*, 2014], [Rypdal et al., 2015] , [Hebert, 2017], [*Hébert et al.*, 2020]. However, without truncations, pure power law CRFs lead to divergences: the "runaway Green's function effect" [*Hébert and Lovejoy*, 2015], a model unstable to infinitesimal step function increases in forcing: the Equilibrium Climate Sensitivity is infinite. These can be tamed either by a high frequency truncation ([Hebert, 2017], [*Hébert et al.*, 2020]), or by constraining forcings to return to zero *[Rypdal, 2016],* [*Myrvoll-Nilsen et al.*, 2020].

However, [*Lovejoy*, 2019a], [*Lovejoy et al.*, 2021], argued that it is not the CRF itself, but rather the earth's heat storage mechanisms that respect the scaling symmetry. This hypothesis implies that the corresponding storage (the derivative term) in the energy balance equation (EBE) is of fractional rather than integer order: the fractional energy balance equation (FEBE). Denoting the order of the derivative term in the equation by $h$, it was shown empirically that if the derivative was of order $h \approx$ 0.4 - 0.5 (rather than the classical EBE value $h = 1$), that it could account for both the low frequency multidecadal memory [Hebert, 2017], [*Hébert et al.*, 2020] needed for climate projections, as well as the high frequency macroweather (i.e. the regime at longer time scales than the lifetime of planetary structures, here, monthly to decadal) memory needed for monthly, seasonal and annual macroweather forecasts, [Lovejoy et al., 2015], [*Del Rio Amador and Lovejoy*, 2019; *Del Rio Amador and Lovejoy*, 2021a; *Del Rio Amador and Lovejoy*, 2021b]. Indeed, the FEBE CRF can be used directly to make climate projections that are compatible with the Coupled Model Intercomparison Project 5 (CMIP5) multi-model ensemble mean projections but with substantially smaller uncertainties ([*Procyk et al.*, 2020]). Finally, it is possible to generalize the classical (3D) continuum equation to the Fractional Heat Equation from which the (inhomogeneous, 2D) FEBE governs the surface temperature (work in progress).

In spite of empirical and theoretical support, the FEBE is essentially a phenomenological global model; in this paper we show how – at least for the $h = 1/2$ special case- it can be placed on a firmer theoretical basis while simultaneously extending it to two spatial dimensions. Our model is for macroweather temperature anomalies i.e. at time scales longer than the lifetimes of planetary structures, typically 10 days. Following Budyko and Sellers, the system averaged over weather scales is considered to be a continuum justifying the application of the continuum mechanics heat equation. Our starting point is thus the same as the classical EBMs: radiative, advective and conductive heat transport using the standard continuum mechanics energy equation. Also following the classical approaches, the longwave black body radiation is treated in its linearized form.

This work is divided into two parts. The first part is classical, it focuses on the homogeneous heat equation pointing out the consequence that with semi-infinite geometry (depth) and with (realistic) conductive - radiative boundary conditions, that the surface temperature satisfies the homogeneous HEBE. We relate this to the usual box models, Budyko-Sellers models, and classical diurnal heating models including the notions of thermal admittance and impedance and complex climate sensitivities useful in understanding the annual cycle. We underscore the generality of the basic (long memory) storage mechanism. The second part extends this work to the horizontal, first to the homogeneous case (but with inhomogeneous forcing, including a direct comparison with the classical latitudinally varying 1-D Budyko-Sellers model on the sphere), and then - using Babenko's method - to the general inhomogeneous case. Part II also contains several appendices that discuss empirical parameter estimates, spatial statistics useful for Empirical Orthogonal Functions and understanding the horizontal scaling properties as well as the changes needed to account for spherical geometry.

## 2. The Transport Equations

### 2.1 Conductive and advective heat fluxes

In most of what follows, the earth's spherical geometry plays no role, we use Cartesian coordinates with the $z$ axis pointing upwards and horizontal coordinates $\underline{x} = (x,y)$ (however in section II.2.3 and appendix II.C of part II, we treat the latitudinally varying case on a sphere). The horizontal is essentially the same as in the Budyko-Sellers model: horizontal diffusive and advective heat fluxes are atmospheric column averages lying on the surface ($z = 0$). What is new is the treatment of the vertical with radiative and conductive fluxes crossing the surface either into the subsurface (downward, the negative $z$ direction where it can propagate to $-\infty$), or to outer space (upward, $z > 0$) so that heat is effectively stored in the half-volume ($x,y,z < 0$). Although in principle this means that all the semi-infinite region $z \leq 0$ is modelled, we will see that ultimately only the vertical surface temperature derivative is needed and this is well defined as long as the surface layer is of the order of a few diffusion depths (tens or hundreds of meters). Later, we show that the main equations only explicitly depend on the local relaxation times and climate sensitivities, the vertical and horizontal transport details are only implicit. Finally, the fields are assumed to be in the macroweather regime i.e. they have been averaged over the weather – macroweather transition scale (about 10 days) or longer, and possibly for tens or hundreds of kilometers in space (the space-time limits are not yet clear).

Since ten days is the typical lifetime of planetary atmospheric structures, much of the actual turbulent atmospheric transport processes are averaged out, giving some justification to the parametrization. Future work is needed to clarify several foundational issues.

We start with energy transport by diffusion: Fick's law $\underline{Q}_d = -\rho c \kappa \nabla T$ where $\underline{Q}_d$ is the diffusive heat flux vector, $\kappa$ is the thermal diffusivity, $\rho$ the density, $c$ the specific heat, and $T(\underline{x},z,t)$ the temperature. Following standard energy balance models, we use eddy diffusivities that are different in the horizontal ("$h$") and vertical ("$v$"), $\kappa_h(\underline{x})$, $\kappa_v(\underline{x})$:

$$\underline{Q}_d = -\rho c \kappa_h \nabla_h T - \rho c \kappa_v \frac{\partial T}{\partial z} \hat{z}; \quad \nabla_h = \frac{\partial}{\partial x} \hat{x} + \frac{\partial}{\partial y} \hat{y} \tag{1}$$

(the circonflex indicates unit vectors). To include advection, we consider the heat equation for a fluid in a horizontal velocity field $\underline{v}_h$:

$$c\rho \frac{DT}{Dt} = -\nabla \cdot \underline{Q}_d; \quad \frac{DT}{Dt} = \frac{\partial T}{\partial t} + \underline{v}_h \cdot \nabla T \tag{2}$$

Where $D/Dt$ is the advective derivative. The heat equation is therefore:

$$c\rho \frac{\partial T}{\partial t} = -c\rho \underline{v}_h \cdot \nabla_h T + \nabla_h \cdot \left( \rho c \kappa_h \nabla_h T \right) + \frac{\partial}{\partial z} \left( \rho c \kappa_v \frac{\partial T}{\partial z} \right) \tag{3}$$

If the volumetric specific heat ($c\rho$) is constant and using the continuity equation, $\nabla \cdot \left( c\rho \underline{v}_h \right) = 0$ and we can write:

$$c\rho \frac{\partial T}{\partial t} = -\nabla \cdot \left( \underline{Q}_a + \underline{Q}_d \right); \qquad \underline{Q}_a = c\rho \underline{v}_h \left( T - T_0 \right); \qquad \underline{Q}_d = -\rho c \kappa_h \nabla_h T - \rho c \kappa_v \frac{\partial T}{\partial z} \hat{z} \tag{4}$$

$Q_a$ is the advective heat flux and $T_0$ is a constant reference temperature (it disappears when the divergence is taken). This is the classical fluid heat equation, it can readily be verified that it conserves energy (integrate both sides over a volume and then use the divergence theorem). $\kappa_h(\underline{x})$, $\kappa_v(\underline{x})$, $v_h(\underline{x})$ are taken to be independent of $t$ and $z$, they are part of the climate state and are empirically determined so as to reproduce the time independent climate temperature distribution. In future work, they could be given their own time-varying anomalies.

**2.2 Radiative heat fluxes**

At the surface, there is an incoming energy flux $R_\downarrow$ :

$$R_\downarrow(\underline{x},t) = Q_0(\underline{x}) + F(\underline{x},t) \tag{5}$$

Where $F$ is the anomalous forcing and $Q_0(\underline{x})$ is the local solar radiation:

$$Q_0(\underline{x}) = QS(\underline{x})a(\underline{x}) \tag{6}$$

$Q$ is the mean top of the atmosphere flux ($\approx 341$ W/m$^2$), $S(\underline{x})$ is the dimensionless local solar constant with local coalbedo $a(\underline{x})$ (in the notation of [*North and Kim*, 2017]) and the time dependent part of the radiative balance is specified by the additional incoming energy flux, the "forcing" $F(\underline{x},t)$. Although in this paper we mostly ignore temporal albedo variations (see however section 3.3), they are important for studying temperature-albedo feedbacks and climate transitions. If needed, even if they include a (potentially nonlinear) temperature dependence, they are easy to incorporate. For example, they could be included in $F$ by using $a(\underline{x},t) = a_0(\underline{x}) + a_1(\underline{x},t,T(\underline{x},t))$ in place of $a(\underline{x})$ in eq. 6 and

$F(\underline{x},t) = F_0(\underline{x},t) + QS(\underline{x})a_1(\underline{x},t,T(\underline{x},t))$ in place of $F$ in eq. 5.

As usual, $F(\underline{x},t)$ includes solar, volcanic and anthropogenic forcings. However since macroweather includes random internal variability, $F(\underline{x},t)$ also includes a stochastic internal variability component. Finally, for macroweather scales shorter than a year, $F$ could also include the annual cycle and therefore possible cyclical albedo variations due to seasonally varying cloudiness (section 3.3). Alternatively $T$ and $F$ can be deseasonalized in the usual way to yield standard monthly climate "normals" so that the mean anomalies are zero over the climate normal reference period.

$R_\downarrow(\underline{x},t)$ is partially balanced by the outgoing $R_\uparrow(\underline{x},t)$ that depends on the surface temperature and the effective emissivity $\varepsilon(\underline{x})$:

$$R_\uparrow(\underline{x},t) = \sigma\varepsilon(\underline{x})T(\underline{x},0,t)^4 \tag{7}$$

where $\sigma$ is Stefan-Boltzmann constant. The $R_\downarrow$, $R_\uparrow$ imbalance drives the system, it implies that heat diffuses across the surface which is the top boundary condition needed to solve eq. 3 for $T(\underline{x},z<0,t)$:

$$\left(\sigma\varepsilon(\underline{x})T(\underline{x},z,t)^4 + \rho c\kappa_v(\underline{x})\frac{\partial T(\underline{x},z,t)}{\partial z}\right)\Bigg|_{z=0} = Q_0(\underline{x}) + F(\underline{x},t) \tag{8}$$

The derivative term $\rho c\kappa_v \left.\partial T/\partial z\right|_{z=0} = Q_s$ is the conductive (sensible) heat flux across the surface, into the earth, see fig. 1. The radiative fluxes thus impose a "mixed" conductive - radiative boundary condition involving both $T$ and $\partial T/\partial z$ (they

are a special case of "Robin" boundary conditions [*Hahn and Ozisk*, 2012 ]).  If we add the initial condition $T\left(\underline{x},z,t=0\right)=0$

(or later, $T\left(\underline{x},z,t=-\infty\right)=0$ ) and the Dirichlet boundary condition at great depth $T\left(\underline{x},z=-\infty,t\right)=0$ and assume that the

system is periodic or infinite in the horizontal, then, in principle, these are enough to determine the temperature for $T(\underline{x},z<0,t>0)$

(or eventually, $T\left(\underline{x},z,t=-\infty\right)=0$ ).  Instead of avoiding this conductive - radiative BC below we show how it directly yields

an equation for the surface temperature.

### 2.3 The Climatological and anomaly fields

Let us now decompose the heat flux and temperature into time independent (climatological) and time varying (anomaly) components: $Q_c$, $T_c$ and $Q$, $T$.  As usual, we linearize the outgoing black body radiation, although we do so around the spatially varying surface temperature $T_c(\underline{x},z=0)$ (i.e. not the global average temperature) which yields spatially varying coefficients:

$$R_\uparrow\left(T_c\left(\underline{x},0\right)+T\left(\underline{x},0,t\right)\right)\approx R_\uparrow\left(T_c\left(\underline{x},0\right)\right)+\frac{T\left(\underline{x},0,t\right)}{s\left(\underline{x}\right)} \qquad (9)$$

($T_c+T$ is the actual temperature), with climate sensitivity:

$$s\left(\underline{x}\right)=\frac{1}{4\sigma\varepsilon\left(\underline{x}\right)T_c\left(\underline{x},0\right)^3} \qquad (10)$$

Since typical macroweather temperature anomalies are only a few degrees, the black body emission is quite linear with the temperature anomaly.  However due to feedbacks, the proportionality coefficient – the climate sensitivity – as estimated in eq. 10 is not accurate; below, we simply consider $s(\underline{x})$ to be an empirically determined function of position.

The incoming radiation at the location $\underline{x}$ drives the system.  The radiative imbalance $\Delta R$ going into the subsurface is therefore equal to the conductive flux $Q_s$ into the surface; it specifies the conductive-radiative surface boundary condition for $T_c$ and the anomalies $T$:

$$\Delta R=Q_s; \qquad \Delta R=R_\downarrow - R_\uparrow; \qquad Q_s=-Q_{d,z} \qquad (11)$$

Where $Q_{d,z}$ is the (upward) vertical component of the heat flux at the surface given by Fick's law: $Q_{d,z}=-\rho c\kappa_v\left.\frac{\partial T}{\partial z}\right|_{z=0}$ .

The conductive - radiative surface boundary conditions for the time independent climate and anomaly temperatures is therefore:

$$\left( R_\uparrow\left( T_c\left(\underline{x},z\right)\right)+\rho c\kappa_v\frac{\partial T_c\left(\underline{x},z\right)}{\partial z}\right)\Bigg|_{z=0}=Q_0\left(\underline{x}\right)$$

$$\left( \frac{T\left(\underline{x},z,t\right)}{s}+\rho c\kappa_v\frac{\partial T\left(\underline{x},z,t\right)}{\partial z}\right)\Bigg|_{z=0}=F\left(\underline{x},t\right) \tag{12}$$

$s$, $\rho$, c and $\kappa$ are all presumed to be functions of $\underline{x}$. Note: the conductive heat flux is a sensible heat flux; the boundary condition involves its vertical component that represents heat stored in the subsurface. While eqs. 11, 12 involve the vertical temperature derivative at the surface (i.e. over an infinitesimal layer), $l_v = s\rho c\kappa_v$ defines the diffusion depth (typically $\approx$ 10 - 100m in

thickness, see part II); so that physically the model need only be realistic over this fairly shallow depth where most of the (anomalous) heat is stored.

Now, in the temperature eq. 3, replace $T$ by $T_c+T$. The equation for the time independent climate part is:

$$c\rho\frac{\partial T_c}{\partial t}=0=-\rho c\underline{v}_h\cdot\nabla_h T_c+\nabla_h\cdot\left(\rho c\kappa_h\nabla_h T_c\right)+\frac{\partial}{\partial z}\left(\rho c\kappa_v\frac{\partial T_c}{\partial z}\right) \tag{13}$$

and for the time-varying anomalies:

$$c\rho\frac{\partial T}{\partial t}=-\rho c\underline{v}_h\cdot\nabla_h T+\nabla_h\cdot\left(\rho c\kappa_h\nabla_h T\right)+\frac{\partial}{\partial z}\left(\rho c\kappa_v\frac{\partial T}{\partial z}\right) \tag{14}$$

These equations must now be solved using boundary conditions eqs. 11, 12 for respectively $T_c$, $T$ and $T_c = T = 0$ at $z = -\infty$ (all $t$), and $T\left(\underline{x},z,t=0\right)=0$ (or see below, $T\left(\underline{x},z,t=-\infty\right)=0$ ).

The separation into one equation for the time invariant climate state and another for the time-varying anomalies is done for convenience. As long as the outgoing long wave radiation is approximately linear over the whole range of temperatures

(as is commonly assumed in EBMs), this division involves no anomaly smallness assumptions nor assumptions concerning their time averages; the choice of the reference climate depends on the application. Below, we choose anomalies defined in the standard way (although not necessarily with the annual cycle removed, section 3.3), this is adequate for monthly and seasonal forecasts as well as 21$^{st}$ century climate projections. However, a different choice might be more appropriate for modelling transitions between different climates including possible chaotic behaviours.

### 2.4 The climatological temperature distribution and Budyko-Sellers models

In order to simplify the problem, starting with [*Budyko*, 1969] and [*Sellers*, 1969], the usual approach to obtaining $T_c$ is somewhat different. First, the climatological temperature field is only defined at $z = 0$, i.e. $T_c(\underline{x}) = T_c(\underline{x},0)$. Without a vertical coordinate, the climatological radiative imbalance $Q_0(\underline{x}) - R_\uparrow(T_c(\underline{x}))$ no longer forces the system via the vertical surface derivative (eq. 11), instead the imbalance is conventionally redirected in the meridional direction away from the equator (fig. 2).

To see how this works, return to eq. 4 for the climatological component and put $\dfrac{\partial}{\partial z} = 0$:

$$\underline{Q}_c(\underline{x}) = \underline{Q}_{c,a}(\underline{x}) + \underline{Q}_{c,d}(\underline{x}) + sign(y)\big(Q_0(\underline{x}) - R_\uparrow(T_c(\underline{x}))\big)\hat{y} \tag{15}$$

(in this formulation, one usually uses the latitude angle instead of the meridional coordinate $y$ see part II, section 2.3). The direction of the redirected vertical flux is always away from the equator ($y = 0$; hence $sign(y)$), in any event, zonal fluxes will cancel when averaged over latitudinal bands.

The usual Budyko-Sellers type models then average $\underline{Q}_c$ over lines of constant latitude yielding a 1-D model:

$$\overline{\underline{Q}}_c(y) = \left( \rho c\left( v_y \overline{T}_c - \kappa_h \frac{\partial \overline{T}_c}{\partial y}\right) + sign(y)\big(Q_0(y) - R_\uparrow(\overline{T}_c)\big)\right)\hat{y} \tag{16}$$

(overbar indicates averaging over all longitudes, $x$).

In the more popular Seller's version, the basic horizontal transport is due to the eddy thermal diffusivity, the $\kappa_h$ term. There may also be a small advection velocity $v$ but it is not considered to be a true physical velocity but only an ad hoc parameter needed to prevent $\kappa_h$ from being negative ([*Sellers*, 1969]), the standard presentation ([*North et al.*, 1981], [*North and Kim*, 2017]) avoids the problem by using the diffusivity, see section 3.1. The horizontal eddy diffusivity $\kappa_h$ is often taken as the sum of contributions from water, water vapor and air. In the pure Budyko version, there is no eddy diffusivity, the heat flux is assumed to be proportional to the temperature difference with respect to a reference (e.g. mean) value; $(Q)_y \propto (T - T_0)$. Comparing this with eq. 4 for $Q_a$, we see that this implies that Budyko horizontal heat fluxes are purely advective.

The final step to obtaining the energy equation is to take the divergence:

$$\nabla \cdot \overline{\underline{Q}}_c = \frac{\partial \overline{\underline{Q}}_c}{\partial y} = -\rho c \frac{\partial \overline{T}_c}{\partial t} \tag{17}$$

Budyko and Sellers only considered the time independent case and obtained:

$$\frac{\partial \overline{Q_c}(y)}{\partial y} = 0$$

$$\overline{Q_c}(y) = const$$

(18)

By appropriately choosing a reference temperature (usually the global average), the constant can be adjusted for convenience. Somewhat later, [*Dwyers and Petersen*, 1975] considered the time independent case (eq. 17) which is second order in *y*. Subsequently the model has been widely used for studying different past and future climates and the corresponding transitions.

Note that the $\rho c \frac{\partial \overline{T_c}}{\partial t}$ term corresponds to energy storage; in the time independent case there is no storage.

**3. The classical origin of the fractional operators: conductive-radiative boundary conditions in a semi-infinite domain**

**3.1 The zero dimensional homogeneous heat equation**

**3.1.1 The key parameters**

No matter how the climate temperature equation is solved, the equation for the time dependent anomaly temperature remains eq. 14. We now rewrite it in a way that brings out the critical mathematical properties. Since $\rho c$ and $\kappa_v$ are taken to be only functions of $\underline{x}$, eq. 14 can be rewritten:

$$\left(\frac{\partial}{\partial t} - \kappa_v \frac{\partial^2}{\partial z^2}\right)T = -\underline{v}\cdot\nabla_h T + \kappa_h \nabla_h^2 T; \qquad \begin{array}{c} \underline{v} = \underline{v}_h - \underline{v}_d \\ \\ \underline{v}_d = \frac{1}{\rho c}\nabla_h\left(\kappa_h \rho c\right) \end{array}$$

(19)

Where we have defined an effective diffusion velocity $\underline{v}_d$ and effective advection velocity $\underline{v}$. Eq. 19 must be solved with the
265 boundary conditions in eq. 12.

The roles of the various terms are clearer if the equation is nondimensionalized. For this, we note that if we include the boundary conditions, the anomaly temperature is entirely determined by the dimensional quantities $\kappa$, *s*, $\rho$ and *c*. From these, there exists a unique dimensional combination $\tau(\underline{x})$ with dimensions of time, we will see that this controls the relaxation of the system back to energy balance, it is a "relaxation time" (for the zero-dimensional model, energy balance is the same as
thermodynamic equilibrium). Using $\kappa_v$ yields:

$$\tau = \kappa_v \left( \rho c s \right)^2; \qquad l_v = \left( \tau \kappa_v \right)^{1/2} = \kappa_v \rho c s \qquad (20)$$

where $l_v(\underline{x})$ is the vertical relaxation length of the surface energy balance processes.   In the next section, we give some rough parameter estimates.   We may also define the horizontal diffusion length $l_h$, speed $V$, nondimensional (square root) diffusivity ratio β and nondimensional advection vector $\underline{\alpha}$:

$$\underline{\alpha} = \frac{\underline{v}}{V}; \quad V = \frac{l_h}{\tau}; \quad l_h = \left( \tau \kappa_h \right)^{1/2} = \beta \kappa_h \rho c s; \quad \beta = \left( \frac{\kappa_v}{\kappa_h} \right)^{1/2} \qquad (21)$$

The continuity equation for energy becomes $\nabla \cdot \left( \dfrac{\beta}{s} \underline{\alpha} \right) = 0$.   For global (zero dimensional) models, τ has been estimated as 2.4 – 7.0 years (90% confidence, [*Procyk et al.*, 2020]) which is comparable to the classical exponential relaxation time scales mentioned above ([*Hebert*, 2017],) and in section 3.3 we estimate $\tau \approx 2.75$ years.

In order to understand the classical origin of fractional derivatives, it is helpful to consider the homogeneous Seller-type (diffusive transport) heat equation where τ, $l_v$ and $l_h$ are constants and can thus be used to nondimensionalize the operators. The nondimensional $t$ is therefore in terms of relaxation times, the nondimensional $\underline{x}$ in terms of diffusion lengths $l_h$ and the nondimensional $z$ in terms of diffusion depths $l_v$.   By taking $s = 1$, we effectively use a forcing $F$ with dimensions of temperature.   The result is an equation with nondimensional operators acting on temperatures.   In part I, we consider only the "zero dimensional" equation where the "zero" refers to the number of horizontal dimensions (i.e. only vertical, $z$ and time $t$).

Using the dimensional parameters in eqs. 20, 21, we can write the equations as:

$$\underline{Q}_h = -\frac{l_h}{s} \nabla_h T + \frac{\underline{\alpha}}{s} \left( T - T_0 \right); \quad Q_z = -\frac{l_v}{s} \frac{\partial T}{\partial z} \qquad (22)$$

$$\tau \frac{\partial T}{\partial t} = -\zeta T - l_v s \frac{\partial Q_z}{\partial z}; \qquad \begin{array}{c} \zeta = l_h s \nabla_h \cdot \left( \dfrac{\underline{\alpha} - l_h \nabla_h}{s} \right) \\[2ex] \zeta T = l_h s \nabla_h \cdot \underline{Q}_h \end{array} \qquad (23)$$

Where ζ is the dimensionless horizontal transport operator.   We have ignored the reference temperature $T_0$ by either taking it to be zero or by assuming $\nabla_h \cdot \left( s^{-1} \underline{\alpha} \right) = 0$ which is true if β = constant.

If the advection is chosen appropriately (as in eq. 24), then we may write the horizontal transport operator in the form:

$$\zeta = -s\nabla_h \cdot \left(\frac{l_h}{s}\right)\nabla_h; \quad \underline{\alpha} = s\nabla_h\left(\frac{l_h}{s}\right) \tag{24}$$

This is convenient for comparing the HEBE with the 1-D B-S equations on a sphere in part II section 2.3, and it avoids the unphysical negative diffusivities reported by Sellers. Following [*North and Kim*, 2017], in spherical geometry, we can introduce $D_F$ which is the diffusion constant per radian:

$$\zeta = -sR\nabla_h \cdot D_F\nabla_h; \quad D_F = \frac{l_h(\underline{x})}{Rs(\underline{x})} = \kappa_h \frac{\beta\rho c}{R} \tag{25}$$

Where $R$ is the earth radius.

### 3.1.2 Parameter estimates

Before proceeding, it is useful to get a feel for typical values of the parameters in the equations. In part II, section 2.3 and appendix II.A we combine these parameter estimates with analyses of monthly space-time temperature anomalies in order to analyse which terms in the equations are dominant at different time scales, the following are order of magnitude estimates. The basic parameters are the horizontal diffusivity $\kappa_h$, and the volumetric specific heat $\rho c$, the sensitivity $s$, vertical diffusivity $\kappa_v$, $\kappa_h$ and $\tau$. They can be estimated as follows:

a) Volumetric specific heat $\rho c$: Ocean and land values are similar, the values for water and soil are respectively $\rho c \approx 4\times10^6$, $\approx 1\times10^6$ J/(m$^3$K). The soil value depends on moisture and soil type, this is an order of magnitude estimate.

b) Climate sensitivity $s$: Using the $CO_2$ doubling value 3±1.5K, 90% confidence interval and 3.71 W/m$^2$ for $CO_2$ doubling, the global mean value is $s \approx 0.8\pm0.4$ K/W/m$^2$, with regional values a factor of $\approx 2$ higher or lower (IPCC AR5) yielding $\rho cs \approx 3\times10^6$ s/m.

c) Relaxation time $\tau$: Based on responses to anthropogenic forcings since 1880, [*Hebert*, 2017], [*Hébert et al.*, 2020; *Procyk et al.*, 2020], give the global estimate $\tau \approx 10^8$s ($\approx 4$ years). This is comparable to the relaxation times for global box models.

d) Horizontal Diffusivity $\kappa_h$: As detailed in Part II, section 2.3, [*North et al.*, 1981], [*North and Kim*, 2017] uses a diffusion constant per radian $D_F$ (eq. 25) combined with global scale climatological forcing and temperature data to estimate a global thermal conductivity $K = 4.1\times10^6$ Wm$^{-1}$K$^{-1}$ from which we estimate the horizontal (eddy) diffusivity as $\kappa_h = K/(\rho c) \approx 1$ m$^2$/s. [*Sellers*, 1969] gives values about 100 times larger for the ocean.

e) Vertical diffusivity $\kappa_v$: The vertical diffusivity is not used in the usual energy balance models, however in climate models, ocean values of $\kappa_v \approx 10^{-4}$ m$^2$/s are typical [*Houghton et al.*, 2001]. For soil, rough values are $\kappa_v \approx 10^{-6}$ m$^2$/s (wet) and

$\kappa_v \approx 10^{-7}$ m$^2$/s (dry) see [*Márquez et al.*, 2016]. Alternatively we can use $\kappa_v = \tau/(\rho cs)^2$ and the global estimates of $\tau \approx 10^8$s to obtain $\kappa_v \approx 10^{-5}$ m$^2$/s which is close to the model values.

f) Diffusion depth $l_v$: Using $l_v = \kappa_v \rho cs$ we find for the ocean and soils respectively $l_v \approx 300$m, $\approx 3 - 10$m. Using the global estimates $\kappa_v \approx 10^{-5} - 10^{-4}$ m$^2$/s yields $l_v \approx 30 - 100$m.

g) Diffusion length $l_h$: This is a key parameter: $l_h = \left(\tau \kappa_h\right)^{1/2} = \beta \kappa_h \rho cs$ (eq. 21). Using $l_h = \left(\kappa_h \kappa_v\right)^{1/2} \rho cs$, $l_h \approx 30$ km (ocean), 3 km (land). Using $l_h = \left(\tau \kappa_h\right)^{1/2}$ and $\kappa_h \approx 1$ m$^2$/s yields a global estimate $l_h \approx 10$ km.

h) Diffusive based velocity parameter $V$: $V \approx l_h/\tau \approx 3 \times 10^{-3} - 3 \times 10^{-4}$ m/s.

i) Nondimensional advection velocity $\alpha$: The best transport model – diffusive, advective – or both - is not clear, therefore let us estimate the magnitude of the advective velocity $v$ assuming that it dominates the transport. The appropriate value is not obvious since most models just use eddy diffusivity – not advection - for transport. One way - for example [Warren and Schneider, 1979] - is to note that typical meridional heat fluxes are of the order of 100 W/m$^2$ over meridional bands whose temperature gradients $\Delta T$ are several degrees K. If this heat is transported by advection, it implies $v \approx Q_a/(\rho c \Delta T) \approx 10^{-5} - 10^{-4}$m/s (eq. 4), hence, using $V \approx 10^{-4}$m/s (above), we find $\alpha = v/V \approx 0.1 - 1$.

### 3.1.3 The nondimensional equations

With $z$, $t$ in dimensionless form, the homogeneous zero dimensional heat equation is:

$$\left(\frac{\partial}{\partial t} - \frac{\partial^2}{\partial z^2}\right) T\left(t;z\right) = 0 \tag{26}$$

We use the following notation: the first argument is $t$ then horizontal space, then a semicolon followed by the depth $z$. The transfer is confined to the semi-infinite region $z \leq 0$ with boundary conditions: $T\left(t;-\infty\right) = 0$ (bottom). The system is forced by the conductive - radiative surface boundary condition at $z = 0$ (the top):

$$\left.\frac{\partial T}{\partial z}\right|_{z=0} + T\left(t;0\right) = F\left(t\right) \tag{27}$$

For initial conditions, in this section, the forcing is "turned on" at $t>0$ (i.e. $T(t;z) = 0$ for $t \leq 0$), allowing use of Laplace transforms (see section 3.3 for Fourier methods).

Performing a Laplace transform ("L.T.") of the heat equation we obtain:

$$\left(\frac{d^2}{dz^2} - p\right)\hat{T}(p;z) = -T(0;z) = 0 \tag{28}$$

Where the circonflex indicates the Laplace transform in time (with conjugate variable $p$). Solving:

$$\hat{T}(p;z) = A(p)e^{\sqrt{p}z} + B(p)e^{-\sqrt{p}z} \tag{29}$$

Where $A$, $B$ are determined by the BC's. Since we require the temperature at depth ($z<<0$) to remain finite, we must have $B = 0$, hence:

$$\hat{T}(p;z) = A(p)e^{\sqrt{p}z} \tag{30}$$

To determine $A(p)$, we Laplace transform the surface boundary condition:

$$\left.\frac{d\hat{T}}{dz}\right|_{z=0} + \hat{T}(p;0) = \hat{F}(p); \qquad F(t) \overset{L.T.}{\leftrightarrow} \hat{F}(p) \tag{31}$$

yielding:

$$A(p) = \frac{\hat{F}(p)}{1+\sqrt{p}} \tag{32}$$

It is more convenient to determine the response $G_\delta(t;z)$ to the impulse forcing $F(t) = \delta(t)$; the impulse Green's function. Using eq. 30, 32 we obtain:

$$\widehat{G_\delta}(p;z) = \frac{e^{\sqrt{p}z}}{1+\sqrt{p}}; \qquad F(t) = \delta(t) \overset{L.T.}{\leftrightarrow} \hat{F}(p) = 1 \tag{33}$$

The above assumes that the subsurface is infinitely deep. If instead it has a finite thickness $L$, and we take the bottom boundary condition as $T(t;-L) = 0$ (rather than $T(t;-\infty) = 0$), then $B(p) \approx O\left(e^{-2L\sqrt{p}}\right)$ and

$\widehat{G_\delta}(p;0) = \dfrac{1}{1+\sqrt{p}} - \dfrac{2e^{-2L\sqrt{p}}\sqrt{p}}{\left(1+\sqrt{p}\right)^2} + O\left(e^{-4L\sqrt{p}}\right)$ so that the influence of the bottom condition on the surface decreases

exponentially fast as its depth $L$ increases. Physically, as long as the depth is of the order of a few diffusion depths (estimated

as $\approx$ 100m in the ocean, $\approx$ 10m for land), the semi-infinite geometry assumption is unimportant. In the following, we therefore ignore any finite thickness corrections.

Taking the inverse Laplace transform of eq. 33 we obtain the integral representation:

$$G_\delta(t;z) = \frac{1}{\pi} \int_{-\infty}^{\infty} \frac{\zeta e^{-\zeta^2 t}}{1+\zeta^2} \left(-\sin z\zeta + \zeta \cos z\zeta\right) d\zeta \overset{L.T.}{\longleftrightarrow} \widehat{G}_\delta(p;z) = \frac{e^{\sqrt{p}z}}{1+\sqrt{p}} \qquad (34)$$

($z{\leq}0$; where we have used contour integration on the Bromwich integral).

### 3.1.4 The surface temperature

For the surface, the integral (eq. 34) can be expressed with the help of higher mathematical functions:

$$G_{0,1/2}(t;0) = G_\delta(t;0) = \frac{1}{\sqrt{\pi t}} - e^t \, erfc\sqrt{t} = \overset{L.T.}{\longleftrightarrow} \widehat{G}_{0,1/2}(p;0) = \widehat{G}_\delta(p;0) = \frac{1}{1+\sqrt{p}}; \quad erfc(z) = \frac{2}{\sqrt{\pi}} \int_{z}^{\infty} e^{-u^2} du$$

$$(35)$$

$G_\delta(t;0)$ is the $h = 1/2$ impulse response Green's function, also denoted $G_{0,1/2}$, the "0" for $0^{th}$ integral of the impulse, the "1/2" for the order of the derivative for its equation, see below), it is sometimes called a "generalized exponential", itself expressed in terms of Mittag-Leffler functions.

For long times after an impulse, the response $G_\delta(t;0) \approx t^{-3/2}$ ($t{\gg}1$, eq. 37 below) so that the system rapidly returns to its original temperature. It is more interesting to consider the response of the system to a step (Heaviside) forcing $F(t) = \Theta(t)$ ($= 1$, for $t{>}0$, $= 0$ for $t{\leq}0$) after which the system eventually attains a new energy balance (for the zero dimensional model, this corresponds to thermodynamic equilibrium). Since $\Theta(t) = \int_{0}^{t} \delta(u) du$, we have the step response

$G_\Theta(t;z) = \int_{0}^{t} G_\delta(u;z) du$ (also denoted $G_{1,1/2}$, eq. 36), and $G_\Theta(t;0) \approx 1 - \frac{1}{\sqrt{\pi t}}$ (eq. 37) i.e., a slow power law approach to thermodynamic equilibrium. Figs. 3, 4 show this at different times and depths. With unit step forcing, the boundary condition (eq. 27) indicates that the fraction of the heat flux that is transformed into long wave radiation is equal to the temperature with unit forcing. Therefore the $z = 0$ curve in fig. 3 shows that at first, all the forcing flux is conducted into the subsurface, but that this fraction rapidly vanishes as the surface approaches equilibrium. At equilibrium, the temperature has increased so that the short and long wave fluxes are once again in balance and there is no longer any conductive flux.

For future reference, we give the corresponding step response $G_{1,1/2} = G_\Theta$ which is the integral of $G_{0,1/2}$ that describes relaxation to energy balance (for this model, thermodynamic equilibrium) when $F$ is a step function. Similarly, the ramp (linear forcing) response $G_{2,1/2}$ is the integral of the step response, the second integral of the Dirac:

$$G_{1,1/2}(t) = G_{\Theta,1/2}(t) = \int_0^t G_{0,1/2}(s)\,ds = 1 - e^t erfc(t^{1/2}) \tag{36}$$

$$G_{2,1/2}(t) = \int_0^t G_{1,1/2}(s)\,ds = 1 - 2\sqrt{\frac{t}{\pi}} + t - e^t erfc(t^{1/2})$$

For small and large $t$:

$$G_{0,1/2}(t) = G_{\delta,1/2}(t) \approx \begin{cases} \dfrac{1}{\sqrt{\pi t}} - 1 + 2\sqrt{\dfrac{t}{\pi}} - t + \dfrac{4}{3}t\sqrt{\dfrac{t}{\pi}} - \dots & t \ll 1 \\[3mm] \dfrac{1}{2t\sqrt{\pi t}} - \dfrac{3}{4}\dfrac{1}{t^2\sqrt{\pi t}} + \dots & t \gg 1 \end{cases} \tag{37}$$

$$G_{1,1/2}(t) = G_{\Theta,1/2}(t) \approx \begin{cases} 2\sqrt{\dfrac{t}{\pi}} - t + \dfrac{4}{3}\dfrac{t^{3/2}}{\sqrt{\pi}} - \dots & t \ll 1 \\[3mm] 1 - \dfrac{1}{\sqrt{\pi t}} + \dfrac{1}{2t\sqrt{\pi t}} - \dots & t \gg 1 \end{cases}$$

$$G_{2,1/2}(t) \approx \begin{cases} \dfrac{4}{3}t\sqrt{\dfrac{t}{\pi}} - \dfrac{t^2}{2} + \dfrac{8}{15}t^2\sqrt{\dfrac{t}{\pi}} - \dfrac{t^3}{6} + \dots & t \ll 1 \\[3mm] t + 1 - 2\sqrt{\dfrac{t}{\pi}} - \dfrac{1}{\sqrt{\pi t}} + \dfrac{1}{2t\sqrt{\pi t}} - \dots & t \gg 1 \end{cases}$$

The asymptotic equation for the step response ($G_{1,1/2}$) shows that equilibrium is approached slowly: as $t^{3/2}$. It is this power law step response (empirically with $\approx t^{1.5}$) that was discovered semi-empirically by [*Hebert*, 2017], [*Lovejoy et al.*, 2017], [*Lovejoy et al.*, 2021] and was successfully used for climate projections through to 2100 [*Hébert et al.*, 2020]. Similarly, $\approx t^{0.4}$ behaviour was used for macroweather (monthly, seasonal) forecasts close to the short time $t^{1/2}$ expansion [*Lovejoy et al.*, 2015], [*Del Rio Amador and Lovejoy*, 2019].

If we take this as a model of the global temperature, we can use the ramp Green's function to estimate the ratio of the equilibrium climate response (ECS) to the transient climate response (TCR), we find: $TCR/ECS = G_{2,1/2}(\Delta t)/\Delta t$ where $\Delta t$ is the nondimensional time over which (for the TCR) the linear forcing acts. Using $\tau = 4$ years, and the standard $\Delta t = 70$ years for the TCR ramp, we find the plausible ratio TCR/ECS $\approx 0.78$.

### 3.1.5 Comparison with temperature forcing boundary conditions

It is interesting to compare this with the classical surface boundary condition when the system is forced by the surface temperature, an alternative – periodic surface heat forcing - is discussed in section 3.3. If the surface ($z = 0$) boundary condition $T_{force}(t)$ is imposed:

$$T_{temp}(t;0) = T_{force}(t) \tag{38}$$

then there will be vertical surface gradients that imply that heat is conducted through the surface. To obtain the impulse response Green's function, we take $T_{force}(t) = \delta(t)$ and repeating the Laplace transform approach, we obtain $A(p) = 1$ (eq. 31 with no derivative term). This yields the following Laplace Transform pairs for the impulse and step Green's function:

$$G_{temp,\delta}(t;z) = \frac{ze^{-z^2 t}}{2\sqrt{\pi t^3}} \overset{L.T.}{\longleftrightarrow} \hat{G}_{temp,\delta}(p;z) = e^{\sqrt{p}z}$$

$$\tag{39}$$

$$G_{temp,\Theta}(t;z) = 1 + erf\left(\frac{z}{2\sqrt{t}}\right) \overset{L.T.}{\longleftrightarrow} \hat{G}_{temp,\delta}(p;z) = \frac{e^{\sqrt{p}z}}{p}$$

In the context of the Earth's temperature, using heat conduction, (not temperature) boundary conditions, [*Brunt*, 1932] obtained the analogous classical formula noting that "this solution is given in any textbook".

These classical Green's functions provide useful comparisons with the conductive - radiative BC's. For example, integrating eq. 34 with respect to time and simplifying, we obtain:

$$\Delta G_\Theta(t;z) = G_{\Theta,temp}(t;z) - G_\Theta(t;z) = \frac{1}{\pi}\int_{-\infty}^{\infty}\frac{e^{-\zeta^2 t}e^{iz\zeta}d\zeta}{(1+i\zeta)}; \quad \begin{matrix} t \geq 0 \\ z \leq 0 \end{matrix} \tag{40}$$

Since the step response $G_\Theta$ describes the approach to thermodynamic equilibrium, $\Delta G_\Theta(t;z)$ (fig. 5) succinctly expresses the differences between the temperature and conductive - radiative forced boundary conditions. The leading large $t$

approximation to the integral in eq. 40 is $\Delta G_\Theta\left(t;z\right) \approx e^{-\frac{z^2}{4t}} / \sqrt{\pi t}$ so that as the figure shows, although they both slowly approach each other and eventually attain equilibrium, that the differences are important (especially in the diffusion layer, $z \approx$ <1) and they decay very slowly with time and depth, we discuss this further in section 3.3.

### 3.1.6 Surface temperatures, Fractional derivatives and the HEBE

Let us now introduce the $h^{th}$ order fractional derivative $_{t_0}D_t^h$ to represent the fractional derivative order $h$ of an arbitrary function $f$ over the domain from $t_0$ to $t$:

$$_{t_0}D_t^h f = \frac{1}{\Gamma\left(1-h\right)} \int_{t_0}^{t}\left(t-u\right)^{-h} f'\left(u\right)du; \qquad f'\left(u\right)=\frac{df}{du}; \qquad 0 \le h \le 1 \tag{41}$$

Fractional derivatives of order $h$ are most commonly interpreted in the Caputo (as above) or Riemann-Liouville sense ([*Podlubny*, 1999], for $h \le 1$, the main case of interest here, the distinction is not important) and they most commonly use $t_0 = 0$ in the above. Fractional derivatives and their inverses, fractional integrals (with $h<0$) are thus power law weighted convolutions; fractional integrals of noises are often associated with long memory stochastic processes. Many studies have found long memories in macroweather ([*Blender and Fraedrich*, 2003], [*Bunde et al.*, 2005], [*Rybski et al.*, 2006], [*Varotsos et al.*, 2013]) and a Gaussian noise forced model (fractional Gaussian noise) have been proposed as models of internally forced (macroweather) temperature variability ([*Rypdal and Rypdal*, 2014], [*Lovejoy*, 2015], [*Del Rio Amador and Lovejoy*, 2019], [*Del Rio Amador and Lovejoy*, 2021a]).

Most applications of fractional derivatives are for forcings that start at $t = t_0 = 0$ (i.e. $F = 0$ for $t \le 0$), see [*Miller and Ross*, 1993], [*Podlubny*, 1999] and are convenient for deterministic forcings, however they singularize $t = 0$ whereas we often wish to include periodic or statistically stationary internal stochastic forcings so that $F\left(-\infty\right)=0$ (or in the periodic case, the mean over a cycle = 0) is more convenient, in which case we take $t_0 = -\infty$ and hence $T_s\left(t = -\infty\right) = 0$ (or periodic). As discussed in [*Lovejoy*, 2019b], this corresponds to the semi-infinite range "Weyl" fractional derivative. Deterministic, stochastic and periodic forcings can be combined into a single framework simply by using the Weyl derivatives with for example the deterministic part of the forcing starting at $t = 0$ (with the deterministic $F(t) = 0$ for $t \le 0$) and the stochastic forcing at $t = -\infty$. These fractional derivatives have the following transformation properties:

$$_0D_t^H \overset{L.T.}{\leftrightarrow} p^H$$
$$_{-\infty}D_t^H \overset{F.T.}{\leftrightarrow} (i\omega)^H \tag{42}$$

Where $\omega$ is the Fourier conjugate to $t$, (see e.g. [*Miller and Ross*, 1993], [*Podlubny*, 1999]). In this part I (except for section 3.3), we consider deterministic forcings, putting $t_0 = 0$ in eq. 41, we using $_0D_t^{1/2} \overset{L.T.}{\leftrightarrow} \sqrt{p}$ ($h = 1/2$ in eq. 42), we obtain the HEBE for the surface temperature Green's function:

$$\left(_0D_t^{1/2} + 1\right)G_\delta(t;0) = \delta(t) \overset{(L.T.)}{\leftrightarrow} \left(\sqrt{p} + 1\right)\hat{G}_\delta(p;0) = 1 \tag{43}$$

This proves that the surface temperatures implied by the heat equation with conductive - radiative boundary conditions can be determined directly from the HEBE using the same Green's function. For the dimensional equations, the surface temperature therefore satisfies the dimensional HEBE:

$$\tau^{1/2}{}_0D_t^{1/2}T_s + T_s = sF(t); \qquad T_s(t) = \lambda\int_0^t G_\delta\left(\frac{t-u}{\tau};0\right)F(u)\frac{du}{\tau} \tag{44}$$

(where the surface temperature is $T_s(t) = T(t;0)$).

This HEBE equation for the surface temperature could be regarded as a significant nonclassical example of the Mori-Zwanzig formalism, ([*Gottwald et al.*, 2017], [*Mori*, 1965], [*Zwanzig*, 1973], [*Zwanzig*, 2001]), and empirical model reduction formalisms [*Ghil and Lucarini*, 2020], whereby memory effects arise if we only look at one part of the system, ignoring the others. In the HEBE, the surface temperature is analogously expressed directly in terms of the forcing, ignoring the subsurface
degrees of freedom. Although such memories are usually considered exponential and hence small, the HEBE shows that the classical continuum heat equation has on the contrary, strong power law memories. This points to serious limitations to conventional dynamical systems approaches to climate science that assume that the dynamical equations are integer ordered with exponential memories. The HEBE shows that the fundamental radiatively exchanging components of the climate system will generally be characterized by long memories, associated with fractional rather than integer ordered derivatives. We
develop this insight elsewhere.

### 3.2 The HEBE, zero dimensional and box models and Newton's law of Cooling

Phenomenological models of the temperature based on the energy balance across a homogeneous surface may represent either the whole earth or only a subregion. The former are global "zero dimensional" energy balance models (sometimes called "Global Energy Balance Models", GEBMs (see the review [*McGuffie and Henderson-Sellers*, 2005 ]) whereas in the

Field Code Changed

Field Code Changed

Field Code Changed

latter, they may represent the balance across the surface of a homogeneous subsection, a "box". The boxes have spatially uniform temperatures that store energy according to their heat capacity, density and size. Often several boxes are used, mutually exchanging energy, and the basic idea can be extended to column models. Since the average earth temperature can be modelled either as a single horizontally homogeneous box, or by two or more vertically superposed boxes, in the following, "box model" refers to both global and regional models.

A key aspect of these models is the rate at which energy is stored and at which it is exchanged between the boxes. Stored heat energy is transferred across a surface and it is generally postulated that its flux obeys Newton's law of cooling (NLC). The NLC is usually only a phenomenological model, it states that a body's rate of heat loss is directly proportional to the difference between its temperature and its environment. In these horizontally homogeneous models, it is only the heat energy/area (= $S$) that is important so that the NLC can be written:

$$Q_s = \frac{dS}{dt} = \frac{1}{Z}\left(T_{eq} - T\right) \tag{45}$$

$S$ is the heat in the body and $Q_s$ is the heat flux across the surface into the body (see fig. 6). $T_{eq}$ is the equilibrium temperature, and $Z$ is a transfer coefficient, the "thermal impedance" (units: $m^2 K/W$), its reciprocal $Y$ is the surface "thermal admittance" see the next section). Identifying the equilibrium temperature with $T_{eq}(t) = sF(t)$ and using the dimensional surface boundary condition (eq. 12), it is easy to check that a direct consequence of the HEBE's conductive - radiative boundary condition is that it also satisfies the NLC:

$$Q_{s,HEBE} = \frac{dS_{HEBE}}{dt} = \rho c \kappa_v \left.\frac{\partial T}{\partial z}\right|_{z=0} = \frac{\left(T_{eq} - T\right)}{s}; \qquad T_{eq} = sF \tag{46}$$

Unlike the usual phenomenological box applications that simply postulate the NLC, the HEBE satisfies it as a consequence of its energy conserving surface boundary condition. Comparing eqs. 41, 42, we may also conclude that thermal impedance $Z = s$.

While the HEBE and box models both obey the NLC, the relationships between the surface heat flux $Q_s = dS/dt$ and the surface temperature $T$ are quite different. For example, for forcings starting at time $t = t_0$, using the HEBE we have:

$$Q_{s,HEBE} = \frac{dS_{HEBE}}{dt} = \frac{\tau^{1/2}}{s} {}_{t_0} D_t^{1/2} T; \qquad \tau = \rho c s l_v; \quad l_v = \kappa_v \rho c s \tag{47}$$

Although this relation between surface heat fluxes and temperatures has been known for some time ([*Babenko*, 1986], [*Podlubny*, 1999], see e.g. [*Sierociuk et al.*, 2013], [*Sierociuk et al.*, 2015] for applications), to my knowledge, it has never

been applied to conduction - radiative models, nor has it been combined with the NLC to yield the homogeneous HEBE. In comparison, box models satisfy:

$$Q_{s,box} = \frac{dS_{box}}{dt} = \frac{\tau_{box}}{s}\frac{dT}{dt}; \quad \tau_{box} = \rho csL; \quad L = \frac{C}{\rho c} \tag{48}$$

Field Code Changed

Where $L$ is the effective thickness of the surface layer and $C$ is the specific heat per area, $\tau_{box}$ is the classical EBE relaxation time. [*Geoffroy et al.*, 2013] used a two box model to fit outputs of a dozen GCM and found $\tau_{box} \approx 4.1\pm1.1$ years (the mean and spread of 12 models) and $\approx 40$ - 800 years for the second box whereas the [*IPCC*, 2013] recommends a 2 box model with relaxation scales $\tau_{box} = 8.4$ and 409 years, with the FEBE, [*Procyk et al.*, 2020] finds $H = 0.38\pm0.05$, $\tau = 4.7\pm2.3$ years.

The HEBE and box heat transfer models can conveniently be compared and contrasted by placing them both in a more general common framework. Define the $h^{th}$ order heat storage as:

$$S_h(t) = \frac{\tau^h}{s\Gamma(1-h)}\int_{t_0}^{t} T(u)(t-u)^{-h}du; \quad 0 \le h \le 1 \tag{49}$$

If we take $T(t_0) = 0$ (this is equivalent to fixing the reference of our anomalies), then integrating by parts:

$$S_h(t) = \frac{\tau^{h-1}}{s\Gamma(1-h)}\int_{t_0}^{t} T'(u)(t-u)^{1-h}du; \quad 0 \le h \le 1 \tag{50}$$

Putting $h = 1$ yields the simple: $S_1(t) = T(t)/s$ so that $S_1 = S_{box}$.

Over the interval $t_0$ to $t$, the fractional derivative of order $h$ is defined as the ordinary derivative of the $1-h$ order (Reimann-Liouville) fractional integral:

$${}_{t_0}D_t^h T = \frac{d}{dt}\left({}_{t_0}D_t^{h-1}T\right) = \frac{d}{dt}\left[\frac{1}{\Gamma(1-h)}\int_{t_0}^{t}(t-u)^{-h}T(u)du\right]; \quad 0 \le h \le 1 \tag{51}$$

Therefore $S_{1/2} = S_{box}$ and:

$$\frac{dS_h}{dt} = s^{-1}\tau^h {}_{t_0}D_t^h T; \qquad \begin{matrix} h_{HEBE} = 1/2; & \tau_{FEBE} = l_v \rho cs \\ h_{box} = 1; & \tau_{box} = L\rho cs \end{matrix} \tag{52}$$

Combining this with the NLC, in both cases we obtain:

$$\tau^h \, {}_{t_0}D_t^h T + T = sF \tag{53}$$

Hence the box and HEBE models are special cases of the Fractional order Energy Balance Equation (FEBE [*Lovejoy*, 2019a], [*Lovejoy*, 2019b] derived phenomenologically in [*Lovejoy et al.*, 2021]). Whereas the box model changes its heat content instantaneously with its current temperature ($T(t)$), at any moment, the energy stored in the HEBE model depends on the past temperatures, and since their weights fall off slowly – there is a long memory – it potentially depends on the temperature and hence energy stored in the distant past. Box or column models all have surfaces that exchanges heat both radiatively and conductively so that – contrary to standard practice – these surfaces should instead exchange heat fractionally with $h = 1/2$ not $h = 1$. Note that when we consider box interfaces with purely conductive heat exchanges (without radiative transfer e.g. between a "deep ocean" and "mixed layer" in global two box model), then the thermal contact conductance that characterizes the interface is needed.

At a theoretical level, the advantage of the HEBE is that unlike the box models, it is a direct consequence of the standard (energy conserving) continuum heat equation combined with standard energy conserving surface boundary conditions. It is therefore natural to ask if the $h = 1$ heat transfer (i.e. $dS_1/dt = (C/s)dT/dt$) can be derived from the heat transport equation.

Returning to the nondimensional boundary condition ($\left.\dfrac{\partial T}{\partial z}\right|_{z=0} + T(t;0) = F(t)$) it is easy to verify, that in order to recover $H = 1$ heat transfer, one must instead use $\left.\dfrac{\partial^2 T}{\partial z^2}\right|_{z=0} + T(t;0) = F(t)$. We therefore conclude that box model $h = 1$ transfer is *not* simultaneously compatible with heat equation and energy balance boundary conditions.

To summarize: we are currently in the unsatisfactory position of having zero and one dimensional (box and Budyko-Sellers) energy balance equations neither of which satisfy the correct radiative - conductive surface boundary conditions. For the box models, the consequence is that the energy storage processes have rapid (exponential) rather than slow (power law) relaxation. For the Budyko-Sellers models, the consequence is that at best, they are 1-D and even with this restriction, their time dependent versions have derivatives of the wrong order (part II, section 2.3). In comparison, the zero dimensional HEBE is a consequence of correcting the Budyko-Sellers boundary conditions. It satisfies the NLC and corrects the order $H$ reducing it from the phenomenological value $h = 1$, to $h = 1/2$. As a bonus, in part II we see that the HEBE can easily be extended from zero to two spatial dimensions, enlarging the scope of energy balance models while simultaneously eliminating these weaknesses.

### 3.3 Thermal impedance and Complex climate sensitivities and the annual cycle

 **3.3.1 Conductive versus conductive - radiative boundary conditions**

Up until now, we have discussed forcing that is "turned on" at $t = 0$, this allowed for convenient solutions using Laplace transform methods. However, for forcing that is periodic or that is a stationary noise (i.e. the internal variability) Fourier techniques are more useful.

The first applications of Fourier techniques to the problem of radiative and conductive heat transfer into the Earth, was by [*Brunt*, 1932] and [*Jaeger and Johnson*, 1953] who considered the (weather regime) diurnal cycle. We already mentioned that [*Brunt*, 1932] also considered step function heat forcing, that he claimed might be a plausible model of the diurnal cycle near sunset or sunrise. However, in zero - dimensional models, the long time temperatures after step heat flux forcings are divergent (but not in 2D models, see part II) so that later in his paper Brunt considered periodic diurnal heat flux forcing with no net heat flux across the surface and used Fourier methods instead. In this classical diurnally forced problem, the periodic temperature response lags the forcing by a phase shift of $\pi/4 = 3$ hours. If we apply the same shift to the annual cycle – assuming that the Earth is forced by heat flux into its subsurface – the corresponding lag is 1.5 months $\approx$ 46 days which is generally too long (we shall see that it corresponds to an infinite relaxation time).

Following [*Brunt*, 1932] and [*Jaeger and Johnson*, 1953], let us consider the response to a single Fourier component forcing (this is equivalent to Fourier analysis of the equation). In this case, assuming a periodic temperature response and substituting this into the 1-D dimensional heat equation (time and depth, i.e. the dimensional version of eq. 22), we find that the variation of amplitude with depth is:

$$T\left(t;z\right) = T_s e^{i\omega t} e^{\sqrt{\frac{i\omega}{\kappa_v}}z}; \quad z \leq 0 \tag{54}$$

Where $T_s$ is the amplitude of the surface temperature oscillations, it depends on the nature of the forcing, here on the boundary conditions ("$s$" for "surface"). Following Brunt, using the classical heat surface heat forcing $F_s e^{i\omega t}$ as the surface boundary condition (with this forcing, $F_s = Q_s$ is the heat crossing the surface entering the system in the downward direction, see figs. 1, 6) we find:

$$\rho c \kappa_v \frac{\partial T_{heat}}{\partial z}\bigg|_{z=0} = F_s e^{i\omega t} \tag{55}$$

("heat" for heat forcing), we obtain:

$$T_{s,heat} = \frac{F_s}{\sqrt{i\omega(\rho c)^2 \kappa_v}} = Z(\omega)F_s; \qquad Z(\omega) = \frac{s}{\sqrt{i\omega\tau}} \qquad (56)$$

Where, $Z(\omega)$ is the complex frequency dependent thermal impedance, the reciprocal of the thermal admittance. For a given surface heat flux, $Z(\omega)$ quantifies the surface temperature response (we have written the impedance with the help of $s$ in order to nondimensionalize the denominator). Thermal impedance and admittance are standard in areas of heat transfer engineering and were introduced into the problem of diurnal Earth heating by [*Byrne and Davis*, 1980]. From $Z(\omega)$, we can thus easily understand the key [*Brunt*, 1932], [*Jaeger and Johnson*, 1953] result: that $\arg(Z(\omega)) = \arg(i^{-1/2}) = -\pi/4$ ("arg" indicates the phase).

So far, this approach has only been applied to weather scales (the diurnal cycle). Let's now apply the same approach but with an eye to longer macroweather timescales, notably the annual cycle. The climate sensitivity is an emergent macroweather quantity that is determined by numerous feedbacks that over the weather scales are quite nonlinear but over macroweather scales are considerably averaged (and at least for GCMs, [*Hébert and Lovejoy*, 2018]) are already fairly linear. In any event, for the annual cycle we use radiative - conductive boundary conditions rather than the pure conductive ones used by Brunt.

Using conductive - radiative surface BCs with external forcing $F_s e^{i\omega t}$ yields:

$$F_s = Q_s + Q_{s,rad} = s^{-1}\left(1 + (i\omega\tau)^{1/2}\right)T_s$$

$$Q_{s,rad} = s^{-1}T_s \qquad\qquad F(t) = F_s e^{i\omega t} \qquad (57)$$

$$Q_s = \rho c \kappa_v \left.\frac{\partial T}{\partial z}\right|_{z=0} = s^{-1}(i\omega\tau)^{1/2}T_s$$

Where here $F_s$ is the radiative (downward) forcing radiative flux and $Q_s$ and $Q_{s,rad}$ are the surface conductive (into the subsurface) and long wave radiative emission (away from the surface) fluxes respectively. Solving, we obtain the same depth dependence (eq. 54), but with the amplitude of the surface oscillations given by:

$$T_s = s(\omega)F_s; \qquad s(\omega) = Z(\omega) = \frac{s}{1+(i\omega\tau)^{1/2}} \qquad (58)$$

Where we have introduced the complex climate sensitivity $s(\omega)$ which by definition is equal to the complex thermal impedance $Z(\omega)$. In the context of the Earth's energy balance, it is more useful to think in terms of sensitivities than impedances so that below we use $s(\omega)$. With this, we obtain:

$$Q_s = \frac{s(\omega)}{s}\left(i\omega\tau\right)^{1/2} F_s; \qquad Q_{s,rad} = \frac{s(\omega)}{s} F_s \tag{59}$$

Since $\text{Arg}(i^{1/2}) = \pi/4 (= 45°)$, we see that as mentioned earlier, the conductive and long wave radiative fluxes are out of phase by 45°, but the phase of the temperature lags the forcing by $\arg(s(\omega))$, which only reaches 45° in the large $\tau$ limit (see fig. 7).

Note that we could have deduced eq. 59 directly by Fourier analysis of the HEBE using $F.T.\left(_{-\infty}D_t^{1/2}\right) = \left(i\omega\right)^{1/2}$, but the above allowed us to compare the results with the classical model. The Fourier method allows us to extend the complex climate sensitivity to the more general FEBE:

$$s_h(\omega) = \frac{s}{1 + \left(i\omega\tau\right)^h} \tag{60}$$

the usual EBE is the $h = 1$ special case.

### 3.3.2 Empirical estimates of complex climate sensitivities

Figs. 7, 8 compare the phases and amplitudes of $s(\omega)$ for the classical and conductive - convective boundary conditions ($h = 1/2$) HEBE as well as the $h = 1$ EBE. The plots use $\omega = 2\pi$ $rad/yr$. From fig. 7, we see that taking the empirical value $\tau \approx 4.7$ years ([*Procyk et al.*, 2020]), that the HEBE lag is a little over a month. From the detailed maps in [Donohoe et al., 2020] (see also [*Ziegler and Rehfeld*, 2020]) we estimate that in the extratropical regions, over land, the summer temperature maximum is typically 30 - 40 days after the solstice, but only 20 - 30 days after the maximum forcing (insolation) and for ocean, 60 - 70 days after the solstice but only 30 - 40 days after the maximum insolation. The HEBE result is thus close to the observed lag between the summer solstice and maximum temperatures over most land areas.

In contrast, if we use [*Brunt*, 1932]'s classical the heat forcing result we obtain $\pi/4 = 1.5$ months = 46 days which is already too long for most of the globe and the $h = 1$ EBE result (close to 3 months = 91 days) is much too long. Over the ocean, the lag is typically longer than over land probably because of the strong albedo periodicity associated with seasonal ocean cloud cover [Stubenrauch et al., 2006], [Donohoe et al., 2020]. This delays the summer solstice forcing maximum over the ocean, potentially explaining the extra lag.

Although a complete analysis with modern data is out of our present scope, we can get a feel for the realism of this approach by using the zonally averaged [*North and Coakley*, 1979] Sellers model discussed in the review [*North et al.*, 1981],

updated in [*North et al.*, 1983] where most of the earth follows the EBE phase lags of ≈90 days. The model uses a 2nd order Legendre polynomial to take into account the latitudinal variations and a sinusoidal annual cycle with empirically fit parameters that effectively zonally average over land and ocean. Empirical parameters are given for the albedo, top of the atmosphere insolation, temperature and outgoing IR radiation such that the global temperature maximum lags the solstice by 32.5 days [*North and Coakley*, 1979], [*North et al.*, 1983]. An updated 2-D version of the Sellers model has used it to estimate phase lags with respect to the solstice finding lags of ≈ 90 days over oceans and ≈30-40 days over land ([*Zhuang et al.*, 2017], [*Ziegler and Rehfeld*, 2020]).

Before continuing, recall that the zero-dimensional theory discussed here assumes that all radiative flux imbalances are all stored, it ignores the divergence of the horizontal heat transport which according to [*Trenberth et al.*, 2009] – is small even though the heat fluxes may be significant. Although at least for temperature anomalies, we argue that this effect is mostly important at small scales, the magnitude of horizontal heat divergence at macroweather scales is not well known and is presumably quite variable from place to place depending on (inhomogeneous) local transport parameters (see part II). A simple way to parameterize the transport is to maintain the assumption that the Earth has homogeneous parameters and to assume that the transport is due to horizontally inhomogeneous forcing. In part II, we show that for a horizontal wavenumber $k$, the effect

of horizontal transport is to modify the storage term as $\left(i\omega\tau\right)^{1/2} \to \left(i\omega\tau + \left(l_h k\right)^2\right)^{1/2}$ , therefore for pure periodic horizontal

forcing:

$$Q_{s,h} = \frac{s_h\left(\omega\right)\left(i\omega\tau + \left(l_h k\right)^2\right)^{1/2}}{s}F_s; \qquad Q_{s,rad} = \frac{s_h\left(\omega\right)}{s}F_s; \qquad s_h\left(\omega\right) = \frac{s}{1 + \left(i\omega\tau + \left(l_h k\right)^2\right)^{1/2}}$$

(61)

("*h*" for "horizontal inhomogeneity; in [*Lovejoy et al.*, 2021] there is an analogous calculation for the FEBE with $h \neq 1/2$). In North et al's 1-D model, the top of the atmosphere forcing is exactly a cosine variation i.e. with a single wavenumber $k = 1$ cycle around the Earth. The only differences are that we neglected the curvature of the Earth and assumed that the Earth's transport properties are constant. We nevertheless use eq. 61 as an approximation for the horizontal transport.

From the data in table 1 of [*North et al.*, 1981] , we may deduce:

$$\begin{aligned}
F_s &= \left(212 \pm 28\right)e^{-3.27i}\sin\theta; & W/m^2 \\
Q_{s,rad} &= 38e^{-3.65i}\sin\theta; & W/m^2 \\
T_s &= 15.5e^{-3.70i}\sin\theta; & K
\end{aligned}$$

(62)

Where the forcing $F_s$ is the product of the solar constant with the co-albedo (= 1- albedo) and θ is the latitude and the phases are taken with respect to the winter solstice. The variation (about ±13%) in the amplitude of $F_s$ is due to the latitudinal variation of the coalbedo. In the model, the long wave radiation $Q_{s,rad}$ and the surface temperature response $T_s$ have exact sinθ dependencies. The phases (in radians) are taken with respect to the winter solstice so that the summer solstice has a phase π = 3.14 rads, (in the northern hemisphere, June 21). Due to the coalbedo variations, the actual forcing has a phase = 3.27 rads peaking on June 28th. Also, the phase of the temperature and longwave emissions are larger = 3.70 rad, 3.65 rad corresponding to maxima on July 26th, July 23rd respectively (all results are appropriately symmetric for the southern hemisphere and for the cold lag following the winter solstice). The near identity of the phases of temperatures and long wave responses (a three day difference, probably not empirically significant), is already support for the model that predicts that they should be in phase. We also note that these lags (of 28, 25 days) are considerably shorter than the 46 day lag (Aug 12th) that would have been obtained had we applied Brunt's heat conductive forcing.

We can use these data to estimate the climate sensitivity, relaxation time τ and horizontal conduction term $l_h k$ by using the following:

$$s = \frac{T_s}{Q_{s,rad}} = 0.41 + 0.02i \approx 0.41; \quad K/\left(W/m^2\right)$$

$$s_h(\omega) = \frac{T_s}{F_s} = (0.068 \pm 0.009) + (0.031 \pm 0.004i); \quad K/\left(W/m^2\right)$$

$$(63)$$

$$i\omega\tau + (l_h k)^2 = \left(\frac{F_s}{Q_{s,rad}} - 1\right)^2 = (13.20 \pm 4.6) + (17.3 \pm 5.1)i$$

From this (with ω = 2π/yr), we obtain:

$$\tau = 2.75 \pm 0.8\, yrs$$
$$l_h k = 3.63 \pm 0.64$$

$$(64)$$

The relaxation time is within the rough bounds deduced by considering atmosphere - ocean coupling time scale (≈ 2 years, Hebert et al 2020), low frequency climate records (≈ 4.7±2.3 years, [*Procyk et al.*, 2020]), and the high frequency EBE relaxation times ≈ 4.1±1.1 years [*Geoffroy et al.*, 2013]. We also see that the ratio of the storage to transfer is 17.3/13.2 ≈ 1.3

so that most of the heat is indeed stored so that the above homogeneous theory is plausible. The nondimensional $l_h k$ characterizes the typical horizontal transport over the period of a year. Rather than interpreting it deterministically in terms of a global scale horizontal variation over a homogeneous earth, we consider it a nondimensional empirical parameter that we

will try to clarify in future work. In any case, the horizontal transport and storage are in quadrature so that the effect of the transport on the magnitude of sensitivity is smaller: $\left|(i\omega\tau)^{1/2}+1\right|/\left|\left(i\omega\tau+(l_h k)^2\right)^{1/2}+1\right|\approx 0.88$ (i.e. about 12%) but the change in

the phase is more substantive ($\approx$ 15 days). We can note that the EBE $h = 1$ value (ignoring transport, with $\tau = 2.75$ years) gives 87 days i.e. a maximum on September 21$^{st}$ which is much too late (fig. 7).

The static climate sensitivity $s$ should be purely real; its imaginary part is indeed small, it corresponds to 3 days and is

650 probably within the error of the model and empirical estimates, it will be ignored below. $s$ can be converted to $K/(CO_{2eq}$ *doubling*) by multiplying it by the canonical value 3.71 $W/m^2/$ (*$CO_{2eq}$ doubling*) to yield 1.51 $K/(CO_{2eq}$ *doubling*) which is at the lower part of the IPCC 90% confidence range ($3\pm1.5$ K/ (*$CO_{2eq}$ doubling*)). Since both the methodology and the empirical parameter estimates could be updated and improved, the result is encouraging. In future, instead of assuming latitudinal constancy with a sinusoidal latitudinal dependence, gridded data could be used and the horizontal conduction approximation

(the $l_h k$ term) could be improved.

## 4. Conclusions

This first paper of two parts proposes a new 2D energy balance equation for macroweather scales: ten days and longer. It follows the classical energy balance models pioneered by [*Budyko*, 1969] and [*Sellers*, 1969], and assumes that the dynamics can be adequately modelled by the continuum mechanics heat equation – by advection and diffusion. As reviewed in

[*McGuffie and Henderson-Sellers*, 2005 ], [*North and Kim*, 2017], the classical models treat the parts of the atmosphere and ocean that radiatively interact with outer space as a zero thickness, two dimensional surface. The complex radiative processes that occur in the vertical direction are only treated implicitly. The dimensionality is then further reduced by zonal averaging.

While this original time independent model may be reasonable for the long term (time invariant) climate states, it is inadequate for treating time varying anomalies. The key improvement in realism was by made explicitly introducing a vertical

coordinate $z$. Yet, when this was done, it turned out that a detailed vertical model was still unnecessary: all that was required was the existence of a surface layer whose thickness was of the order of the diffusion depth. This is where most of the energy storage occurs and it determines the vertical temperature derivative at the surface and hence the vertical conductive heat flux. This sensible heat flux is the crucial link between the local radiative imbalances that drive the system, the heat that is stored and the heat that is transported horizontally. Whereas the Budyko-Sellers models have zero thicknesses, our model has a

finite but possibly small thickness; it need only be thick enough to account for energy storage and to determine the surface vertical temperature derivative.

In this first part, we considered only homogeneous zero-dimensional models. These are completely classical, yet as far as we know, have not been solved with conductive – (linearized) radiative boundary conditions. Using standard Laplace and Fourier techniques, we solved the full depth-time heat equation and showed that it's Green's function was identical to a half-order fractional differential equation that directly gives the surface temperature. Although half-order derivatives have occasionally been used in the context of the heat equation, (at least since [*Oldham and Spanier*, 1972; *Oldham and Spanier*, 1974], [*Babenko*, 1986]), the resulting half-order energy balance equation (the HEBE) is apparently new. Mathematically, the result is a direct consequence of the heat equation, the semi-infinite medium and conductive - radiative surface boundary conditions.

The consequences are surprisingly far reaching. For example, the familiar integer ordered differential equations have exponential Green's functions, short memories. In contrast, the more general fractional ordered equations such as the HEBE have Green's functions that are "generalized exponentials", based on power laws and long memories. A general consequence is that while the HEBE respects Newton's law of cooling - i.e. that heat fluxes across a surface are proportional to temperature differences - that the relationship between this heat flux and the surface temperature is quite different: it involves a half order derivative rather than first order one. The energy stored is no longer instantaneously determined by the surface temperature, but rather by the entire prior forcing history. Irrespective of the details, we thus *expect* Earth heat storage processes to generally have long memories.

We also obtained general results on the Earth's response to periodic forcings. Ever since [*Brunt*, 1932], Fourier techniques have used the heat equation to model the Earth's temperature response when subjected to a diurnal heat flux forcing. We extend this from the weather regime to macroweather regime, from diurnally periodic heat forcing to annually periodic radiative - conductive forcing. An immediate consequence is that the surface thermal impedance - equal to the climate sensitivity – is a complex number whose phase determines the lag between the maximum of the forcing (shortly following the summer solstice) and the temperature maximum. Using a simple latitudinally averaged model with empirical parameters, we estimated this complex climate sensitivity and showed how this could readily account for the observed 22-25 day lag, estimating the (static) climate sensitivity at $s \approx 0.41$ K/(W/m$^2$) and relaxation time $\tau \approx 2.75$ years.

In part II, we extend these zero dimensional results to the horizontal. We first continue to use Laplace and Fourier techniques to treat the case of homogenous Earth parameters, but with inhomogeneous forcing. We then – with the help of Babenko's method, extend this to the full inhomogeneous problem with horizontally varying relaxation times, diffusivities, specific heats, climate sensitivities and forcings.

## 5. Acknowledgements

I acknowledge discussions with L. Del Rio Amador, R. Procyk, R. Hébert, D. Clarke and C. Penland. This is a contribution to fundamental science; it was unfunded and there were no conflicts of interest.

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

**Figures**

## The 3D Energy Balance

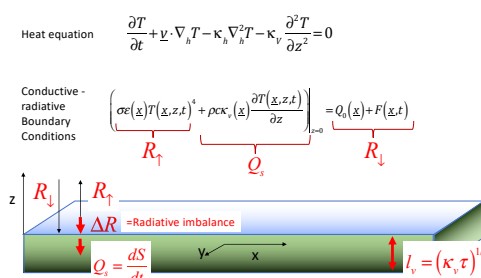

Heat equation
$$\frac{\partial T}{\partial t} + \underline{v} \cdot \nabla_h T - \kappa_h \nabla_h^2 T - \kappa_v \frac{\partial^2 T}{\partial z^2} = 0$$

Conductive - radiative Boundary Conditions
$$\left( \underbrace{\sigma\varepsilon(\underline{x})T(\underline{x},z,t)^4}_{R_\uparrow} + \underbrace{\rho c \kappa_v(\underline{x})\frac{\partial T(\underline{x},z,t)}{\partial z}}_{Q_s} \right)_{z=0} = \underbrace{Q_0(\underline{x}) + F(\underline{x},t)}_{R_\downarrow}$$

$R_\downarrow \quad R_\uparrow$

$\Delta R$ =Radiative imbalance

$$Q_s = \frac{dS}{dt}$$

$$l_v = (\kappa_v \tau)^{1/2}$$

**Fig. 1: A schematic diagram showing the correct 3D energy balance equations with conductive - radiative surface boundary conditions.** $Q_s$ **is the heat flux across the surface into the subsurface,** $S$ **is the energy stored in the subsurface per unit surface area. The picture illustrates the thin surface layer (whose thickness is of the order of the diffusion depth,** $l_v$ **with relaxation time** $\tau$**, eq. 20) in which the radiative exchanges between the earth and outer space occur.**

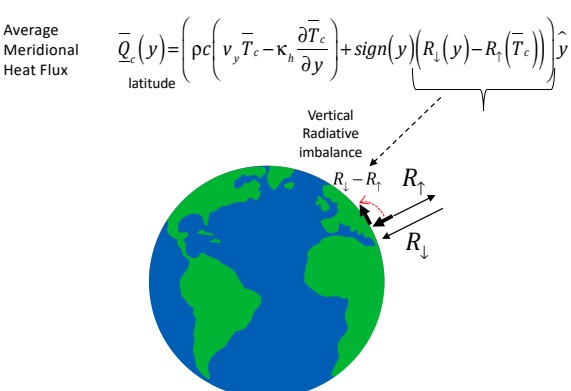

# Budyko-Sellers  3D ➝ 2D ➝ 1D

Average Meridional Heat Flux

$$\overline{\underline{Q}}_c(y) = \left(\rho c\left(v_y \overline{T}_c - \kappa_h \frac{\partial \overline{T}_c}{\partial y}\right) + sign(y)\left(R_\downarrow(y) - R_\uparrow(\overline{T}_c)\right)\right)\hat{y}$$

latitude

Vertical Radiative imbalance

$R_\downarrow - R_\uparrow$  $R_\uparrow$

$R_\downarrow$

**Fig. 2: A schematic diagram showing the Budyko-Sellers 1D energy balance equation obtained by latitudinal averaging and by redirecting the vertical imbalance away from the equator.**

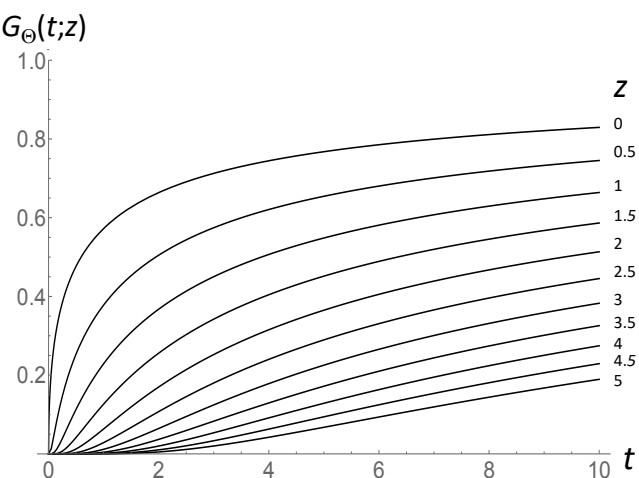

Fig. 3: The nondimensional temperature as a function of nondimensional time for various nondimensional depths with a step
forcing; $G_{\Theta}(t;z)$ (obtained by integrating eq. 34 in time). The (top) surface curve can be interpreted as the fraction of the forcing
that is conductive. At first all the forcing is conductive with no radiation, eventually all the fluxes are radiative, the system reaches
a new thermodynamic equilibrium and there is no conductive heat flux.

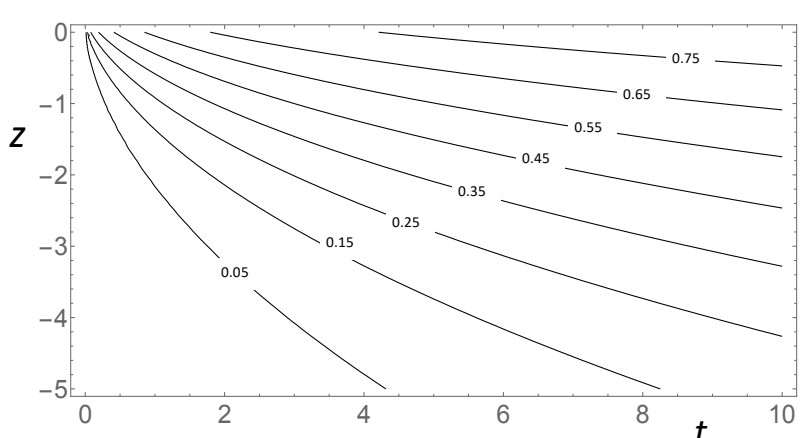

**Fig. 4: Contours of nondimensional temperature as a function of nondimensional time and depth after a step function forcing**
**($G_\Theta(t;z)$).**

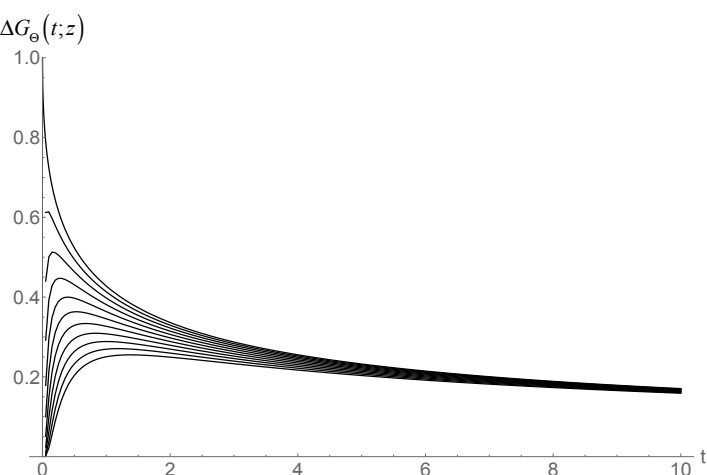

**Fig. 5: The difference** $\Delta G_{\Theta}(t;z)$ **between the classical (temperature forced) and radiative forced step response functions over the diffusion depth (nondimensional $z = 0$ to -1). The top is shows the surface ($z = 0$), the curves from top to bottom are at** 855 **depths $z$ =0., -0., -0.2, -0.3,…-1. While the difference is large over the relaxation time (up to nondimensional $t = 1$), we see that they both slowly converge to thermodynamic equilibrium at large $t$.**

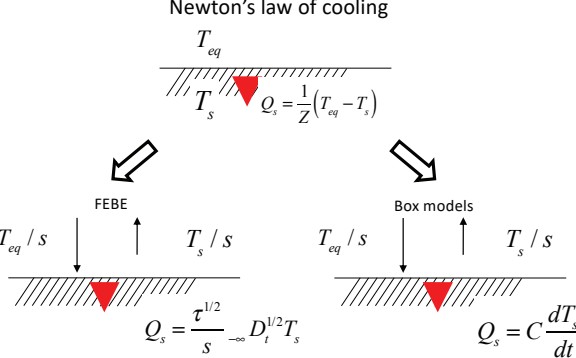

Fig. 6: A schematic showing Newton's law of cooling (NLC) that relates the temperature difference across a surface to the
heat flux crossing the surface, $Q_s$ (into the surface). $T_{eq}$ is the fixed outside temperature, heat will flow as long as the surface
temperature $T_s \neq T_{eq}$, $Z$ is the thermal impedance (equal here to the climate sensitivity $s$). To apply the NLC, we need to relate the
heat flux to the surface temperature. The lower left shows the consequence of applying heat equation with conductive – radiative
BC's, the lower right shows the phenomenological assumption made by box models. The arrows represent heat fluxes, hence the
factor $s$ in the denominators. The system is assumed to be horizontally homogeneous and that the subsurface is much thicker than
the diffusion depth.

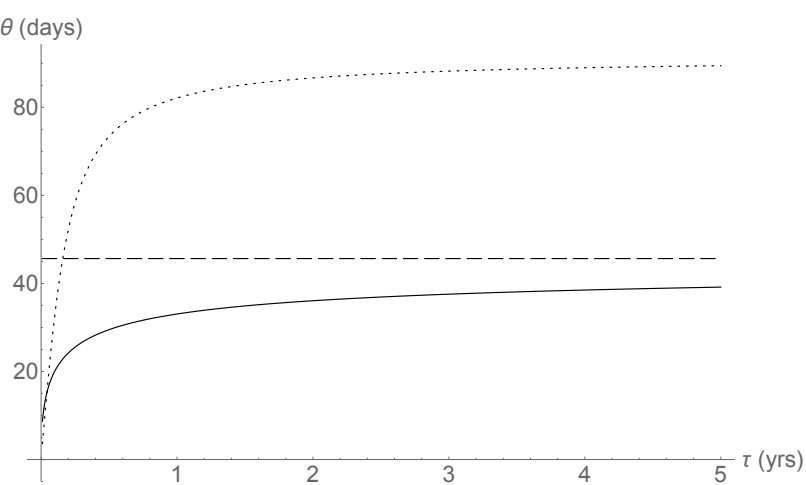

Fig. 7: The temperature phase lag (in months, the negative of argument of the complex climate sensitivity), using the complex climate sensitivity and annual cycle forcing (i.e. with $\omega = 2\pi$ *rads/yr*) with $\tau$ in years. The line with short dashes (top) is the usual EBE ($H = 1$), the solid line is the ($H = 1/2$) HEBE and the line with long dashes is the classical heat forcing model which is the large $\tau$ HEBE limit. All curves ignore any net horizontal heat transport. The data analyzed here yield $\tau \approx 2.75\pm0.8$ years but the actual phase is somewhat shorter due to horizontal heat transport.

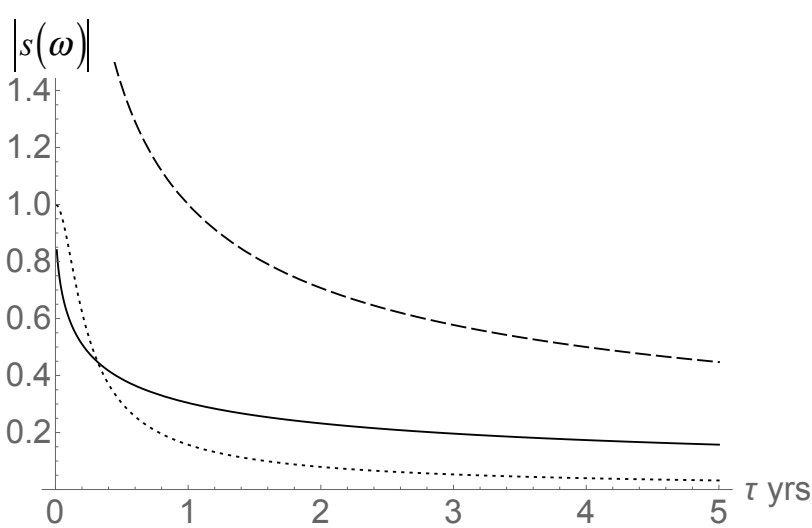

Fig. 8: Same as fig. 7 except for the amplitude of the complex climate sensitivity to annual cycle forcing (i.e. with $\omega = 2\pi$ *rads/yr*) with $\tau$ in years.  The short dash line (bottom) is the usual EBE ($h = 1$), the top line with long dashes is the classical heat forcing model and the solid line is the ($h = 1/2$) HEBE.