# Peer review of "The Half-order Energy Balance Equation, Part 1: The homogeneous HEBE and long memories"

_Earth System Dynamics, 2020_

## Referee Comment (RC1) · Peter Ashwin (Referee) · 15 Jun 2020

This is an interesting and innovative manuscript that proposes the appropriate energy balance model that relates heat (S) and surface temperature (T) should involve a half order time derivative of T. It is a half-order energy balance equation (HEBE), a special case of a fractional order energy balance equation (FEBE) rather than the usual full order time derivative traditionally used for box (0D) and Budyko-Sellers (1D) models. The author convincingly argues that it this model is appropriate for longer timescale (10 day or more) variability, both empirically and from physical principles. This has consequences in expecting a longer memory of imposed forcing than one would expect of an integer order EBE; more precisely the response to step forcing has power law rather than exponential decay. The derivation assumes forcing at a conductive-radiative

boundary condition and advection-diffusion of heat a semi-infinite domain: by using a Laplace-Fourier analysis the author obtains an integral form for the surface temperature that can be interpreted as a solution of a fractional differential equation. The case of periodic (annual/diurnal) forcing also considered and the surface thermal impedance is interpreted as a complex climate sensitivity – this is used to account for the observed phase lag between summer maximum forcing and surface maximum temperature.

---

## Referee Comment (RC2) · Anonymous Referee #2 · 28 Jun 2020

Review of "The Half-order Energy Balance Equation, Part 1: The homogeneous HEBE and long memories" by Lovejoy

Recommendation: Major revisions

This study derived a new version of the energy balance model based on non-integer derivatives. These models seamlessly contain long memory characteristics. This manuscript might be acceptable for publication in ESM after a major revision.

1) Certain parts of the paper are confusing. For instance, the model is called a "zero dimensional" model though it has a vertical dimension. I assume this is because traditionally the vertical axis has been neglected and only a horizontal average considered. I strongly suggest to find a different terminology for this.

[Figure]

2) You refer many times to Part II. I think this is distracting; in my opinion it would make the paper easier to read to remove those references or to just have a short outlook on Part II in the conclusions section.

3) Is your approach valid for all time scales? A long memory climate response should lead to infinite climate sensitivity. So your climate response operator is probably only valid for certain time scales.

4) Line 15: BC needs to be defined.

5) Line 26: I do not think "macroweather" is a widely known term. So please define.

6) Line 32: "latitudinally" probably should be "zonally"

7) I am confused by the z-coordinate system. It is not clear to me what z=0 means? Surface or top of the atmosphere? Also all z values seem to be negative. Also Figure 1 does not help at all in that respect.

8) Line 175: Your linearization is either accurate or not, but not both.

9) Line 266: What do you exactly mean by "top"?

10) in (33) you develop an asymptodic expansion. Why do you stop at the $\frac{1}{2}$ term? There are also higher order term which might lead to different orders on fractional derivatives.

11) Line 350: I am not sure many ESM readers are very familiar with long memory. I suggest that explain why (37) implies long memory.

---

## Author Comment (AC1) · 29 Jun 2020

Earth Syst. Dynam. Discuss.,https://doi.org/10.5194/esd-2020-12-RC1, 2020[©] Author(s) 2020. This work is distributed underthe Creative Commons Attribution 4.0 License.Interactive comment on"The Half-order EnergyBalance Equation, Part 1: The homogeneous HEBE and long memories" by Shaun Lovejoy

Peter Ashwin (Referee) p.ashwin@exeter.ac.uk

This is an interesting and innovative manuscript that proposes the appropriate energy balance model that relates heat (S) and surface temperature (T) should involve a half order time derivative of T. It is a half-order energy balance equation (HEBE), a special

case of a fractional order energy balance equation (FEBE) rather than the usual full order time derivative traditionally used for box (0D) and Budyko-Sellers (1D) models. The author convincingly argues that it this model is appropriate for longer timescale(10 day or more) variability, both empirically and from physical principles. This has consequences in expecting a longer memory of imposed forcing than one would expect of an integer order EBE; more precisely the response to step forcing has power law rather than exponential decay. The derivation assumes forcing at a conductive-radiative boundary condition and advection-diffusion of heat a semi-infinite domain: by using a Laplace-Fourier analysis the author obtains an integral form for the surface temperature that can be interpreted as a solution of a fractional differential equation. The case of periodic (annual/diurnal) forcing also considered and the surface thermal impedance is interpreted as a complex climate sensitivity – this is used to account for the observed phase lag between summer maximum forcing and surface maximum temperature.

Author: We thank the referee for his strong, positive review. As far as I can tell, he has understood the paper very well. He has no specific suggestions for changes.
* * *

---

## Author Comment (AC2) · 29 Jun 2020

ESDDInteractivecommentPrinter-friendly versionDiscussion paper Earth Syst. Dynam. Discuss., https://doi.org/10.5194/esd-2020-12-RC2, 2020© Author(s) 2020. This work is distributed underthe Creative Commons Attribution 4.0 License.
Review of "The Half-order Energy Balance Equation, Part 1: The homogeneous HEBE and long memories" by Lovejoy

Recommendation: Major revisions

This study derived a new version of the energy balance model based on non-integer derivatives. These models seamlessly contain long memory characteristics. This manuscript might be acceptable for publication in ESM after a major revision.

Author: We thank the referee for his/her comments that suggest a few clarifications. These are indicated in the detailed responses below (in italics).

1) Certain parts of the paper are confusing. For instance, the model is called a "zero dimensional" model though it has a vertical dimension. I assume this is because traditionally the vertical axis has been neglected and only a horizontal average considered. I strongly suggest to find a different terminology for this.

Author: We apologize for the admittedly confusing jargon, but we did not invent it! "Zero-dimensional" is the standard term for climate models without HORIZONTAL degrees of freedom. We do indicate this but we will gladly underline it and use alternative expressions when possible.

2) You refer many times to Part II. I think this is distracting; in my opinion it would make the paper easier to read to remove those references or to just have a short outlook on Part II in the conclusions section.

Author: We apologize if references to the second part of the paper are distracting. Many of these references were added after the initial submission at the explicit request of the editor Anders Levermann who thought that the linkage between the two parts was not strong enough. Since the editor was mostly concerned about adding linkages near the beginning of the paper, I could try to remove a few later on, although most of the references to the second part are quite pertinent.

The specific correspondence is on the site, I reproduce it here:

Editor Initial Decision: Start review and discussion after technical corrections (02 Apr 2020) by Anders Levermann Comments to the Author: Dear Shaun See my comment

to part no. 2. The two papers need to be clearly linked. Bests, Anders

The initial comment in part II alluded to above: Editor Initial Decision: Start review and discussion after technical corrections (19 Mar 2020) by Anders Levermann Comments to the Author: Dear Shaun, you have to reference the first part of the paper clearly in the very beginning of the paper, so that the reader can easily find it. I would actually prefer if you could reference it already in the abstract. I did not look very hard, but I was not able to find the reference to the part 1 in the paper. Please help us here. Bests, Anders

3) Is your approach valid for all time scales? A long memory climate response should lead to infinite climate sensitivity. So your climate response operator is probably only valid for certain time scales.

Authors: As discussed in the paper, while the model itself may well be valid over a very wide range of time scales, it has two regimes: one shorter than the relaxation time and one longer. Both regimes are scaling and therefore both could be considered to have long memories. However there is a common - but restrictive - definition of long memory processes that is often applied to Gaussian processes (a divergent integral time scale). If this definition is used for the HEBE, and the forcing is assumed to be a Gaussian white noise, this definition will only apply to the scales below the relaxation scale. According to this definition, the different long-time scaling regime has short memory. Therefore we will clarify this distinction in the revised manuscript.

4) Line 15: BC needs to be defined.

Author: OK.

5) Line 26: I do not think "macroweather" is a widely known term. So please define.

Author: OK.

6) Line 32: "latitudinally" probably should be "zonally"

Author: OK.

7) I am confused by the z-coordinate system. It is not clear to me what z=0 means? Surface or top of the atmosphere? Also all z values seem to be negative. Also Figure1 does not help at all in that respect.

Author: On line 114 it was stated:

"We consider that vertical (radiative and conductive), and horizontal (conductive and advective) heat transport occurs on the surface and in the half-volume (x,y,z<0) respectively. Although physically, this means that the atmosphere and ocean are modelled as regions with z≤0, as mentioned, only the vertical surface temperature derivative is ultimately needed and this is well defined if the surface layer is of the order of a few diffusion depths (hundreds of meters)."

As for figure 1, it clearly shows the positive z direction as "up" with radiation only in this region and with heat conduction into the z< region. Could the referee be more specific about how to clarify this further?

In any case, I will add a short discussion about the physical meaning of z=0: the surface.

8) Line 175: Your linearization is either accurate or not, but not both.

Author: I will rework the sentence.

9) Line 266: What do you exactly mean by "top"?

Author: I mean at z = 0. However this was already stated in the parentheses following the word "top":

"At the top (z = 0), the system is forced by the conductive - radiative surface boundary condition. . ."

The sentence could be reworked to make this clearer.

10) in (33) you develop an asymptotic expansion. Why do you stop at the 1/2 term? There are also higher order term which might lead to different orders on fractional derivatives.

Author: Eq. 33 does not stop at $\frac{1}{2}$ order terms but rather at orders 3/2, 5/2 (G0,1/2), 3, 3/2 (G1,1/2), 3, 3/2 (G2,1/2). In any case, I could easily have given the general nth order term since it is in the literature. The high order terms are simply high and low frequency corrections to the scaling - they do not define their own separate scaling regimes. I will state this in the revised ms. However, the high and low frequencies are dominated by the $\frac{1}{2}$ order part and this is supported by empirical analyses performed prior to the discovery of the HEBE. Indeed, the text immediately following eq. 33 states this:

"The asymptotic equation for the step response (G1,1/2) shows that thermodynamic equilibrium is approached slowly: as t-1/2. It is this power law step response (with empirical exponent 0.5±0.2) that was discovered semi-empirically by [Hebert, 2017], [Lovejoy et al., 2017] and was successfully used for climate projections through to 2100. Similarly, $\approx$ t0.4 behaviour was used for macroweather (monthly, seasonal) forecasts close to the short time t1/2 expansion [Lovejoy et al., 2015], [Del Rio Amador and Lovejoy, 2019]."

11) Line 350: I am not sure many ESM readers are very familiar with long memory. I suggest that explain why (37) implies long memory.

Author: Eq. 37 is simply the definition of a fractional derivative. Since such derivatives are based on power laws, it is common for fractional derivatives to be used in the context of long memory processes. I will add some material to clarify this.

---

## Author Response (AR1)

10

**Anonymous Referee #1**

Received and published: 10 July 2020

General comments: I think this is a notable (two-part) paper. Its key message, that the heat flux 15 at the earth's surface is a derivative of order half of the temperature, and that this modifies the simplest EBMs in an important way is both significant in itself, and provides a foundation for the author's concurrent work on fractional stochastic energy balance models.

Au: Thank you for the enthusiastic review!

20

I have only one gripe that needs attention. It relates to earlier work which needs to be more fully described and integrated into the manuscript. When this is done so it will actually reinforce the author's message, I think.

25 Au: The Oldham references are quite useful, thanks! I respond in more detail below.

**Specific comment: Earlier work on half-order derivatives in heat transfer**

- The list of references on fractional calculus seems to me to be comprehensive in general, but
  to be missing a key reference. Podlubny [1999] notes in his preface that:... from the viewpoint of applications in physics, chemistry and engineering it was undoubtedly the book written by K.
  B. Oldham and J. Spanier [i.e. "The Fractional Calculus", Academic Press, 1974; now in a Dover Edition] which played an outstanding role in the development of the subject which can be called applied fractional calculus. Moreover, it was the first book which was entirely devoted to a
- 35 systematic presentation of the ideas, methods, and applications of the fractional calculus.

Referring back to this book suggests to me that to say, as the manuscript presently does, that"... half-order derivatives have occasionally [sic] been used in the context of the heat equation, (at least since [Babenko, 1986]) "substantially underestimates the extent to which half order derivatives have already been studied in the heat equation context. Oldham and Spanier devote

40 derivatives have already been studied in the heat equation context. Oldham and Spanier devote their chapter 11 to applications of what they call the semi differential operator, i.e. the fractional derivative of half order, to diffusion problems including heat transfer.

The book built on their own papers, particularly Oldham KB, Spanier J (1972) A general solution
of the diffusion equation for semi infinite geometries, J Math Anal Appl 39:665–669 and Oldham KB (1973) Diffusive transport to planar, cylindrical and spher-ical electrodes, J Electroanal Chem Interfacial Electrochem, 41:351–358. They give the diffusion equation as:

$$\frac{\partial}{\partial t}F(\xi,\eta,\zeta,t) = \kappa \nabla^2 F(\xi,\eta,\zeta,t) \tag{1}$$

- 50 and then note that in three special, semi-infinite, cases this can be simplified so that Laplacian depends only on the radial co-ordinate r and t. In the planar case they give:
  - 2

$$\frac{\partial}{\partial t}F(r,t) - \kappa \frac{\partial^2}{\partial r^2}F(r,t) = 0$$
(2)

They take the system is initially in equilibrium F(r,t) =F0, for t <0,r≥0. An unspecified perturbation occurs at t= 0, and for times of interest t <0 it does not affect regions remote from the r= 0 boundary. Hence F(r,t) =F0, for t≤r,r=∞, and in the case of planar geometry they derive the solution:

$$\frac{\partial}{\partial r}F(r,t) = -\frac{1}{\sqrt{\kappa}}\frac{\partial^{1/2}}{\partial t^{1/2}}F(r,t) + \frac{F_0}{\sqrt{\pi\kappa t}}$$
(3)

They then go on to consider the problem of 1D heat conduction in a semi-infinite plane, and so look at the heat equation in the form:

$$\frac{\partial}{\partial t}T(r,t) - \frac{K}{\rho\sigma}\frac{\partial^2}{\partial r^2}T(r,t) = 0$$
(4)

with appropriate boundary conditions of T(r,0) = 0 and  $T(\infty,t) = 0$ . The heat flux sought is

$$J(t) \equiv -K\frac{\partial}{\partial r}T(0,t)$$
(5)

65 which they get from their earlier solution for  $\partial F(r,t)/\partial r$  by putting T for F,K/p $\sigma$  for  $\kappa$ , and using

$$J(T) = -K\frac{\partial}{\partial r}T(0,t) = \sqrt{K\rho\sigma}\frac{\partial^{1/2}}{\partial t^{1/2}}T(0,t)$$
(6)

Because this result, Oldham and Spanier's equation 11.2.10 is closely related to equation 43 in part I of the present ms, I think that it should be explained clearly whether i) the present paper

is effectively an illustration of Oldham and Spanier's result in the EBM context, or ii) whether it offers a derivation in a domain to which Oldham and Spanier's result did not apply. Either situation will be important and publishable but readers need to know which applies. Interestingly, Oldham and Spanier noted that the equation had been obtained by Meyerin 1960 in a Canadian NRC technical report ("A heat-flux-meter for use with thin film surface thermometers"), but rather than being written as a half order derivative it was then given in the alternative integral form:

$$J(T) = \sqrt{\frac{K\rho\sigma}{4\pi}} \left[\frac{2T(0,t)}{\sqrt{t}} + \int_0^t \frac{T(0,t) - T(0,\tau)}{\sqrt{t-\tau}} d\tau\right]$$
(7)

without explicitly using fractional calculus. It was thus known in the heat transfer context even before the first EBMs were derived, in a sense reinforcing the present author's point.

80

75

Au: There are several important differences w.r.t. to Oldham's results.

part 2 (eq. 3 and later) is outside his scope.

a) Oldham considers only a single spatial degree of freedom r corresponding to either the "zerodimensional" model (eq. 22 part 1) or cylindrical or spherical geometries that we do not consider.
85 He nowhere considers fractional space-time operators as in part 2. I.e. he neither treats homogeneous operators but with inhomogeneous boundary conditions, nor does Oldham treat inhomogeneous media (inhomogeneous transport operators). In other words essentially all of

90 b) Our boundary radiative-conductive boundary conditions are special cases of "Robin" boundary conditions i.e. they involve a linear combination of the field and it's normal gradient over a surface. Although Robin boundary conditions are occasionally used in insulating boundary condition problems in convective diffusive equations, they are not identical to the radiative-conductive conditions used here. Oldham mentions Cauchy, Neumann and Dirichlet

95 boundary conditions and says that "any other type" could be used. In other words he realized that his formalism was more general than the applications he developped, but did not pursue these. I will add this information in the revised ms.

c). Although it is not essential, Oldham's application of the method was to use more or less
 standard boundary conditions (Dirichlet) and then deduce the heat flux across surfaces from
 this. As far as I can tell, since then, this is almost invariably the way the method has been applied.

d) A final more minor difference is that we also treated the Weyl derivative and used the 105 corresponding Fourier techniques.

5

We added references to these difference in the new ms.

**110 Anonymous Referee #2**

Received and published: 13 July 2020

This second part reviewed here extends the approach of Part 1 to higher spatial dimension and inhomogeneous thermal models of the earth's response to radiative forcing. There is an appropriate summary of Part 1 that puts the new contribution into context. The full model considered here includes varying horizontal and vertical thermal diffusivities, thermal capacities, sensitivities and spatio-temporal forcing. By a heuristic method of Babenko, the author expands the inhomogeneous operator to give 2D energy balance equations that will be useful for studying

- 120 spatio-temporal responses to forcing. The manuscript includes a number of appendices that examine horizontal structures, cross-correlations, space-time factorization of quantities such as autocorrelation and that extends the results from flat space to the sphere. The analysis seems to be carefully done, and care is taken to distinguish cases where there may not be a rigorous justification.
- 125

Au: I thank the referee for the very positive review!

I would be interested to see a bit more discussion of the "bottom boundary condition" T=0 at z=-infinity. I think it would also be useful to include some discussion of how atmosphere/ocean convection is/is not represented in the model.

Au: The role of the bottom boundary condition was addressed in part I where (just after eq. 29) it is shown that the influence of the bottom BC decays exponentially quickly with depth so that below a few diffusion depths it is essentially irrelevant. In oceans this would likely imply depths

135 of hundreds of meters. In part I I added some new material clarifying the nature of the surface.

**The Half-order Energy Balance Equation, Part 2: The inhomogeneous HEBE and 2D energy balance models**

| 140 | Shaun Lovejoy                                                    |                        |
|-----|------------------------------------------------------------------|------------------------|
|     | Physics dept., McGill University, Montreal, Que. H3A 2T8, Canada |                        |
|     | Correspondence: Shaun Lovejoy (lovejoy@physics.mcgill.ca)        |
Field Code Changed |

Abstract: In part I, we considered the zero-dimensional heat equation showing quite generally that conductive - radiative 145 surface boundary conditions lead to half-ordered derivative relationships between surface heat fluxes and temperatures: the Half-ordered Energy balance Equation (HEBE). The real Earth - even when averaged in time over the weather scales (up to  $\approx$  10 days) – is highly heterogeneous, in this part II, we thus extend our treatment to the horizontal direction. We first consider a homogeneous Earth but with spatially varying forcing, both on a plane and also on the sphere: we compare our new equations with the canonical 1-D Budyko-Sellers equations. Using Laplace and Fourier techniques, we derive the Generalized HEBE 150 (the GHEBE) based on half-ordered space-time operators. We analytically solve the homogeneous GHEBE, and show how these operators can be given precise interpretations.

We then consider the full inhomogeneous problem with horizontally varying diffusivities, thermal capacities, climate sensitivities and forcings. For this we use Babenko's operator method which generalizes Laplace and Fourier methods. By expanding the inhomogeneous space-time operator at both high and low frequencies, we derive 2-D energy balance equations that can be used for macroweather forecasting, climate projections and for studying the approach to new (thermodynamic equilibrium) climate states when the forcings are all increased and held constant.

**1** Introduction**

In part I, we showed that when the surface of a body exchanges heat both conductively and radiatively, that its flux depends on the half order derivative of the surface temperature. This implies that energy stored in the subsurface effectively has a huge

160 power law memory. This contrasts with the usual phenomenological assumption used notably in box models (including zero dimensional global energy balance models) that the order of derivative is an integer (one) and that on the contrary, the memory is only exponential (short). The result followed directly by assuming that the continuum mechanics heat equation was obeyed and the depth of the media was of the order of a few diffusion depths, for the Earth, perhaps several hundred meters. The basic result was a classical application of the heat equation barely going beyond results that [Brunt, 1932] already found "in any textbook".

165

A consequence was that although Newton's law of cooling is obeyed, that the temperature obeyed the half-order energy balance equation (HEBE) rather than the phenomenological first order Energy balance Equation (EBE). When applied to the Earth, the HEBE and its implied long memory explains the success of both climate projections through to 2100 [Hebert, 2017], [Lovejoy et al., 2017], [Hébert et al., 2020] and macroweather (monthly, seasonal) temperature forecasts [Lovejoy et

170 al., 2015], [Del Rio Amador and Lovejoy, 2019]. [Del Rio Amador and Lovejoy, 2020a; Del Rio Amador and Lovejoy, 2020b]. We also considered the responses to periodic forcings showing that surface heat fluxes and temperatures are related by a complex thermal impedance ( $Z(\omega), \omega$  is the frequency). In the Earth system,  $Z(\omega) = \mathcal{F}(\omega)$  where  $\mathcal{F}(\omega)$  is the complex climate sensitivity that we estimated from a simple semi-empirical model.

Although in part 1 we discussed the classical 1-D application of the heat equation to the Earth's latitudinal energy balance (Budyko-Sellers models) - especially their ad hoc treatment of the surface boundary condition - we restricted the discussion 175

8

to zero horizontal dimensions. In this part II, we first (section 2) extend the part I treatment to horizontally systems with homogeneous properties but with inhomogeneous forcings, first in the horizontal plane (section 2.1, 2.2), then - following Budyko-Sellers - latitudinally varying on the sphere (section 2.3). systems but with inhomogeneous forcings, we then consider the more realistic case of horizontally inhomogeneous media. The homogeneous case is quite classical and can be treated with standard Laplace and Fourier techniques, it leads to the (horizontally) Generalized HEBE: the GHEBE. Although the GHEBE has a more complex (space-time) fractional derivative operator that is unlike anything we know of in the literature, - like the HEBE, it can nevertheless be given precise meaning via its Green's function.

In section 3, we derive the inhomogeneous GHEBE and HEBE needed for applications. This is done by using of Babenko's method [Babenko, 1986] which is essentially a generalization of the Laplace and Fourier transform techniques. 185 The challenge with Babenko's method is to interpret the inhomogeneous space-time fractional operators. Following Babenko, we do this using both high and low frequency expansions corresponding respectively to processes dominated by storage and by horizontal heat transport. The long time limit describes the new energy balance climate state that results when the forcing is increased everywhere and held fixed: for the model this corresponds to equilibrium. We also include several appendices focused on empirical parameter estimates (appendix A), the implications for two point and space-time temperature statistics (when the system is stochastically forced, internal variability, appendixees B, C), and finally (appendix Dappendix C), the

190

180

**2. The two-dimensional homogeneous heat equation**

**2.1 The homogeneous GHEBE**

In part I we recalled the heat equation for the time-varying temperature anomalies (T) with diffusive and (horizontal) effective 195 advective velocity (v):

changes needed to account for the Earth's spherical geometry, including the definition of fractional operators on the sphere.

$$\left(\frac{\partial}{\partial t} - \kappa_{\nu} \frac{\partial^2}{\partial z^2}\right) T = -\underline{\nu} \cdot \nabla_h T + \kappa_h \nabla_h^2 T \tag{1}$$

(This is written in the still general form of eq. 19, part I).  $\kappa_h, \kappa_\nu$  are horizontal and vertical thermal diffusivities, z the vertical coordinate (pointing upwards, the Earth is  $z \le 0$ ), t the time,  $\underline{x} = (x, y)$  the horizontal coordinates,  $\nabla_{t_{a}} = \hat{x} \partial / \partial x + \hat{y} \partial / \partial y$  (the circonflexes indicate unit vectors). These equations must now be solved using the conductive-radiative surface boundary 200 condition:

$$\left(\frac{T(\underline{x},z,t)}{s} + \rho c \kappa_{v} \frac{\partial T(\underline{x},z,t)}{\partial z}\right)_{z=0} = F(\underline{x},t)$$
(2)

 $\rho$ , *c* are the fluid densities and specific heats  $x \to -is$  the climate sensitivity and *F* is the anomaly forcing. The initial conditions are T = 0 at  $z = -\infty$  (all *t*), and  $T(\underline{x}, z, t = 0) = 0$  (Riemann-Liouville) or below,  $T(\underline{x}, z, t = -\infty) = 0$  (Weyl).

205 In part I, we nondimensionalized the zero-dimensional homogeneous operators by nondimensionalizing time by the relaxation time:  $t \rightarrow t/\tau$  (with  $\tau = \kappa_v (\rho cs)^2$ ) and nondimensionalizing the vertical distance by the vertical diffusion depth:  $z \rightarrow z/l_v$ , with  $l_v = (\tau \kappa_v)^{1/2}$ . Considering now the full equation with advective and diffusive transport, we nondimensionalize the horizontal coordinates by the horizontal diffusion length:  $\underline{x} \rightarrow \underline{x}/l_h$ , (with  $l_h = (\tau \kappa_h)^{1/2}$ ) and use the nondimensional advection velocity  $\underline{\alpha} = \frac{\underline{v}}{V}$  (with speed  $V = \frac{l_h}{\tau}$ ). If we now take  $\underline{s} \neq 1$  (equivalent to using dimensions 210 of temperature for the forcing *F*), we obtain:

$$\left(\frac{\partial^2}{\partial z^2} - \left(\frac{\partial}{\partial t} + \left(-\nabla_h^2\right) - \underline{\alpha} \cdot \nabla_h\right)\right) T = 0$$

$$\frac{\partial T}{\partial z}\Big|_{z=0} + T\left(t, \underline{x}; 0\right) = F\left(t, \underline{x}\right)$$
(3)

For the heat equation and the conductive-radiative surface boundary condition respectively. For initial conditions such that T = 0 for  $t \le 0$ , as in part I, we take Laplace transforms in time, but we now take Fourier transforms in the horizontal:

215
$$\left(\frac{\partial^2}{\partial z^2} - \left(\frac{\partial}{\partial t} + \left(-\nabla_h^2\right) - \underline{\alpha} \cdot \nabla_h\right)\right) T = 0 \stackrel{L.T.(I), F.T.(\underline{x})}{\longleftrightarrow} \left(\frac{d^2}{dz^2} - \left(p + k^2 - i\underline{\alpha} \cdot \underline{k}\right)\right) \hat{T} = 0$$
(4)

Where "F.T." is the Fourier transform in horizontal space,  $\underline{k}$  for the conjugate of  $\underline{x}$ ,  $k = |\underline{k}|$  (the vector modulus) with conjugate variable  $r = |\underline{x}|$  (as usual,  $\nabla_h \stackrel{F.T.}{\leftrightarrow} i\underline{k}$ ). Fourier transforms in space are convenient for either infinite horizontal media, or media with periodic horizontal boundary conditions. In appendix Dappendix C, we consider the changes needed to account for spherical geometry.

220 When  $F(t,\underline{x}) = \delta(t)\delta(\underline{x})$ , the solution  $T(t,\underline{x}) \to G_{\delta}(t,\underline{x})$  and  $\hat{T}(p,\underline{k}) \to \hat{G}_{\delta}(p,\underline{k})$  where  $G_{\delta}$  is the impulse (Dirac) response Green's function, part I, eq. 30. From eq. 4, we see that this is the same as the zero dimensional equation (eq. 24, part I) but with  $p \to p + k^2 - i\underline{\alpha} \cdot \underline{k}$  i.e. for the corresponding Green's function: Formatted: Font: Italic

$$\widehat{G_{\delta}}(p,k;z) = \widehat{G_{\delta}}(p+k^2-i\underline{\alpha}\cdot\underline{k};z)$$

A note on notation: the first argument is time, with the vertical separated by a semi-colon. When there is a horizontal coordinate it comes after time, before the semicolon. With this notation, the right hand side of eq. 5 is the L.T. of the zero-dimensional (time-depth) Green's function  $G_{\delta}(t;z)$ , the left hand side is the Laplace (time) and Fourier transform (horizontal, space) transform.

(5)

We can now use the basic Laplace shift property:

$$e^{\left(-k^{2}+i\underline{\alpha}\cdot\underline{k}\right)^{L,T,(t)}}G_{\delta}\left(t;z\right) \stackrel{L,T,(t)}{\longleftrightarrow} \widehat{G_{\delta}}\left(p+k^{2}-i\underline{\alpha}\cdot\underline{k};z\right)$$

$$\tag{6}$$

230 To conclude that:

$$\hat{G}_{\delta}(t,\underline{k};z) = e^{\left(-k^2 + i\underline{\alpha};\underline{k}\right)^{t}} G_{\delta}(t;z)$$
(7)

Decomposing this into a circularly symmetric diffusion part  $\widehat{G}_{\delta,dif}(t,k;z)$  and a factor  $e^{i\underline{k}\cdot \alpha t}$  that shifts phases, we obtain:

$$\hat{G}_{\delta}(t,\underline{k};z) = e^{i\underline{k}\cdot\underline{\alpha}t}\hat{G}_{\delta,dif}(t,k;z); \qquad \hat{G}_{\delta,dif}(t,k;z) = e^{-k^{2}t}G_{\delta}(t;z)$$

$$\tag{8}$$

By circular symmetry of  $\hat{G}_{\delta,dif}(t,k;z)$ , its inverse (2-D) Fourier transform reduces to an inverse Hankel transform ("H.T."). 235 Using:

$$\frac{e^{-r^2/(4t)}}{2t} \stackrel{H.T.}{\leftrightarrow} e^{-k^2t} \tag{9}$$

We therefore obtain for the diffusive part of the surface impulse response (i.e. the response with source spatial forcing  $\delta(\underline{x}) = \delta(r)/(2\pi r)$ ):

$$G_{\delta,dif}(t,r;z) = \frac{e^{-r^2/(4t)}}{2t} G_{\delta}(t;z)$$
(10)

240 Where  $G_{\delta}(t;z)$  is the zero-dimensional impulse response. If needed, its integral representation is given in eq. 3034, part I. The last step is to take into account the advective term associated with the phase shift  $\underline{k} \cdot \underline{\alpha}t$ . For this final step, we use the Fourier shift theorem to obtain:

$$G_{\delta}(t,\underline{x};z) = G_{\delta,dif}(t,|\underline{x}-\underline{\alpha}t|;z) = \frac{e^{-|\underline{x}-\underline{\alpha}t|^{2}/(4t)}}{2t}G_{\delta}(t;z)$$
(11)

This is the general surface result for the diffusive-advective transport part of the spatially homogeneous case. As 245 expected, the advective transport simply displaces the center centre of the impulse response with nondimensional velocity  $\underline{\alpha}$ . As usual, the solutions for arbitrary forcing F(t,x) can be obtained by convolution.

For the surface we obtain the simpler expressions:

$$G_{\delta,dif}(t,r;0) = \frac{e^{-r^{2}/(4t)}}{2t} \left( \frac{1}{\sqrt{\pi t}} - e^{t} erfc \sqrt{t} \right)$$

$$G_{\Theta,dif}(t,r;0) = \int_{0}^{t} G_{\delta,dif}(t,r;0) dt = \frac{1}{r} erfc \left( \frac{r}{2\sqrt{t}} \right) - \int_{0}^{t} \frac{e^{\frac{r^{2}}{4s}+s}}{2s} erfc(s^{1/2}) ds$$
(12)

 $\begin{vmatrix} 250 & \text{(see eq. } \underline{3435}\text{, part I).} \text{ From these, the general surface results including advection are obtained with } r \rightarrow |\underline{x} - \underline{\alpha}t| \text{, i.e.} \\ G_{\delta}(t,\underline{x};0) = G_{\delta,dif}(t,|\underline{x} - \underline{\alpha}t|;0). \end{aligned}$

Since the advection term has this simple consequence, below we take  $\underline{\alpha} = 0$ , considering only diffusive transport, advection can easily be included if needed (i.e. below, we take  $G_{\delta}(t,r;0) = G_{\delta,dif}(t,r;0)$ ).

To better understand the impulse response, fig. 1 shows this surface  $G_{\delta}(t,r;0)$  for various radial distances *r* and fig. 2 shows the corresponding time dependence of the time integral of  $G_{\delta}$ ; the unit step response  $G_{\Theta}$  for various distances *r*, illustrating the power law approach to thermodynamic equilibrium at large *t* (discussed in section 2.2). The corresponding long time, short distance expansions are:

$$G_{\delta}(t,r;0) \approx \frac{t^{-5/2}}{4\sqrt{\pi}} - \frac{(6+r^2)}{16\sqrt{\pi}} t^{-7/2} + O(t^{-9/2}) \qquad \qquad t > 1$$

$$G_{\Theta}(t,r;0) \approx G_{therm,\delta}(r;0) - \frac{t^{-3/2}}{6\sqrt{\pi}} + \frac{(6+r^2)}{40\sqrt{\pi}}t^{-5/2} + O(t^{-7/2})$$

$$r \ll 1$$

260 Where  $G_{therm,\delta}(r,0)$  is the Green's function for the (spatial Dirac) "hotspot" thermodynamic\_equilibrium response discussed below (eq. 20). Note that the leading term in  $G_{\delta}(t,r;0)$  is independent of r, and the leading term in the approach to thermodynamic\_equilibrium  $G_{\Theta}(t,r;0)$  is also independent of r.

Just as we derived the zero-dimensional HEBE by showing that it had the same Green's function as the z = 0 transport equation Green's function, we can likewise derive the homogeneous Generalized Half-Order Energy Balance Equation (GHEBE) which is the space-time surface equation whose Green's function is given in eq. 12. Following the derivation of the HEBE, in part I

eq. 29, and replacing  $p \rightarrow p + k^2 - i\underline{\alpha} \cdot \underline{k}$  we obtain:

$$\hat{G}_{\delta}(p,\underline{k};z) = \frac{e^{\sqrt{p+k^2 - i\underline{\alpha}\cdot\underline{k}z}}}{\sqrt{p+k^2 - i\underline{\alpha}\cdot\underline{k}} + 1}$$
(14)

Hence, for z = 0:

265

$$\left[ \left( \frac{\partial}{\partial t} + \left( -\nabla_{h}^{2} \right) - i\underline{\alpha} \cdot \nabla_{h} \right)^{1/2} + 1 \right] G_{\delta}(t,\underline{x};0) = \delta(t) \delta(\underline{x}) \overset{(L.T.(t),F.T.(\underline{x}))}{\longleftrightarrow} \left( \sqrt{p + k^{2} - i\underline{\alpha} \cdot \underline{k}} + 1 \right) \hat{G}_{\delta}(p,\underline{k};0) = 1 (15)$$

270 The left hand equation is the homogeneous GHEBE whose Green's function is given by eq. 12. We have therefore found a surprisingly simple explicit formula for the (inverse) half-order space-time GHEBE operator:

$$\left[ \left( \frac{\partial}{\partial t} + \left( -\nabla_h^2 \right) - i\underline{\alpha} \cdot \nabla_h \right)^{1/2} + 1 \right]^{-1} = G_\delta(t, \underline{x}; 0) *$$
(16)

where "\*" indicates convolution. This allows us to give a precise interpretation of the half-order operator. Therefore the dimensional, homogeneous, GHEBE and its full solution are:

$$\begin{pmatrix} \tau \frac{\partial}{\partial t} + \left(-l_{h}^{2} \nabla_{h}^{2}\right) - il_{h} \underline{\alpha} \cdot \nabla_{h} \end{pmatrix}^{1/2} T_{s}\left(t, \underline{x}\right) + T_{s}\left(t, \underline{x}\right) = sF\left(t, \underline{x}\right)$$

$$\quad T_{s}\left(t, \underline{x}\right) = s \int_{surf} \int_{0}^{t} G_{\delta}\left(\frac{t-t'}{\tau}, \frac{|\underline{x}-\underline{x}'|}{l_{h}}; 0\right) F\left(t', \underline{x'}\right) \frac{dt'}{\tau} \frac{d\underline{x'}}{l_{h}^{2}}$$

$$= \frac{s}{l_{h}^{2}} \int_{surf} \int_{0}^{t} \frac{e^{-t|\underline{x}-\underline{x'}-l_{h}\underline{\alpha}(t-t')|t|^{2}/(4l_{h}^{2}(t-t'))}}{2\left(t-t'\right)} \left(\sqrt{\frac{\tau}{\pi\left(t-t'\right)}} - e^{(t-t')^{2}} erfc\sqrt{\frac{\left(t-t'\right)}{\tau}}\right) F\left(t', \underline{x'}\right) dt' d\underline{x'}$$

$$(17)$$

13

Field Code Changed

("surf" is the surface over which the forcing acts, the bottom line uses the explicit eq. 12 for  $G_{\delta}$ ).

The above shows that even with the purely classical integer-ordered Budyko-Sellers type heat equation, that surface temperatures already obey long memory, half order equations. However, it is not certain that the classical heat equation is in

fact the most appropriate model. Straightforward generalizations to fractional heat equations - where  $\tau \frac{\partial T}{\partial t} \rightarrow \tau^{2H} {}_{\infty} D_t^{2H} T$

280 lead directly to fractional energy balance equations for surface temperatures, we investigate fractional heat equations elsewhere. Physically, this generalization from the classical fractional value H = 1/2 could be a consequence of turbulent diffusive transport which since at least Richardson been known to have anomalous diffusion.

**2.2 Energy balance, Thermodynamic equilibrium**

| 285 | If $F(t,x) = 0$ then there is a radiative energy balance at time t, point x, but the temperature may be changing. However, iff                                                                                                                       |
|-----|------------------------------------------------------------------------------------------------------------------------------------------------------------------------------------------------------------------------------------------------------|
|     | <math>F(t,x) = 0</math> for a long enough time, and for all <math>xF(t) = 0</math>, then the time derivatives <math>(\frac{\partial}{\partial t} = 0)</math> vanish and Earth is in a steady                                                  |
|     | energy balance ("climate") state, $T_{clim}(x)$ , so that the temperature anomaly $T(t,x) = 0$ . Now consider a step function increase                                                                                                               |
|     | $\underline{F(t,\underline{x}) = \Theta(t)F_0(\underline{x})}$ . Then as $\underline{t \to \infty}$ , the time derivatives will vanish and a new (steady) climate state (with temperature)                                                           |
|     | $T_0(\underline{x})$ ) will be reached in which the horizontal transport and anomalous black body emission balance the new forcing:                                                                                                                  |
| 290 | $\frac{\left(\left(-\nabla_{h}^{2}\right)^{1/2}+1\right)T_{0}\left(\underline{x}\right)=F_{0}\left(\underline{x}\right)}{F_{0}\left(\underline{x}\right)}$ . The new state is steady in time and is in energy balance with outer space and its local |
|     | surroundings, but it is not strictly correct to describe $T_0(\underline{x})$ as one of thermal equilibrium. This is because thermal                                                                                                                 |
|     | equilibrium would imply that the temperature everywhere is constant (thermodynamic equilibrium is an even more stringent                                                                                                                             |
|     | condition). Nevertheless the term "radiative equilibrium" is commonly used in the context of planetary energy balance, so                                                                                                                            |
|     | we will use the terms energy balance and equilibrium synonymously.                                                                                                                                                                                   |
| 295 | Let us now investigate the equilibrium state. Since, then the system is at equilibrium and will stay there. However, if F is a                                                                                                                       |

step function in time, then as  $t \rightarrow \infty$ , a new equilibrium will be established. At equilibrium, d/dt = 0, so that the conjugate variable  $p = 0_{\underline{s}}$ . With this and  $\underline{\alpha} = 0$  in eq. 15, we obtain the equation for the (spatial) surface impulse response  $\underline{G}_{eq,\delta}(r;0)$  for thermodynamic equilibrium (subscript "thermeq"):

$$\left(\left(-\nabla_{h}^{2}\right)^{1/2}+1\right)G_{eq,\delta}=\delta(\underline{x})\overset{F.T.}{\longleftrightarrow}(k+1)\hat{G}_{eq,\delta}=1$$

14

| Formatted: Font: Italic            |
|------------------------------------|
| Formatted: Font: Italic            |
| Formatted: Font: Italic, Underline |
| Formatted: Font: Italic            |
| Formatted: Font: Italic, Underline |
| Formatted: Font: Italic, Underline |
| Formatted: Font: Italic            |
| Formatted: Font: Italic, Subscript |
| Formatted: Font: Italic, Underline |
| Formatted: Font: Italic            |
| Formatted: Font: Italic            |
| Formatted: Font: Italic, Underline |

**Field Code Changed**

(18)

300 i.e. the same as eq. 4 but with p = 0 (and  $\alpha = 0$ ) hence:

$$\frac{\hat{G}_{eq,\delta}(k;z) = \frac{e^{kz}}{1+k}}{(19)}$$

The equilibrium surface temperature (spatial) impulse (Dirac "hotspot") Green's function is therefore:

$$G_{eq,\delta}(r,0) = \frac{1}{r} + \frac{\pi}{2} \left( Y_0(r) - H_0(r) \right) \stackrel{(H.T.)}{\leftrightarrow} \hat{G}_{eq,\delta}(k;0) = \frac{1}{1+k}$$
(20)

Where  $H_0$  is the zeroth order Struve function and  $Y_0$  is the zeroth order Bessel function of the second kind. For large r, we 305 have the expansions:

$$G_{eq,\delta}(r;0) \approx \frac{1}{r^3} - \frac{9}{r^5} + O(r^{-7}); \quad r >> 0$$
(21)

$$\frac{G_{eq,\delta}(r;0) \approx \frac{1}{r} + \log r + \gamma_E - \log 2 - r + \frac{r^2}{4} (1 + \log 2 - \gamma_E) - \frac{r^2}{4} \log r + \dots;}{r \approx 1 + \log r + \gamma_E - \log 2 - r + \frac{r^2}{4} (1 + \log 2 - \gamma_E) - \frac{r^2}{4} \log r + \dots;}$$

The  $1/r^3$  asymptotic decay is fast and implies that spatial hotspots remain fairly localized; indeed, it is easy to show that if

0

310 instead we had a Dirac surface heat flux source driving the system (i.e. with surface BC  $\frac{\partial T}{\partial z}\Big|_{z=0} = \delta(\underline{x})$  i.e. without radiation)

that the decay would be the much faster (1/r). Forcing inhomogeneities thus remain much more localized than would otherwise be the case.

To study the convergence to thermodynamic equilibrium, consider a simple model of a surface "hot spot" where the forcing is confined to a unit circle and turned on and held at a constant unit temperature at t = 0. This is the spatial equivalent of a step forcing in space, we combine it with a step (Heaviside) in time:

$$F(t,r) = \Theta(t)\Pi_{1}(r); \quad \Pi_{1}(r) = \begin{array}{cc} 1 & r \le 1 \\ 0 & r > 1 \end{array}$$
(22)

 $\Pi_1(\mathbf{r})$  is the corresponding indicator function. We now use the transform pair  $\Pi_1(\mathbf{r}) \stackrel{H.T.}{\leftrightarrow} \frac{J_1(k)}{k}$  to perform the convolution:

$$T_{s}(t,r) = G_{\Theta}(t,r;0) * \Theta(t) \Pi_{1}(r) \stackrel{H.T.}{\leftrightarrow} \frac{J_{1}(k)}{k} \hat{G}_{\Theta}(t,k;0)$$
(23)

( $J_1$  is the first order Bessel function of first kind). Taking the limit  $t \rightarrow \infty$  we obtain the thermodynamic-\_equilibrium 320 temperature distribution. Alternatively we could find it directly by from eq. 19:

$$T_{eq,s}(r) = T_s(\infty, r) \stackrel{H.T.}{\longleftrightarrow} \frac{J_1(k)}{k(1+k)}$$
(24)

Fig. 4 shows the cross section as a function of the distance from the circle's center at various times (the inverse Hankel transforms were done numerically). We note that the temperature rises very quickly at first, then slowly reaches equilibrium (thick). The figure also shows (dashed) the thermodynamic\_\_equilibrium when the forcing is purely due to unit conductive heating over the unit circle. The difference between the dashed and the thick thermodynamic\_\_equilibrium curves are purely due to the radiative loses in the latter. (Note that in the zero-dimensional case (part I), using pure heating forcing boundary conditions leads to diverging temperatures, there is no-thermodynamic\_\_equilibrium. This explains why Brunt instead used temperature forcing boundary conditions. Here, in two horizontal dimensions, boundary conditions that impose a fixed temperature over the circle are problematic since they imply infinite horizontal temperature gradients and infinite horizontal 330 heat fluxes).

Figs. 5, 6 shows the same evolution but with temperature as a function of time for various distances (fig. 5) and as contours in space-time (fig. 6). We see that equilibrium is largely established in the first two relaxation times (here  $\tau = 1$ ) and most of the perturbation is confined to two horizontal diffusion distances (here  $l_h = 1$ ).

335 **2,3** Comparison of the HEBE with the standard 1-D Budyko - Sellers model on a sphere It is helpful to clearly understand the similarities and differences between the HEBE and the usual 1-D (latitudinal) B-S approach (see the comprehensive monograph [North and Kim, 2017], and see [Zhuang et al., 2017], [Ziegler and Rehfeld, 2020] for recent applications, development). Since the latter model is on a sphere but with only latitudinal dependence, we write the horizontal transport term  $\nabla_h \cdot D_{R-S} \nabla_h$  using gradient and divergence operators:

340
$$\nabla_h = -\frac{1}{R} \frac{d}{d\mu} \sqrt{1-\mu^2}; \quad \nabla_h = -\frac{\sqrt{1-\mu^2}}{R} \frac{d}{d\mu}$$
 with  $\theta$  = colatitude and  $\mu = \cos \theta$ . In standard notation [North and

Field Code Changed Formatted: Font: Cambria

Kim, 2017]) the B-S equation is thus written:

$$\frac{c \frac{\partial T}{\partial t} - \frac{\partial}{\partial \mu} \left( D_{\mu,\nu}(\mu) (1-\mu^{2}) \frac{\partial}{\partial \mu} T \right) + \beta(\mu)T + A(\mu) = Q_{\mu}H(\mu); \quad H(\mu) = S(\mu)a(\mu)$$
(25)
(Formatted: Equation
(Formatted: Fort: Edit
(Formatted:

$$\begin{array}{c|c} & T(t,\mu) = \sum_{n=0}^{\infty} T_n(t) P_n(\mu) \stackrel{t.r.}{\leftrightarrow} \hat{T}(p,\mu) = \sum_{n=0}^{\infty} \hat{T}_n(p) P_n(\mu) \\ & F(t,\mu) = \sum_{n=0}^{\infty} F_n(t) P_n(\mu) \stackrel{t.r.}{\leftrightarrow} \hat{T}(p,\mu) = \sum_{n=0}^{\infty} \hat{F}_n(p) P_n(\mu) \\ & F(t,\mu) = \sum_{n=0}^{\infty} F_n(t) P_n(\mu) \stackrel{t.r.}{\leftrightarrow} \hat{T}(p,\mu) = \sum_{n=0}^{\infty} \hat{F}_n(p) P_n(\mu) \\ & (29) \\ & F(t,\mu) = \sum_{n=0}^{\infty} F_n(t) P_n(\mu) \stackrel{t.r.}{\leftrightarrow} \hat{T}(p,\mu) = \sum_{n=0}^{\infty} \hat{F}_n(p) P_n(\mu) \\ & (29) \\ & (29) \\ & F(t,\mu) = \sum_{n=0}^{\infty} F_n(t) P_n(\mu) \stackrel{t.r.}{\leftrightarrow} \hat{T}(p,\mu) = \sum_{n=0}^{\infty} \hat{F}_n(p) P_n(\mu) \\ & (29) \\ & (10) \\ & (10) \\ & (10) \\ & (10) \\ & (10) \\ & (10) \\ & (10) \\ & (10) \\ & (10) \\ & (10) \\ & (10) \\ & (10) \\ & (10) \\ & (10) \\ & (10) \\ & (10) \\ & (10) \\ & (10) \\ & (10) \\ & (10) \\ & (10) \\ & (10) \\ & (10) \\ & (10) \\ & (10) \\ & (10) \\ & (10) \\ & (10) \\ & (10) \\ & (10) \\ & (10) \\ & (10) \\ & (10) \\ & (10) \\ & (10) \\ & (10) \\ & (10) \\ & (10) \\ & (10) \\ & (10) \\ & (10) \\ & (10) \\ & (10) \\ & (10) \\ & (10) \\ & (10) \\ & (10) \\ & (10) \\ & (10) \\ & (10) \\ & (10) \\ & (10) \\ & (10) \\ & (10) \\ & (10) \\ & (10) \\ & (10) \\ & (10) \\ & (10) \\ & (10) \\ & (10) \\ & (10) \\ & (10) \\ & (10) \\ & (10) \\ & (10) \\ & (10) \\ & (10) \\ & (10) \\ & (10) \\ & (10) \\ & (10) \\ & (10) \\ & (10) \\ & (10) \\ & (10) \\ & (10) \\ & (10) \\ & (10) \\ & (10) \\ & (10) \\ & (10) \\ & (10) \\ & (10) \\ & (10) \\ & (10) \\ & (10) \\ & (10) \\ & (10) \\ & (10) \\ & (10) \\ & (10) \\ & (10) \\ & (10) \\ & (10) \\ & (10) \\ & (10) \\ & (10) \\ & (10) \\ & (10) \\ & (10) \\ & (10) \\ & (10) \\ & (10) \\ & (10) \\ & (10) \\ & (10) \\ & (10) \\ & (10) \\ & (10) \\ & (10) \\ & (10) \\ & (10) \\ & (10) \\ & (10) \\ & (10) \\ & (10) \\ & (10) \\ & (10) \\ & (10) \\ & (10) \\ & (10) \\ & (10) \\ & (10) \\ & (10) \\ & (10) \\ & (10) \\ & (10) \\ & (10) \\ & (10) \\ & (10) \\ & (10) \\ & (10) \\ & (10) \\ & (10) \\ & (10) \\ & (10) \\ & (10) \\ & (10) \\ & (10) \\ & (10) \\ & (10) \\ & (10) \\ & (10) \\ & (10) \\ & (10) \\ & (10) \\ & (10) \\ & (10) \\ & (10) \\ & (10) \\ & (10) \\ & (10) \\ & (10) \\ & (10) \\ & (10) \\ & (10) \\ & (10) \\ & (10) \\ & (10) \\ & (10) \\ & (10) \\ & (10) \\ & (10) \\ & (10) \\ & (10) \\ & (10) \\ & (10) \\ & (10) \\ & (10) \\ & (10) \\ & (10)$$

$$e^{-t} \stackrel{L.T.}{\longleftrightarrow} \frac{1}{1+p}$$
$$\sqrt{\frac{1}{\pi t}} - e^{t/\tau} \operatorname{erfc} \sqrt{t} \stackrel{L.T.}{\longleftrightarrow} \frac{1}{1+p^{1/2}}$$

375

(eq. 35, part I), combining this with eq. 32, we obtain for the impulse responses:

$$G_{\delta,B-S}^{(n)}(t) = \tau^{-1} e^{-(1+\xi_{B-S,n})t/\tau}$$
$$G_{\delta,F}^{(n)}(t) = \tau^{-1} e^{-\xi_{F,n}t/\tau} \left(\sqrt{\frac{\tau}{\pi t}} - e^{t/\tau} erfc\sqrt{\frac{t}{\tau}}\right)$$

Integrating these with respect to *t*, we obtain the step responses:

$$G_{\Theta,B-S}^{(n)}(t) = \frac{1}{\xi_{B-S,n} + 1} \left( 1 - e^{-(\xi_{B-S,n} + 1)/\tau} \right)$$

$$G_{\Theta,F}^{(n)}(t) = \frac{\sqrt{\xi_{F,n}} erf \sqrt{\xi_{F,n}} \frac{t}{\tau} - 1 + e^{-t(\xi_{F,n} - 1)/\tau} erfc \sqrt{\frac{t}{\tau}}}{\xi_{F,n} - 1}$$
(35)

The long time limit represents Earth energy balance (equilibrium):

$$\begin{aligned} G_{eq,B-S}^{(n)} &= G_{\Theta,F}^{(n)}(\infty) = \frac{1}{1 + \xi_{B-S,n}} = \frac{1}{1 + sD_{B-S}n(n+1)} \\ G_{eq,F}^{(n)} &= G_{\Theta,F}^{(n)}(\infty) = \frac{1}{1 + \sqrt{\xi_{F,n}}} = \frac{1}{1 + \sqrt{sD_Fn(n+1)}} \end{aligned} ; \quad \xi \ge 0 \end{aligned}$$

If ξ<0, then there is an unphysical divergence so that sDF must be>0. Since Pa(μ) has *p* zeroes, *p* plays the role of wavenumber, *j* it specifies structures of horizontal size ≈ πR/μ. Therefore we see that the B-S model (where G falls off as p2) will yield a *p* much smoother equilibrium temperature than the HEBE where it falls off as p1. Note that when generalized from the HEBE to the FEBE (with p→p24), this equilibrium result is unchanged. For the HEBE, the short and long time behaviours are:

|     | 2 N                |                                      |
|-----|--------------------|--------------------------------------|
|     |                    |                                      |
|     |                    |                                      |
|     |                    |                                      |
|     |                    |                                      |
|     |                    |                                      |
|     |                    |                                      |
|     |                    |                                      |
|     |                    | Formatted: Equation                  |
|     |                    |                                      |
| *   |                    |                                      |
|     |                    |                                      |
|     |                    |                                      |
|     |                    |                                      |
|     |                    | Formatted: Font: Italic              |
|     | Å                  | Formatted: Equation                  |
|     | - / (              | Field Code Changed                   |
|     | //                 |                                      |
| -   | / (                | Formatted: Equation                  |
|     | / /                | Field Code Changed                   |
| 1   | / //(              | Formatted: Font: Symbol              |
| /   | - ///              | Formatted: Font: Italic              |
|     | -////              | Formatted: Font: Italic              |
|     | (                  | Formatted: Font: Italic, Subscript   |
|     | /////              | Formatted: Font: Symbol              |
| 1   |                    | Formatted: Font: Italic              |
| 1   | /////              | Formatted: Font: Italic              |
| *   |                    | Formatted: Font: Symbol              |
|     |                    | Formatted: Font: Italic              |
|     |                    | Formatted: Font: Italic              |
| i   |                    | Formatted: Font: Italic              |
| .]  |                    | Formatted: Font: Italic              |
| a 🖉 | $\mathbb{P}^{(1)}$ | Formatted: Superscript               |
| R   |                    | Formatted: Font: Italic              |
| #   | ·(                 | Formatted: Superscript               |
| ~   | $\leq$             | Formatted: Font: Italic              |
|     | $M_{2}$            | Formatted: Font: Italic              |
|     | _/(                | Formatted: Superscript               |
|     | X                  | Formatted: Font: Italic, Superscript |
|     |                    |                                      |

(33)

(34)

(36)

$$G_{\Theta,F}^{(n)}(t) = \frac{2t^{1/2}}{\sqrt{\pi\tau}} - \frac{t}{\tau} - \frac{2(\xi_{F,n} - 2)}{3\sqrt{\pi}} \left(\frac{t}{\tau}\right)^{3/2} + \frac{1}{2}(\xi_{F,n} - 1)\left(\frac{t}{\tau}\right)^2 + ...; \quad t <<\tau; \quad n \ge 0$$

$$G_{\Theta,F}^{(0)}(t) = 1 - \frac{1}{\sqrt{\pi t}} + \frac{1}{2t\sqrt{\pi t}} - ...; \quad t >>\tau; \quad n = 0 \quad (37)$$

$$G_{\Theta,F}^{(n)}(t) = \frac{1}{1 + \sqrt{\xi_{F,n}}} - \frac{e^{-\xi_{F,n}^{1/\tau}}}{2\sqrt{\pi}\xi_{F,n}} \left(\frac{t}{\tau}\right)^{-3/2} \left(1 - \frac{3}{2}\left(\frac{1 + \xi_{F,n}}{\xi_{F,n}t/\tau}\right) + ...\right); \quad t >>\tau; \quad n \ge 1$$

The asymptotic response for  $G_{\Theta,F}^{(n)}(t)$  is interesting because it tells us how quickly equilibrium is reached. When n = 0 we 385 have  $P_0(\mu) = 1$ , so that this component corresponds to the mean. Since  $\xi_{F,0} = 0$  we see that it is identical to the zerodimensional result in part I: equilibrium is approach in a power law fashion ( $f_{12}^{1/2}$  for large t), whereas for  $\mu = 0$ , the B-S model approach to equilibrium is exponential. However for  $p \ge 1$ , HEBE power law terms are exponentially damped, with exponential decay time:  $\tau_{F,n} = \tau / \xi_{F,n}$  whereas the B-S model is exponentially damped for all *n* with  $\tau_{B-S,n} = \tau / (1 + \xi_{B-S,n})$ . In order to make a more detailed comparison between the models, we can follow [North and Kim, 2017] who consider a model 390 with constant DS-B and that is north-south symmetric so that the odd numbered polynomials vanish. They empirically give the climate equilibrium values for n = 0, 2, 4; the (constant) p = 0 term is used to obtain the mean temperature 288K. Other pertinent empirical data are  $s = 1/B = 0.50 \text{ KW}^{-1}\text{m}^2$ ,  $F_2 = -180.7 \text{ W/m}^2$ ,  $F_4 = 20.8\text{K}$ ,  $T_2 = -30\text{K}$ ,  $T_4 = -4\text{K}$ . From eq. 36 for the equilibrium temperature Green's function, we obtain:  $T_{eq,n} = sG_{eq,B-S}^{(n)}F_n$ . The n = 2 relationship is use to estimate  $D_{B-S} = \frac{1}{6s} \left( \frac{sF_2}{T_2} - 1 \right) = 0.67 \text{ Wm}^2 \text{K}^{-1}, \text{ with this estimate, we obtain} \underline{T_4} = F_4 / \left( 1 + \xi_{B-S,n} \right) = F_4 / \left( 1 + 20D_{B-S} \right) \approx \frac{1}{2} \left( 1 + \frac{1}{2} \right) \left( 1 + \frac{1}{2}$ 1.35K which is not far from the empirical estimate  $T_4 = -4K$  ([North and Kim, 2017]), it also yields the dimensionless quantity 395 sDB-S = 0.33. If we follow the same procedure for the HEBE, we estimate  $D_F = \frac{1}{6s} \left(\frac{sF_2}{T_2} - 1\right)^2$ , comparing this with the B-S relation, we find:  $\mathcal{S}D_F = 6(\mathcal{S}D_{S,B})^2$  the dimensionless  $\mathcal{S}D_F = 0.67$ , and  $\mathcal{D}_F = 1.33$  Wm2K-1,  $T_4 = 2.23$ K (again not far from the data). We note that the ratio  $D_F / D_{B-S} \approx 2$  so that the estimates are close.

| Formatted: Font: Italic |
|-------------------------|
| Formatted: Subscript    |
| Formatted: Font: Symbol |
| Formatted: Font: Italic |
| Formatted: Superscript  |
| Formatted: Font: Italic |
| Formatted: Font: Italic |
| Formatted: Font: Italic |
| Field Code Changed      |
| Formatted: Font: Italic |
| Formatted: Subscript    |
| Formatted: Font: Italic |
| Field Code Changed      |
| Formatted: Font: Italic |
| Field Code Changed      |
| Field Code Changed      |
| Field Code Changed      |
| Formatted: Font: Italic |

| 1   | We can use this information to estimate lh in the HEBE. From the definition of DB-S as a thermal conduction coefficient+                                                               |     | Formatted: Indent: First line: 1 cm |
|-----|----------------------------------------------------------------------------------------------------------------------------------------------------------------------------------------------------------------------------|-----|-------------------------------------|
| 400 | per radian we obtain $D_{\rm PS} = K/R$ so that $K = K / \rho_{\rm C} = RD / \rho_{\rm C} \approx 1m^2 / s$ . To find the transport length, we can use                                                                     |     | Formatted [1]                       |
| 100 | per radian we obtain $D_{B-3}$ for so that $K_h$ is per rad $B-S$ , per rad $rad S_{B-S}$ is induced an apprendix in the decamport rengal, we can appe                                                                     |     | Field Code Changed                  |
|     | $(\kappa)^{1/2}$                                                                                                                                                                                                           | 1   | Field Code Changed                  |
|     | $l_h = \beta \kappa_h \rho cs$ , $\beta = \left  \frac{\alpha_v}{\kappa} \right $ , to obtain:                                                                                                                             |     | Field Code Changed                  |
|     |                                                                                                                                                                                                                            |     |                                     |
|     | 1                                                                                                                                                                                                                          |     | Formatted: Equation                 |
|     | $\frac{h}{R} = \beta s D_{B-S} \tag{38}$                                                                                                                                                                                   |     | Field Code Changed                  |
|     |                                                                                                                                                                                                                            |     |                                     |
|     | Alternatively, we can estimate $\frac{1}{h}$ from the global scale $D_{H}$ .                                                                                                                                               |     | Formatted [2]                       |
|     | 1                                                                                                                                                                                                                          |     | Formatted: Equation                 |
|     | $\frac{c_h}{R} = sD_F$                                                                                                                                                                                                     |     | Field Code Changed                  |
|     | (39)                                                                                                                                                                                                                       |     |                                     |
| 405 | We see that these l estimates differ by a factor of $\beta D_{B_s} D_F \approx \beta/2$ . Since typical numerical models with resolutions of hundreds                                                        | -1  | Formatted [3]                       |
|     | of kilometers use $\kappa_{t} \approx 10^{-4} \text{ m}^{2}/\text{s}$ , and $\kappa_{t} \approx 1 \text{m}^{2}/\text{s}$ , at least at these scales $\beta \approx 10^{-2}$ so that the difference in the estimates may be |     |                                     |
|     | large. For example since $sD_{B_cS} \approx 0.33$ , we find that the former yields $J_k \approx 20$ km, while the latter yields, $l_h \approx 4000$ km. One                                                                | /// |                                     |
|     | way to reconcile the difference is to assume that $\beta$ - that characterizes the horizontal-vertical effective diffusivity ratio - has a                                                                                 | //  |                                     |
|     | systematic scale dependence due to a difference in the scaling properties of $\kappa_{k}$ and $\kappa_{k}$ so that at global scales $\beta \approx 1$ (this may                                                            | /   |                                     |
| 410 | arise as a consequence of the scaling anisotropic horizontal structure of the atmosphere at weather scales, notably of the                                                                                                 |     |                                     |
|     | horizontal wind field, the 23/9D model, [Schertzer and Lovejoy, 1985]).                                                                                                                                                    |     |                                     |
|     | A different (possibly additional) way of reconciling the estimates is to consider the potentially large (multifractal) intermittency                                                                                       |     |                                     |
|     | of the diffusivities that introduces s strong scale effect. For example, 0 [Havlin and Ben-Avraham, 1987], [Weissman, 1988],                                                                                               |     |                                     |
|     | [Lovejoy et al., 1998]) show that in 1-D, the large scale effective thermal resistance $\rho_T$ – the inverse diffusivity - is the average                                                                                 |     | Formatted [4]                       |
| 415 | of the small scale resistances. If we denote the spatial averages over a scale L by a subscript, and assume that the resistivity is                                                                                 | //  |                                     |
|     | scaling (scale invariant) up to planetary scales (denote this by R), then it will generally follow the following multifractal                                                                                              |     |                                     |
|     | statistics:                                                                                                                                                                                                                |     |                                     |
|     | $\langle \ \rangle K_{s}(q)$                                                                                                                                                                                               | _   | Formatted: Equation                 |
|     | $\langle \rho_{TL}^q \rangle = \left(\frac{R}{2}\right)^{\rho(T)} \langle \rho_{TR}^q \rangle$                                                                                                                             |     | Field Code Changed                  |
|     | $\begin{pmatrix} 1 \\ L \end{pmatrix}  \begin{pmatrix} 1 \\ r \\$                                                                                                                  |     |                                     |
|     | Where the angle brackets denote statistical averages and $K_{\alpha}(q)$ is the moment scaling function that characterizes the scaling of                                                                                  |     | Formatted                           |
| 420 | the $a_i^{\text{th}}$ order statistical moment order of the thermal resistance.                                                                                                                                            | /   |                                     |
|     | The thermal resistance is proportional to the inverse thermal diffusivity, therefore the effective HEBE diffusive transport                                                                                                |     |                                     |
|     | coefficient at scale L satisfies:                                                                                                                                                                                          |     | Formatted: Font: Italic             |
| I   |                                                                                                                                                                                                                            |     |                                     |
|     | 21                                                                                                                                                                                                                         |     |                                     |

|     | $(\mathbf{p})^{-K_{\rho}(-1)}$                                                                                                                            |                           | Formatted: Equation                |
|-----|-----------------------------------------------------------------------------------------------------------------------------------------------------------|---------------------------|------------------------------------|
|     | $D_{r,r} \propto \kappa_{r,r} \propto \left(\rho_r^{-}\right)^{-1} \approx \left \frac{\Lambda}{r}\right  \qquad D_{r,r}$                                 |                           | Field Code Changed                 |
|     | $F_{L} = n_{L} \left( F_{L} \right) \left( L \right) = F_{R} $ $(41)$                                                                                     |                           |                                    |
|     |                                                                                                                                                           |                           |                                    |
|     | Finally using $l \propto D$ we obtain:                                                                                                                    | 1                         | Field Code Changed                 |
|     |                                                                                                                                                           |                           |                                    |
|     | $I_{h,L} \propto \left(\frac{L}{R}\right)^{K_{\rho}(-1)} I_{h,R} \tag{12}$                                                                                |                           | Formatted: Equation                |
|     |                                                                                                                                                           |                           | Field Code Changed                 |
| 125 |                                                                                                                                                           |                           |                                    |
| 425 | (42)                                                                                                                                                      |                           |                                    |
|     | Which relates the transport length at small scales L and planetary scales R. Depending on $K_0(-1)$ , the ratio $l / l$ can be                            | _                         | Formatted: Font: Italic            |
|     |                                                                                                                                                           |                           | Formatted: Font: Italic            |
|     | where $V_{i}$ is a second state of the second state of the second state $V_{i}(z) = C_{i}(z, 1)$ and $V_{i}(z) = C_{i}(z, 1)$                             |                           | Field Code Changed                 |
|     | quite small. For example, if the thermal resistivity statistics are taken as log-normal, then: $\kappa_{\rho}(q) = C_1 q(q-1)$ so that             |                           | Field Code Changed                 |
|     |                                                                                                                                                           |                           |                                    |
|     | $K_1(-1) = 2C_2$ , so that $l_{r,r} \propto (L/R)^{2C_1} l_{r,r}$ . As discussed in appendix A, $C_1 \approx 0.16$ for the temperature in space (see also |                           | Field Code Changed                 |
|     |                                                                                                                                                           |                           | Formatted: Font: Italic            |
|     | [Lovejoy, 2018]). Using this value as a guide, we find $l_{h,L} \propto (L/R)^{0.32} l_{h,R}$ so that depending on the small scale resolution             |                           | Formatted: Subscript               |
|     |                                                                                                                                                           |                           | Field Code Changed                 |
| 430 | L, we can easily explain a factor of 10 or more increase in the effective transport length at large scales. Clearly the scale                             |                           | Formatted: Font: Italic            |
|     | dependence of $\kappa_{0}$ , $\kappa_{i}$ is an important topic for future FEBE research.                                                                 |                           | Formatted: Font: Italic, Subscript |
|     |                                                                                                                                                           |                           | Formatted: Font: Italic            |
| I   | ۸                                                                                                                                                  | $\langle \rangle \rangle$ | Formatted: Font: Symbol, Italic    |
|     |                                                                                                                                                           | $\langle \rangle$         | Formatted: Font: Italic, Subscript |
|     | 3 The inhomogeneous heat equation                                                                                                                         |                           | Formatted English (IIK)            |
|     | or the hitomogeneous near equation                                                                                                                        |                           |                                    |

**3.1 Babenko's method**

- 435 The homogeneous heat equation in a semi-infinite domain is a classical problem and conductive - radiative surface boundary conditions naturally lead to fractional order operators, the HEBE and GHEBE. Although we have seen that fractional operators appear quite naturally, their advantages are much more compelling for the more realistic inhomogeneous equations relevant for the Earth. We will therefore now proceed to derive the inhomogeneous HEBE and GHEBE using Babenko's method. The more usual application is to find the surface heat flux given a solution to the conduction equation (see for example [Magin et 440 al., 2004], [Chenkuan and Clarkson, 2018]), the following application appears to be original.
- In the inhomogeneous case with  $\tau = \tau(\underline{x})$ ,  $l_h = l_h(\underline{x})$ ,  $l_v = l_v(\underline{x})$ ,  $\underline{\alpha} = \underline{\alpha}(\underline{x})$ , there is no unique nondimensionalization. Therefore, we express the inhomogeneous anomaly heat equation with nondimensional operators as:

$$\left(\tau \frac{\partial}{\partial t} + l_h \zeta - \left(l_v \frac{\partial}{\partial z}\right)^2\right) T = 0; \qquad \zeta = \left(\underline{\alpha} \cdot \nabla_h + l_h \left(-\nabla_h^2\right)\right)$$

(4325)

Field Code Changed

Field Code Changed

Where we have used  $\kappa_v(\underline{x}) = l_v^2 \frac{\partial^2}{\partial z^2} = \left(l_v \frac{\partial}{\partial z}\right)^2$  and  $\zeta$  is a time independent horizontal transport operator allowing for

both advective and diffusive transport. Under the fairly general conditions, when  $\zeta$  operates on the temperature field, it is proportional to the nondimensional divergence of the horizontal heat flux (discussed in part I, see eq. 4). Since the forcing is via the surface boundary condition rather than by an inhomogeneous term, eq. 25-43 is mathematically homogeneous.

The first step in Babenko's method (see e.g. [Podlubny, 1999], [Magin et al., 2004]), is to factor the differential operator:

$$\left(\Lambda + l_{v}\frac{\partial}{\partial z}\right)\left(\Lambda - l_{v}\frac{\partial}{\partial z}\right)T = 0; \quad \Lambda = \left(\tau\frac{\partial}{\partial t} + l_{h}\zeta\right)^{1/2}$$
(4426)

450 As usual, the general solution of a homogeneous equation is a linear combination of elementary solutions A+ and A-:

$$\left(\Lambda + l_{v}\frac{\partial}{\partial z}\right)A_{+}\left(t,\underline{x};z\right) = 0; \quad \left(\Lambda - l_{v}\frac{\partial}{\partial z}\right)A_{-}\left(t,\underline{x};z\right) = 0 \tag{4527}$$

The A+ solution leads to solutions that diverge at  $Z = -\infty$  whereas A- leads to the required physical solutions with  $T(-\infty) = 0$

, ([Podlubny, 1999]). Therefore we are interested in solutions to:

$$\left(\Lambda - l_{v}\frac{\partial}{\partial z}\right)T(t,\underline{x};z) = 0$$
(4628)

455 putting z = 0 and  $using_{z} = -(l_{v} / s)\partial T / \partial z$  (part I, eq. 22)6, we obtain:

$$\left(\tau \frac{\partial}{\partial t} + l_{h}\zeta\right)^{1/2} T_{s} = l_{v} \frac{\partial T}{\partial z}\Big|_{z=0} = sQ_{s}; \qquad \begin{array}{c} T_{s}(t,\underline{x}) = T(t,\underline{x};0)\\ Q_{s}(t,\underline{x}) = -\left(\underline{Q}_{d}(t,\underline{x};0)\right)_{z} \end{array}$$
(4729)

where  $T_s(t, \underline{x})$  is the surface temperature anomaly and  $Q_s$  is the heat flux into the surface (the negative of  $Q_{s,d,z}$  which is the z component of the surface conductive (sensible) heat flux). Before interpreting the half order operator on the left, we can

| For  | matted: Font: Italic     |
|------|--------------------------|
| For  | matted: Font: Not Italic |
| For  | matted: Font: Not Italic |
| Fiel | d Code Changed           |
|      |                          |

| ( |      |         |
|----------|------|---------|
| Field    | Code | Changod |
| FIEIU    | Coue | Changed |

already give this equation a physical interpretation. When  $Q_s > 0$ , sensible heat is forced into the Earth, some of it is stored in

460 the subsurface (the  $\tau \frac{\partial}{\partial t}$  term, the same horizontal position x but stored by heating up the subsurface, z<0), and some of the

heat (the  $l_h \zeta$  term), is transported horizontally to neighbouring regions (and conversely when  $Q_s < 0$ ). We can also understand the basic difference between the  $A_+$  and  $A_-$  solutions: whereas the physically relevant  $A_-$  solution correspond to energy storage and horizontal transport in the region z<0, the  $A_+$  solutions correspond to the region z>0 assumed to be devoid of conducting material.

465 The final step is to use the fact that the conductive heat flux  $Q_{\rm c}$  is equal to the radiative imbalance (part I, fig. 1):

$$Q_s = R_{\uparrow} - R_{\downarrow} = \frac{T_s}{s} - F \tag{4830}$$

Combining the equations 29, 30 we obtain the inhomogeneous Generalized Half-order Energy Balance Equation (GHEBE):

$$\left(\tau(\underline{x})\frac{\partial}{\partial t} + l_h(\underline{x})\zeta(\underline{x})\right)^{1/2} T_s(t,\underline{x}) + T_s(t,\underline{x}) = s(\underline{x})F(t,\underline{x})$$
(4931)

If needed, the internal field  $T(t,\underline{x};z)$ , can be found by solving eq.  $31 \pm 49$  for  $T_s(t,\underline{x})$  which is the z = 0 boundary condition for 470 the full eq. 2543. We see that eq.  $31\pm 49$  reduces to the homogeneous GHEBE (eq. 17) when  $\tau$ ,  $l_h, \underline{x}, \underline{\alpha}$  are constant. By comparing this derivation with that of the homogeneous GHEBE via the classical Laplace-Fourier transform method (section 2.1), it is clear that Babenko's method is very similar, but is more general. Whereas in the homogeneous equation, where the transforms reduce the derivative operations to algebra, the difficulty with Babenko's method is to find proper

475 Laplace (or Fourier) transform methods still apply in the time domain, in the next section we discuss the more challenging interpretation of the fractional inhomogeneous spatial operators.

interpretations of the fractional operators. However, in the above, we assumed that  $\tau$  was only a function of position, so that

**3.2 The zeroth order high frequency GHEBE: the HEBE**

480

Before discussing the inhomogeneous GHEBE, consider the case where the horizontal term  $l_h\zeta$  is small compared to  $\tau \frac{\partial}{\partial x}$ .

below we argue that this is a good approximation for scales up to years and decades and greater than tens of kilometers (table 1, appendix A). Recall that the this horizontal transport term is in fact proportional to the divergence of the horizontal heat

flux so that it may be small even when heat fluxes are significant [Trenberth et al., 2009]. Alternatively, in globally averaged

24

| Field Code Changed |  |
|--------------------|--|
|--------------------|--|

Field Code Changed

Field Code Changed

models, there are no horizontal inhomogeneities so that  $\zeta = 0$ . In these cases  $\Lambda = \tau(\underline{x})^{1/2} \frac{\partial^{1/2}}{\partial t^{1/2}}$ ; and we obtain the

inhomogeneous HEBE as a special case of the inhomogeneous FEBE:

$$\tau(\underline{x})^{H} - D_{t}^{H} T_{s}(t, \underline{x}) + T_{s}(t, \underline{x}) = s(\underline{x}) F(t, \underline{x}); \qquad H = 1/2$$
(5032)

We have written it with a general *H* since as in part I, an inhomogeneous version of the EBE may be obtained with H =1. We have also used the Weyl derivative (i.e. from  $t = -\infty$ ) since this accommodated periodic or statistically stationary forcing as well as forcing starting at t = 0 (I this case we simply consider F = 0 for  $t \le 0$ ). Eq. 32-50 shows that the HEBE only depends on the local climate sensitivity and the local relaxation time. We'll see below that explicit dependence on the horizontal transport ( $v, \kappa_h$ ) and specific heat per volume  $\rho c$  is only important at scales somewhat smaller than the transport length scale (or alternatively at extremely long time scales, section 3.56). Before solving the HEBE, it is instructive to introduce the notation  $T_{\infty}(t, \underline{x}) = s(\underline{x})F(t, \underline{x})$ .  $T_{\infty}$  is the equilibrium temperature that would be reached at time t if at each location  $\underline{x}$ . Fwas suddenly stopped and fixed at that value. With this notation, we may integrate both sides of eq. 32-50 by order *H*, and multiply by  $\tau$ .H to obtain:

$$T_{s}(t,\underline{x}) = \frac{1}{\Gamma(H)} \int_{-\infty}^{t} \left( \frac{t-u}{\tau(\underline{x})} \right)^{H-1} \left( T_{\infty}(u,\underline{x}) - T_{s}(u,\underline{x}) \right) \frac{du}{\tau(\underline{x})}; \quad 0 < H < 1$$
(5133)

Written in this form, it is obvious that the temperature is constantly relaxing in a power law manner to  $T_{\infty}$  (although if *F* and is time dependent, equilibrium will in general never in fact be established). In the usual EBM special case (*H* = 1), the power law must be replaced by an exponential, the HEBE is obtained with H = 1/2. Since  $T_{\infty} = sF$ , physically the deviation from  $T_{\infty}$  - the term  $\tau^{H}_{-\infty} D_{t}^{H} T_{s}$  (eq. 3250) - physically corresponds to the energy imbalance, as before, it is a power law, long

memory energy storage term.

500 The FEBE is a linear differential equation that can be solved using Green's functions [*Miller and Ross*, 1993], [*Podlubny*, 1999]. The solution is:

$$T_{s}(\underline{x},t) = \frac{s(\underline{x})}{\tau(\underline{x})} \int_{-\infty}^{t} G_{\delta,H}\left(\frac{t-u}{\tau(\underline{x})}\right) F(\underline{x},u) du$$

(5234)

Field Code Changed

Field Code Changed

Field Code Changed Field Code Changed

where  $G_{0}G_{0,H}$  is the *H* order Mittag-Leffler impulse response Green's function ([*Lovejoy*, 2019a]). In general,  $G_{0,H}$  is only expressible in terms of infinite series, exceptions are the H = 1 EBE ( $G_{0,H} = e^{-t}$ ); and the  $H = \frac{1}{2}$  HEBE (eq. 33 with in the

505 notation above
$$G_{0,1/2}(t) = G_{\delta}(t;0) = \frac{1}{\sqrt{\pi t}} - e^t erfc \sqrt{t}$$
 (eq. 31, part I)

The corresponding step response  $G_{0,1/2} = G_{1,1/2} = G_{0,1/2} = G_{0,1/2$

**510 3.3 Some features of stochastic forcing**

515

The FEBE and the HEBE are examples of fractional relaxation equations; these have primarily been discussed in the context of deterministic forcings that start at t = 0. The corresponding stochastic fractional relaxation processes (in physics, "fractional Langevin equations", (FLE) see the references in [*Lovejoy*, 2019a]) - here corresponds to stochastic internal forcing. The FLE have has received little attention, although [*Kobelev and Romanov*, 2000], [*West et al.*, 2003] discuss the corresponding nonstationary random walks. The statistically stationary stochastic case that results when Weyl rather than Riemann-Liouville fractional derivatives are used is treated in [*Lovejoy*, 2019a], including the HEBE autocorrelation function and prediction problem (and its limits) when *F* is a Gaussian white noise.

To understand the noise driven HEBE, it is helpful to Fourier analyze it using  $\left( \int_{-\infty}^{\infty} D_{t}^{H} \right)^{Fourier} (i\omega)^{H}$  [Lovejoy, 2019a], section 3.3 part I-and appendix C. At high frequencies, the derivative (energy storage) term dominates so that the temperature is a fractional integral (order H) of the forcing. At low frequencies, the derivative term can be neglected so that  $T \approx \frac{\partial F}{\partial F}$  implying

that the equilibrium temperature follows the forcing and that  $s_{\tau}^{2}$  is indeed the usual climate sensitivity. Alternatively, in real space, if F(t) is a unit step function  $\Theta(t)$  and  $\underline{s} \neq = 1$ , then for  $H \neq 1$  the long time relaxation to the equilibrium temperature response; is a power law:  $\underline{G}_{\Theta,H}(t) \approx 1 - t^{-H}$  (part I eq. 33). Similarly, for small t, for H < 1, the impulse response is singular  $\underline{G}_{\delta,H}(t) \approx t^{H-1}$  (part I eq. 33). Due to this singularity, when F(t) is a Gaussian white noise, at high frequencies, T will be a fractional Gaussian noise (fGn) with exponent  $H_{IGn} = H - \frac{1}{2}$ ; averages over time  $\Delta t$  will behave

as  $\langle T_{\Delta t}^2 \rangle^{1/2} \propto \Delta t^{H_{fGn}}$ . When  $H \le 1/2$  ( $H_{fGn} \le 0$ ) and the resolution is increased ( $\Delta t \rightarrow 0$ ), this implies strong resolution dependencies (mathematically, small scale divergences) when the resolution is increased ( $\Delta t \rightarrow 0$ ) and so it is important in data analysis, including the estimation of the temperature of the Earth [*Lovejoy*, 2017]. When forced by a white noise, the HEBE is exactly at the critical value  $H_{fGn} = 0$  corresponding to a "1/f" noise (note that the Earth's internal variability forcing

530 is not necessarily a white noise, it might have a different scaling behaviour research in progress indicates that it is at least close

26

to a white noise). A particularly relevant aspect is that the correlation function and spectrum change very slowly from high to low frequencies [*Lovejoy*, 2019a]. With data over a limited ranges of scales – e.g. months to decades – then, depending on the relaxation time  $\tau$ , the HEBE could mimic the FEBE with any *H* in the range  $0 < H \le \frac{1}{2}$  (hence  $-\frac{1}{2} \le H_{JGn} \le 0$ ). It can therefore potentially account for the geographical variations in *H* reported in [*Lovejoy et al.*, 2017] as being spurious consequences of

535 geographical variations in  $\tau(\underline{x})$ .

At global scales, the high and low frequency HEBE behaviours are close to observations. For example, the global value  $H = 0.5\pm0.2$  was found for the long time behaviour needed to project the earth's temperature to 2100 [*Hebert*, 2017]. [*Hébert et al.*, 2020], and [*Procyk et al.*, 2020] also using centennial scale global temperature estimates but using the FEBE directly, found the less uncertain  $H = 0.38\pm0.05$ ; and using data at monthly and seasonal scales [*Del Rio Amador and Lovejoy*, 2019]

- 540 found the value *H* = 0.42±0.03 and used itfor the internal macroweather variability needed to make monthly and seasonal forecasts [*Del Rio Amador and Lovejoy*, 2019] (note that this was inferred by make the usual assumption that the internal foreing *F* is a Gaussian white noise, and this may not be the case). Appendix B discusses the spatial cross correlation matrix implied by the HEBE that is needed for example in calculating Empirical Orthogonal Functions (EOFs, or for the space-time macroweather model developped in [*Del Rio Amador and Lovejoy*, 2020b]).
- 545 If the could also mention that if F is spatially statistically homogeneous and independent of the parameters  $\lambda$ ,  $\tau$ , then not only will the macroweather temperature fluctuations be well reproduced, but also, up to the relaxation time, the temperature may easily respect a space-time symmetry called space-time statistical factorization, ("STSF"; e.g.  $R_{space-time}(\Delta x, \Delta t) = R_{space}(\Delta x)R_{time}(\Delta t)$  where R represents the autocorrelation function), see appendix C. Empirically, the STSF is at least approximately obeyed by space-time temperature and precipitation fluctuations ([Lovejoy and de Lima, 2015]), and
- 550 if respected, the STSF has important implications for macroweather temperature forecasting. Although the HEBE was derived for anomalies, these were not defined as small perturbations but rather as time-varying components of the full solution of the temperature (energy) equation with the time independent part corresponding to the climate state. The only point at which T was assumed to be small was with respect to the absolute local climate temperature about which the black body radiation was linearized, a fairly weak restriction on T. We could also mention that by allowing
- 555 the albedo or other parameters to change in time, the HEBE could easily be extended to the study of past or future climates where it would broaden the spectrum potentially improving the modeling of glacial cycles. An important feature of fractional differential operators is that they imply long memories, this is the source of the skill in macroweather forecasts ([Lovejoy et al., 2015], [Del Rio Amador and Lovejoy, 2019]). The fractional term with the long memory corresponds to the energy storage process. In contrast, [Lionel et al., 2014] introduced a class of ad hoc Energy
- 560 Balance Models with Memory (EBMM) whose (nonfractional) time derivative depends on integrals over the past state of the system.

**3.4 The first order in space GHEBE**

The HEBE is the GHEBE limit where horizontal transport effects are dominated by temporal relaxation processes and are ignored. Although this spatial scale depends on the time scale, appendix A estimates that at monthly time scales, this spatial scale is of the order ofless ≈10 km and even at centennial scales it may only be only 100km or so. For these small spatial scales, we follow [*Babenko*, 1986], [*Kulish and Lage*, 2000], [*Magin et al.*, 2004], and expand the square root operator using the binomial expansion:

$$\Lambda = \tau^{1/2} \sqrt{\frac{\partial}{\partial t} + V\zeta} \approx \left(\tau \frac{\partial}{\partial t}\right)^{1/2} \left(1 + \frac{1}{2} \left(\frac{\partial}{\partial t}\right)^{-1} V\zeta - \frac{1}{8} \left(\frac{\partial}{\partial t}\right)^{-2} \left(V\zeta\right)^{2} + \dots\right)$$

$$\quad V = \frac{l_{h}}{\tau} = \left(\frac{\kappa_{h}}{\kappa_{v}}\right) \frac{1}{\rho cs}$$
(5335)

**(for the expansion to be strictly valid, $\tau$ must be a constant in time and in space; we have already assumed that $V\zeta$ is**

independent of time). As usual with Babenko's method, a rigorous mathematical justification is not available ([*Podlubny*, 1999]), although recall that  $\tau$ , and  $l_h$  are only functions of position so that for the temporal operator, Laplace and Fourier transforms techniques still work.

575 Considering the spatial part of the fractional operator, we see that it is weighted by the effective heat transport velocity V; as shown below, it plays the role of a small parameter (table 1, appendix A estimate it as  $\approx 10^{-4}$ m/s). Therefore, dropping the subscript "s" here and below, the GHEBE is:

$$\tau^{1/2} \left(\frac{\partial}{\partial t} + V\zeta\right)^{1/2} T + T =$$

$$\tau^{1/2} \sum_{-\infty} D_t^{1/2} T + T + \frac{1}{2} V \tau^{1/2} \left(\sum_{-\infty} D_t^{-1/2} \zeta\right) T - \frac{1}{8} V^2 \tau^{1/2} \left(\sum_{-\infty} D_t^{-3/2} \zeta^2\right) T + \dots = sF$$
(5436)

with the Weyl fractional derivatives (these are partial fractional derivatives).

580 Keeping only the spatial terms leading in the small parameter V, we have the first order (in space) GHEBE:

$$\frac{\tau^{1/2}}{2} D_{t}^{1/2}T + T + \frac{1}{2}V\tau^{1/2}\left(\sum_{\infty} D_{t}^{-1/2}\zeta\right)T = sF$$
(5537)

Or:

$$\tau^{1/2} {}_{-\infty} D_t^{1/2} T + T + \frac{1}{2} \tau^{1/2} {}_{-\infty} D_t^{-1/2} \left( \underline{\nu} \cdot \nabla_h T - \kappa_h \nabla_h^2 T \right) = sF$$
(5638)

This equation is apparently similar to the usual transport equation. To see this, operate on both sides by  $\tau^{-1/2} - D_t^{1/2}$ , to obtain:

$$\frac{\frac{\partial T}{\partial t} + \underline{v}' \cdot \nabla T - \kappa' \nabla^2 T + \tau^{-1/2} \, _{-\infty} D_t^{1/2} T = sF'}{\underline{v}'}$$

$$\frac{\underline{v}' = \frac{1}{2} \underline{v}; \qquad \kappa' = \frac{1}{2} \kappa; \qquad F' = \tau^{-1/2} \, _{-\infty} D_t^{1/2} F$$
(5739)

Except for the factor ½, the half order derivative term and the "effective", (roughened) forcing, this is the usual transport equation. Nevertheless, although tempting, it would be wrong to think of this simply as a usual transport equation with an
extra fractional term. The reason is that the extra term is not a small perturbation, it is dominant except at small spatial scales. On the contrary, it is rather the classical transport terms that are small perturbations to the main HEBE. Alternatively, without

the
$$\frac{\partial T}{\partial t}$$
 term, eq. 41-59 is a generalized fractional diffusion equation (e.g. [*Coffey et al.*, 2012]), although still with a key

difference being that the fractional derivative is Weyl, not Riemann-Liouville (i.e. over the range  $-\infty$  to t, not 0 to t).

**3.5 Climate states, Thermodynamic equilibrium and the low frequency GHEBE**

**595 3.5.1 The equilibrium temperature distribution: The the HEBE thermodynamic climateequilibrium**

The HEBE applies to time scales sufficiently short and to spatial scales sufficiently large that the horizontal temperature fluxes are too slow to be important, they are neglected. The first order correction (eqs. 3856, 3957) makes a small improvement by giving a more realistic treatment of the small scale horizontal transport. However, a long time after performing a step increase of the forcing, the time derivatives vanish and a new climate state is reached. If the temperature followed the pure HEBE, the
 spatial pattern for thermodynamic equilibrium temperature distribution would be determined by setting the HEBE time derivative to zero:

$$T_{eq,HEBE}(\underline{x}) = F_0 s(\underline{x}); \qquad F(t,\underline{x}) = F_0 \Theta(t)$$
(5840)

Where the subscript "eeq" indicates the long time equilibrium (climate) FEBE limit. However, appendix A shows that – depending on the nature of the horizontal transport - at scales perhaps -of the order of millenniacenturies, the horizontal heat fluxes will dominate the relaxation processes so that for very long times, this HEBE estimate is only approximate.

29

Field Code Changed

**3.5.2 Equilibrium and approach to equilibrium in the inhomogeneous GHEBE**

To understand the long time behaviour, we return to the GHEBE but perform a (long-time) binomial expansion of the halforder operator assuming that the transport terms dominate:

$$\int \left( l(\underline{x})\zeta(\underline{x}) + \tau \frac{\partial}{\partial t} \right)^{1/2} T = \left( l\zeta \right)^{1/2} \left( 1 + \left( l\zeta \right)^{-1} \tau \frac{\partial}{\partial t} \right)^{1/2} T$$

$$\approx \left( l\zeta \right)^{1/2} T + \frac{1}{2} \frac{\partial}{\partial t} \left( (l\zeta)^{-1/2} \tau \right) T - \frac{1}{8} \frac{\partial^2}{\partial t^2} \left( (l\zeta)^{-1/2} \tau \left( l\zeta \right)^{-1} \tau \right) T + \dots$$
(5941)

(from here on we drop the "h" subscripts on l and the gradient operator). Again, to be strictly valid,  $\tau$  must be a constant so

that  $l(\underline{x})\zeta(\underline{x})$  and  $\tau \frac{\partial}{\partial t}$  commute). We have to be careful since the advection length and relaxation times are functions of position (but not time) so that the spatial operators don't commute. Keeping terms to first order in time, we obtain:

$$\left(l\zeta\right)^{1/2}T + T + \frac{1}{2}\frac{\partial}{\partial t}\left(\left(l\zeta\right)^{-1/2}\tau\right)T = sF$$
(6042)

To make progess, let's choose the transport operator so that its half powers are easy to interpret. The simplest approach is consider only diffusive transport and to use an isotropic fractional operator defined over the surface of the earth. For an arbitrary test function  $\rho$ , the corresponding order *H* fractional integral is:

$$\left(-\nabla^{2}\right)^{-H/2}\rho = I^{H}_{iso,d}\rho = \frac{1}{\Gamma(H)}\int_{\Omega} \frac{\rho(\underline{y})d^{d}\underline{y}}{\left|\underline{x}-\underline{y}\right|^{d-H}}$$
(6143)

(for  $0 \le H \le d$ , where *d* is the dimension of space, here d = 2, see e.g. [Schertzer and Lovejoy, 1987], appendix A). This can be understood since in Fourier space, the Laplacian is  $-\nabla^2 \xrightarrow{F.T.} |\underline{k}|^2$  and its inverse is  $(-\nabla^2)^{-1} \xrightarrow{F.T.} |\underline{k}|^2$ , the "Poisson solver". Note that eqs. 4260, 4361 involve ½ order inverse Laplacians which are H = 1 (rather than  $H = \frac{1}{2}$ ) isotropic integrals (eq. 4361). With the help of spherical harmonics, Appendix DAppendix C generalizes the results of section 2.3 gives the corresponding operators and their fractional extensions on the surface of the sphere. Applying eq. 4361 to the case d = 2 and H = 1 we have:

30

$$\quad \left(-\nabla^2\right)^{-1/2} \rho = \int_{\Omega} \frac{\rho(\underline{x}') d^2 \underline{x}'}{|\underline{x} - \underline{x}'|} \tag{6244}$$

Therefore, let us define a diffusive type transport operator  $l\zeta$  and its inverse  $(l\zeta)^{-1}$  implicitly from its inverse half-order power:

$$(l\zeta)^{-l/2} = l^{-1} (-\nabla^2)^{-l/2}; \qquad (l\zeta)^{l/2} = (-\nabla^2)^{l/2} l = (-\nabla^2)^{-l/2} (-\nabla^2) l \qquad (\underline{6345})$$

Hence let us define the half-order operator by:

640

$$\quad \left(l\zeta\right)^{-1/2} T\left(\underline{x}\right) = l\left(\underline{x}\right)^{-1} \int_{\Omega} \frac{T\left(\underline{x'}\right) d^2 \underline{x'}}{|\underline{x} - \underline{x'}|} \tag{6446}$$

With this definition the surface temperature equation  $\underline{60}$  becomes:

$$\frac{1}{2}\frac{\partial}{\partial t}\left[l(\underline{x})^{-1}\int_{E}\frac{\tau T(\underline{x}',t)d^{2}\underline{x}'}{|\underline{x}-\underline{x}|}\right]+T(\underline{x},t)-\int_{E}\frac{\nabla^{2}(l(\underline{x}')T(\underline{x}',t))d^{2}\underline{x}'}{|\underline{x}-\underline{x}'|}=s(\underline{x})F(\underline{x},t)$$
(6547)

Where the range of the integration Ω = E is the entire surface of the earth. This equation has only superficial links to equations studied in the literature such as the "generalized fractional advection-dispersion equation" (e.g. [*Meerschaert and Sikorskii*, 2012], [*Hilfer*, 2000]). We can now consider the system reaching equilibrium after a step forcing F(x,t) = F0(x)Θ(t), (increase by F0(x) "turned on" at t = 0). At long enough times, the earth reaches thermodynamic\_equilibrium and, the time derivative term vanishes and we obtain the equation for the equilibrium (climatological) temperatures:

$$T_{eq}(\underline{x}) - \int_{E} \frac{\nabla^{2} \left( l(\underline{x}') T_{eq}(\underline{x}') \right) d^{2} \underline{x}'}{|\underline{x} - \underline{x}|} = s(\underline{x}) F_{0}(\underline{x})$$
(6648)

To obtain an approximate solution, let's now assume that  $T_{eq}(\underline{x})$  differs from the climatological FEBE climate temperature  $T_{eT_{eq},FEBE}(x)$  by a small perturbation  $\delta T(x)$ .

$$T_{eq}\left(\underline{x}\right) = T_{eq,HEBE}\left(\underline{x}\right) + \delta T\left(\underline{x}\right); \quad T_{eq,HEBE}\left(\underline{x}\right) = s\left(\underline{x}\right)F_0\left(\underline{x}\right)$$
(6749)

then, using  $\mathcal{I}_{e}\underline{T}_{eq}(\underline{x}) \approx \underbrace{s}{\underline{s}} \frac{\lambda(\underline{x})}{F_0(\underline{x})}$  in the integral, we obtain the approximation:

$$T_{eq}(\underline{x}) \approx T_{eq,HEBE}(\underline{x}) + \delta T(\underline{x}); \qquad \delta T(\underline{x}) = \int_{E} \frac{\nabla^{2} \left( l(\underline{x'}) s(\underline{x'}) F_{0}(\underline{x'}) \right) d^{2} \underline{x'}}{|\underline{x} - \underline{x'}|}$$
(6850)

 $\delta T(\underline{x})$  is the slow, diffusive correction to the "instantaneous" (fast, high frequency), HEBE climate sensitivity  $\underline{s + (\underline{x})}$  that is estimated at usual (e.g. decadal) scales. As expected, since this is the long time solution after a step perturbation, it doesn't depend on  $\tau$ .

Horizontal transport of heat redistributes the energy fluxes locally, but since the GHEBE is linear, it shouldn't affect the overall (global) energy balance. Let us check this by direct calculation of the globally averaged temperature. Averaging eq. 4866, we obtain:

$$\qquad \overline{T_{eq}(\underline{x})} - \overline{\int_{E}^{\nabla^{2}\left(l\left(\underline{x}'\right)T_{eq}\left(\underline{x}'\right)\right)d^{2}\underline{x}'}}_{|\underline{x}-\underline{x}'|} = \overline{s(\underline{x})F_{0}(\underline{x})}; \qquad \qquad \overline{f} = \frac{1}{A_{E}}\int_{E}^{f}f(\underline{x})d^{2}\underline{x}$$

$$A_{E} = \int_{E}^{d^{2}\underline{x}}$$

$$(6954)$$

Where the spatial averaging operator (overbar) is defined for an arbitrary function *f*. The average of the horizontal heat flux term yields:

$$\frac{1}{A_E} \int_E \sum_E \frac{\nabla^2 \left( l\left(\underline{x'}\right) T_{eq}\left(\underline{x'}\right) \right)}{|\underline{x} - \underline{x}|} d^2 \underline{x} d^2 \underline{x'} = K_E \int_E \nabla^2 \left( l\left(\underline{x'}\right) T_{eq}\left(\underline{x'}\right) \right) d^2 \underline{x'} = \int_{\delta E} d\underline{s} \cdot \nabla \left( l\left(\underline{x'}\right) T_{eq}\left(\underline{x'}\right) \right) = 0$$
(7052)

Where  $K_E$  is an unimportant constant from the x integration, independent of x'. The far right equality is an application of the divergence theorem on the surface *E* whose boundary is  $\delta E$ ,  $d\underline{s}$  is a vector parallel to the bounding line. But since the integration is over the whole earth surface (*E*), there is no boundary, hence the result. We conclude that while horizontal diffusion transports heat over the earth's surface, it does not affect the overall global radiation budget:  $\overline{T_{eq}} = \overline{T_{eq,HEBE}}$ .

**4. Conclusions**

660

Up until now, at macroweather and climate scales, the Earth's energy balance has been modelled using two classical approaches. On the one hand, Budyko - Sellers models assume the continuum mechanics heat equation holds, this yields yielding a 1-D latitudinally varying climate state. On the other hand, there are the zero-dimensional box models that combine Newton's law of cooling with the assumption of an instantaneous temperature-storage relationship. Both models avoid the critical conductive - radiative surface boundary conditions; the former by ignoring heat storage, redirecting radiative

imbalances meridionally away from the equator, the latter by postulating a surface heat flux that is not simultaneously consistent with the heat equation and energy conservation across a conducting and radiating surface (part I).

- This two part paper re-examined the classical heat equation with classical semi-infinite geometry. In the horizontally homogeneous case (part I), the fundamental novelty is the treatment of the conductive radiative boundary conditions, here (part II), it is the use of Babenko's method to extend this to the more realistic horizontally inhomogeneous problem. In both cases, the semi-infinite subsurface geometry is only important over a shallow layer of the order of the diffusion depth where 670 most of the storage occurs (roughly estimated as ≈ 100m in the ocean, ≈<10m over land, see table 1 and appendix A).
- The key result was obtained by using standard Laplace and Fourier techniques. It was shown quite generally that the surface temperatures and heat fluxes are related by a half-order derivative relationship. This means that if Budyko-Sellers models are right that the continuum mechanics heat equation is a good approximation to the Earth averaged over a long enough time then that a consequence is that the energy stored is given by a power law convolution over its past history. This is a general
- 675 consequence of the conductive radiative surface boundary conditions in semi-infinite geometry and is very different from the box models that assume that the relationship between the temperature and heat storage is instantaneous. Although the system itself is classical, this result may be viewed as a nonclassical example of the Mori-Zwanzig mechanism in which system parameters that are not modelled explicitly (here, the subsurface temperatures) imply long (power law) memories for the modelled parameters (here, the surface temperatures). This is in contrast to conventional short (exponential) memory assumption. It implies that any part of the Earth system that exchanges energy both radiatively and conductively into a surface should be modelled with fractional rather than integer ordered derivatives. A far reaching consequence is that classical

  - dynamical systems approaches based on integer ordered differential equations are not necessarily pertinent to the climate system.

If we ignore horizontal heat transport (part I), an immediate consequence of half order storage is that the temperature obeys the Half-order Energy Balance Equation (HEBE) rather than the classical first order-one EBE. Depending on the space-time

- statistics of the anomaly forcing, the HEBE justifies the current Fractional EBE (FEBE) based macroweather (monthly, seasonal) temperature forecasts [Lovejoy et al., 2015], [Del Rio Amador and Lovejoy, 2019], [Del Rio Amador and Lovejoy, 2020a; Del Rio Amador and Lovejoy, 2020b] that are effectively high frequency approximations to the FEBE). Similarly, the low frequency (asymptotic) power law part can produce climate projections with significantly lower uncertainties than current
- 690 GCM based alternatives ([Hebert, 2017], [Hébert et al., 2020] and work in progress directly using the HEBE, [Procyk et al., 2020]).

The implied long time storage behaviour explains the success of scaling based climate projections[Hébert et al., 2020; Procyk et al., 2020] and, the implied short time behaviour potentially explains the success of macroweather forecasts that exploit it[Del Rio Amador and Lovejoy, 2019; 2020a; Del Rio Amador and Lovejoy, 2020b].—When the system is periodically forced, the

- 695 response is shifted in phase and borrowing from the engineering literature the surface is characterized by a complex thermal impedance that we showed is equal to the (complex) climate sensitivity. In part I, we gave evidence that this quantitatively explains the phase lag (typically of about 25 days) between the annual solar forcing and temperature response.
  - 33

In this second part, we investigated the consequences of horizontal heat transport, first in a homogeneous medium with inhomogeneous forcing (section 2)-first on a plane and then – permitting a direct comparison with the usual Budyko-Sellers approach - on the sphere (section 2). In section 3 and then we considered, more generally with inhomogeneous material properties (including variable diffusion lengths, relaxation times, and climate sensitivities, section 3). While Laplace and Fourier techniques can still be used in time, they cannot be used in space, and not so useful here, but theHowever, the extension to inhomogeneous media was nevertheless possible thanks to Babenko's powerful (but less rigorous) operator method. Whereas in part I, the homogeneous fractional space-time operator was given a precise meaning, here - following Babenko - 705 the corresponding inhomogeneous operator was interpreted using binomial expansions for both the short and long time limits.

705 the corresponding inhomogeneous operator was interpreted using binomial expansions for both the short and long time limits and yield 2D energy balance models. Part II thus allows us for the first time to extend energy balance models to 2-D, allowing the treatment of regional temporal anomalies.

The expansions depend both on the space and time scale and on a dimensional parameter: the typical horizontal transport speed (*V*), estimated as  $\approx 10^{-4}$ m/s (appendix A). The zeroth order expansion in time limit yielded the inhomogeneous HEBE, the

- 710 first order correction yielded an equation that superficially resembled the usual heat equation but instead had a leading half order time derivative term. Based on the analysis of NCEP reanalyses (appendix A), it was argued that at spatial scales larger than hundreds of kilometers, that these approximations are likely to be useful for years, decades, and perhaps longer. However, for studying climate states defined for example as the thermodynamic equilibrium state for forcings that are increased everywhere in step function fashion we required low, not high frequency expansions and these are based on fractional spatial
- 715 operators. We defined inhomogeneous fractional diffusion operators in both flat space and on the sphere (appendix Dappendix C), and derived equations for both the therm equilibrium odynamic limit and the approach to the limit. We showed that (as expected) they conserved energy and that the low frequency climate sensitivity is somewhat different from that estimated at higher frequencies (from the EBE or HEBE).

The EBE and HEBE are the H = 1, H = 1/2 special cases of the Fractional EBE (FEBE) that was recently introduced as a phenomenological model [*Lovejoy et al.*, 2020]; (see also\_ [*Lovejoy*, 2019a], [*Lovejoy*, 2019b]) with empirical estimates  $H \approx$

- 0.4 0.5, i.e. very close to the HEBE. Although only a special case, the HEBE illustrates the general features of the FEBE fractional-order energy storage term and power law long memories. in [Lovejoy et al., 2020]. [Lovejoy, 2019a] discussed the statistical properties of the FEBE driven by Gaussian white noise (a model for the internal variability forcing) showing that the high frequency limit is a process called fractional Gaussian noise (fGn). In the special HEBE case with H = 1/2, the fGn
- 725 temperature response has exactly a high frequency 1/f spectrum that is cut-off at the relaxation time (empirically of the order of a few years). [Lovejoy, 2019a] developed optimal predictors and determined the predictability skill.

Whereas the more general FEBE is essentially a phenomenological model up until now justified by the hypothesized scale invariance of the energy storage mechanisms ([Lovejoy et al., 2020]), the HEBE follows directly and quite generally from the continuum mechanics heat equation, thus giving it a more solid theoretical basis. However, the work here suggests another way to obtain the FEBE: to replace the classical heat equation by its fractional generalization, the fractional heat equation, a

possibility that we explore elsewhere. Part II allowed us for the first time to extend energy balance models to 2-D, allowing

730

the treatment of regional temporal anomalies. Depending on the space-time statistics of the anomaly forcing, the HEBE justifies the current Fractional EBE (FEBE) based macroweather (monthly, seasonal) temperature forecasts [Lovejoy et al., 2015], [Del Rio Amador and Lovejoy, 2019]. Similarly, the low frequency (asymptotic) power law part can produce climate

735 projections with significantly lower uncertainties than current GCM based alternatives ([Hebert, 2017; Hébert et al., 2020]

and work in progress directly using the HEBE [Procyk et al., 2020]with R. Procyk). This work was performed in the spirit of Budyko-Sellers models in which the Earth system is averaged over scales longer than typical lifetimes of planetary scale weather structures. Following Budyko-Sellers, the key physical assumption was that the resulting averaged system is a continuum system, thus justifying use of the general continuum mechanics heat equation. From

740 this, the GHEBE and HEBE follow from the surface conductive-radiative boundary condition. In as much as GCMs (that are based on continuum mechanics) reproduce the same statistics as the noise—or anthropogenically forced FEBE and HEBE, the continuum hypothesis is plausible.

As a final comment, we should mention that although this paper focused on the time varying anomalies with respect to a time independent climate state, our approach opens the door to new methods for determining full 2-D climate states (generalizations of the 1-D Budyko-Sellers type climates) but also to determining past and future climates and the transitions between them. This is because the definition of temperature "anomalies" is very flexible. For example, we could first apply the method to determining the existing climate by fixing the forcing at current values and solving the time independent transport equations. Then, the long term effect of changes such as step function increases in forcing could be determined from the GHEBE anomaly equation (section 3.5) which regionally corrects the local climate sensitivities for (slow) horizontal energy

750 transport effects. -Nonlinear effects that can be modelled by temperature dependent forcings (i.e.

 $F(\underline{x},t) \rightarrow F(\underline{x},t,T(\underline{x},t))$  can easily be introduced. Other nonlinear effects needed to account for Milankovitch cycles could thus easily be made, the primary difference being the half-order derivatives and the scaling that they imply. Indeed, the power law relaxation processes implied by the GHEBE suggests straightforward explanations for the observed power law climate regime spanning the range from centennial to Milankovitch scales.

**755 5. Acknowledgements**

I acknowledge discussions with L. Del Rio Amador, R. Procyk, R. Hébert, D. Clarke and C. Penland. This is a contribution to fundamental science; it was unfunded and there were no conflicts of interest. The data used in appendix A are from the NOAA website: <a href="https://www.esrl.noaa.gov/psd/data/gridded/data.ncep.reanalysis.html">https://www.esrl.noaa.gov/psd/data/gridded/data.ncep.reanalysis.html</a>.

Field Code Changed

Field Code Changed

Field Code Changed

**Appendix A: Empirical analysis of the horizontal structure**

760

In order to apply our results to the Earth, we need some idea of the magnitudes of various terms in our equations. To start with, recall that our model is of the Earth system at macroweather and climate time scales i.e. all relevant quantities are averaged over the weather scales  $\approx 10$  days or longer. The resulting averaged system is then treated as a continuum and the general continuum mechanics heat equation is applied. In this, we essentially follow the Budyko - Sellers approach and consider that the diffusive transport is characterized by eddy (not molecular) diffusivities and that the vertical structure of this 765 averaged continuum is homogeneous (although it may vary considerably from place to place in the horizontal, see section 2.3 for a scaling (multifractal) model). Unlike Budyko - Sellers that treat the vertical as negligibly thick - they don't consider it at all - our key main difference is that we assume that it has a thickness of the order of a few diffusion depths, and then we apply the key conductive- radiative surface boundary condition.

Probably the most important aspect is to estimate the relative importance of the temporal relaxation (and storage) terms  $\tau \partial / \partial t$  in comparison to the horizontal transport terms  $l_h \zeta$  with  $\zeta = (\underline{\alpha} \cdot \nabla_h + l_h (-\nabla_h^2))(\underline{c}$  see eq. 2543). Indeed, 770

for judging their relative importance, the key parameter is the ratio of the transport to relaxation terms r:

....

$$r = V \frac{\zeta T}{\left(\partial T / \partial t\right)} = \frac{\left(\underline{\alpha} \cdot \nabla_h + l_h \left(-\nabla_h^2\right)\right) T}{\left(\partial T / \partial t\right)}; \qquad \qquad V = \frac{l_h}{\tau}; \qquad \alpha = \frac{v}{V}$$
(7153)

Where  $\alpha$  is the magnitude of the dimensionless advection velocity vector  $\underline{\alpha} = \underline{v}/V$ . When  $r \ll 1$ , the transport term is small compared to the temporal term, conversely when  $r \gg 1$ . In order to quantify this, it is convenient to consider the advective 775 ("a") and diffusive ("d") terms as well as their derivatives individually:

$$r_{a} = V \frac{\zeta_{a,x} T + \zeta_{a,y} T}{\left(\frac{\partial T}{\partial t}\right)}; \qquad \zeta_{a,x} T \approx \alpha_{x} \frac{\partial T}{\partial x}; \qquad \zeta_{a,y} T \approx \alpha_{y} \frac{\partial T}{\partial y}$$

$$r_{d} = V \frac{\zeta_{d,x} T + \zeta_{d,y} T}{\left(\frac{\partial T}{\partial t}\right)}; \qquad \zeta_{d,x} T = l_{h} \frac{\partial^{2} T}{\partial x^{2}}; \qquad \zeta_{d,y} T = l_{h} \frac{\partial^{2} T}{\partial y^{2}}$$

$$(7254)$$

In the macroweather regime, the temporal temperature fluctuation at time scale  $\Delta t$  is  $\Delta T(\Delta t) \approx T_{\Delta t}$  where  $T_{\Delta t}$  is the anomaly averaged over scale  $\Delta t$ ; empirically this is valid over the macroweather regime i.e. up to 10 - 30 years in the industrial epoch

36

Field Code Changed
(see e.g. *[Lovejoy and Schertzer*, 2013], *[Lovejoy*, 2013], *[Lovejoy et al.*, 2017]). The typical fluctuation can be estimated by the RMS anomaly:

$$s_{\Delta t}\left(\underline{x}\right) = \left(\overline{T_{\Delta t}^2}\right)^{1/2} \approx s_1\left(\underline{x}\right) \left(\frac{\Delta t}{\Delta t_1}\right)^{H_t}$$

(7355)

Where the overbar is the average over all the anomalies in a time series at a single location  $\underline{x}$ .  $\Delta t_1$  is a convenient reference time, here taken as 1 month. Empirically, the exponent  $H_t \approx 0$  to -0.2; this is similar to the high frequency result  $H_t = 0$  (i.e. for  $\Delta t < \tau$ ) predicted from the HEBE with white noise forcing, valid for  $\Delta t \approx < \tau$ . Hence for our present purposes the typical time 785 derivative is:

$$\frac{\partial T}{\partial t} \approx \frac{s_{\Delta t}}{\Delta t}$$

This is the resolution  $\Delta t$  time derivative. Since typical north-south gradients are larger than typical east-west ones, the meridional (y) component of the transport is dominant, so that we will focus on it:

$$\frac{\partial T}{\partial y} \approx \frac{\left(\overline{\Delta T_{\Delta x}(\Delta y)^2}\right)^{1/2}}{\Delta y} = \frac{\Delta s_{\Delta x}(\Delta y)}{\Delta y}; \qquad \frac{\partial^2 T}{\partial y^2} \approx \frac{\Delta \left(\overline{\Delta T_{\Delta x}(\Delta y)^2}\right)^{1/2}}{\Delta y^2} = \frac{\Delta^2 s_{\Delta x}(\Delta y)}{\Delta y^2}$$

790 Hence the meirdional contributions to the ratios ra, rd are:

$$\begin{aligned} r_{a,y} &= V\alpha \, \frac{\Delta t}{\Delta y} \Delta \log s_{\Delta t} \left( \Delta y \right) \\ r_{d,y} &= V l_h \, \frac{\Delta t}{\Delta y^2} \left( \left( \Delta \log s_{\Delta t} \left( \Delta y \right) \right)^2 + \Delta^2 \log s_{\Delta t} \left( \Delta y \right) \right) \end{aligned}$$

Where  $\Delta \log s_{\Delta t} (\Delta y) = \frac{\Delta s_{\Delta t} (\Delta y)}{s_{\Delta t}}$ , is the relative fluctuation in the RMS temperature at time scale  $\Delta t$ , spatial scale  $\Delta y$  and

since we are only interested in an order of magnitude - we took α ≈ αy. The estimate of the diffusive term uses a finite difference approximation to the Laplacian. *lh* is horizontal anomaly relaxationdiffusion length and α is the nondimensional advection speed ν/V (V = *lh*/τ, see below). To gauge the order of magnitudes, in the far right term of eq. 5876, we took the absolute value so that the result is an upper boundsuppressed the signs.

37

|                           | Formatted: Font: 10 pt |
|---------------------------|------------------------|
| $ \neg$                   | Formatted: Font: 10 pt |
| $\langle \rangle$         | Formatted: Font: 10 pt |
| $\langle \rangle \rangle$ | Formatted: Font: 10 pt |
|                           | Field Code Changed     |
|                           | Field Code Changed     |
|                           | Formatted: Font: 10 pt |
|                           | Field Code Changed     |

(7456)

(7557)

(7658)

To estimate  $l_{s}$  consider the volumetric specific heat  $\rho c$ . Ocean and land values are similar (respectively water:  $\rho c \approx 4 \times 10^6$ and soil:  $\rho c \approx 1 \times 10^6$  J/m3). For  $\lambda$ , the global mean value is  $\approx 0.8 \pm 0.4$  K/W/m2, (using the CO2 doubling value  $3 \pm 1.5$ C, 90% confidence interval and 3.71 W/m2 for CO2 doubling) with regional values a factor of ~2 higher or lower (IPCC AR5) yielding 800  $pc\lambda \approx 3x10^{6}$  s/m. The horizontal (eddy) diffusivity is  $\kappa_{\mu} \approx 1$  m2/s ([Sellers, 1969], [North et al., 1981]). The vertical diffusivity

is not used in the usual energy balance models, however in climate models, ocean values of  $\kappa_* \approx 10^4 \, {\rm m}^2/{\rm s}$  are typical [Houghton et al., 2001]. For soil, rough values of  $\kappa_{\star} \approx 10^{-6} \text{ m}^2/\text{s}$  (wet) and  $\kappa_{\star} \approx 10^{-7} \text{ m}^2/\text{s}$  (dry) are measured in [Márguez et al., 2016] so that for soils,  $l_{\star} \approx 3 - 10$ m.

Alternatively we can use  $\kappa_r = \tau/(pc\lambda)^2$  and the global estimates of  $\tau \approx 10^8$ s ([Hebert, 2017], [Procyk et al., 2020]work in\* progress with R. Procyk, or part I, section 3.3). From these, we obtain  $\kappa_{\nu} \approx 10^{-6}$  m2/s which is close to the model values. In conclusion, using  $\kappa_r \approx 10^{-5} - 10^{-4}$  m2/s yields  $l_r \approx 30 - 100$ m,  $l_{\mu} \approx 10$  km. Consequently, the diffusive based velocity parameter is  $V \approx l_{\rm h}/\tau \approx 10^{-4}$  m/s.

The best transport model diffusive, advective or both - is not clear, therefore let us estimate the magnitude of the advective velocity v assuming that it dominates the transport. The appropriate value is not obvious since most models just use eddy

810 diffusivity not advection for transport. One way for example [Warren and Schneider, 1979] is to note that typical meridional heat fluxes are of the order of 100 W/m2 over meridional bands whose temperature gradients AT are several degrees K. If this heat is transported by advection, it implies  $\nu \approx Q_{\rm st}/(\rho c \Delta T) \approx 10^{-5} - 10^{-4} {\rm m/s}$  (eq. 4, part I), hence, using  $V \approx 10^{4} {\rm m/s}$ (above), we find  $\alpha = v/V \approx 0.1 - 1$ .

| Quantity                     | Symbol | Values                                                                                                                          |   |
|------------------------------|---------------|---------------------------------------------------------------------------------------------------------------------------------|---|
| Volumetric specific heat     | рс     | water $\approx 4 \times 10^6$ , soil $\approx 1 \times 10^6$ J/(m 3 K).                                              | - |
|                              |               |                                                                                                                                 |   |
| Climate sensitivity   | s      | water $\approx 4 \times 10^6$ , soil $\approx 1 \times 10^6$ J/(m 3 K)                                               | _ |
| Relaxation time              | Ţ.            | global $\tau \approx 10^8 s$                                                                                                    | _ |
| Horizontal Diffusivity       | Kh            | 1 m²/s                                                                                                                   | - |
| Vertical diffusivity         | Ku            | ocean $\approx 10^{-4}$ m 2 /s, soil $\approx 10^{-6}$ m 2 /s, global $\approx 10^{-5}$ m 2 /s | _ |
| Diffusion depth              | 4             | ocean 300m, soils $\approx 3 - 10$ m, global $\approx 30 - 100$ m                                                               | - |
| Diffusion length             | 4             | ocean $\approx 30$ km, land 3 km, global $\approx 10$ km.                                                                       | _ |
|                              |               |                                                                                                                                 |   |
| Diffusive velocity parameter | V      | $3x 10^{-3} - 3x 10^{-4} \text{ m/s}$                                                                                           | _ |
| Nondimensional advection     | α.            | 0.1 - 1                                                                                                                  | - |
| velocity                     |               |                                                                                                                                 |   |
|                              | L             |                                                                                                                                 |   |

**Formatted: Indent: First line: 0 cm**

| Formatted: Underline                |      |
|-------------------------------------|------|
| Formatted: Centered                 |      |
| Formatted Table                     |      |
| Formatted                           | [6]  |
| Formatted                           | [7]  |
| Formatted: Font: 11 pt              |      |
| Formatted: Indent: First line: 0 cm |      |
| Formatted                           | [8]  |
| Formatted                           | [9]  |
| Formatted: Font: 11 pt              |      |
| Formatted                           | [10] |
| Formatted                           | [11] |
| Formatted                           | [12] |
| Formatted                           | [13] |
| Formatted                           | [14] |
| Formatted                           | [15] |
| Formatted                           | [16] |
| Formatted                           | [17] |
| Formatted                           | [18] |
| Formatted                           | [19] |
| Formatted: Font: 11 pt              |      |
| Formatted                           | [20] |
| Formatted                           | [21] |
| Formatted: Indent: First line: 0 cm |      |
| Formatted                           | [22] |
| Formatted                           | [23] |
| Formatted                           | [24] |
| Formatted                           | [25] |
| Formatted                           | [26] |
|                                     |      |

Table 1: Parameter estimates from part 1 section 3.1.2, see section 2.3 for some planetary scale estimates.

820

Table 1 summarizes the With these dimensional and nondimensional parameter estimates, the final step is to estimate values of the gradient and Laplacian terms (eq. 5876). Since s - and hence log s - are the amplitudes of temporal noises; these amplitudes vary stochastically from one spatial location to another. Due to the space-timetial scaling of the temperature anomalies (analysed in [Lovejoy and Schertzer, 2013]), we expect that their-the statistics of the logarithms (eq. 76) to follow power laws up to large scales. To quantify this, we used NCEP reanalysis data at 2.5° resolution from 1948 to present, and after removing the low frequency anthropogenic trend, we estimated the RMS temperature anomalies at each pixel; s(x). In fig. 6, we then calculated spatial zonal and meridional fluctuations  $\Delta logs(\Delta x)$ ,  $\Delta logs(\Delta y)$ , and from these their root mean square (RMS) values. From the figure, we see that to a good approximation:

$$\Delta \log s \left(\Delta x\right) \approx \left(\frac{\Delta x}{L_{EW}}\right)^{H_x} \qquad \qquad L_{EW} \approx 1.5 \times 10^7 m$$
$$H_x \approx H_y \approx 0.5$$
$$\Delta \log s \left(\Delta y\right) = \left(\frac{\Delta y}{L_{NS}}\right)^{H_y} \qquad \qquad L_{NS} \approx 3 \times 10^6 m$$

(7759)

(7860)

(7961)

825 The fluctuations we used are Haar fluctuations, but because Hx ≈ Hy > 0, they are nearly equal to difference fluctuations [Lovejoy and Schertzer, 2012]. We see that the zonal and meridional lines are roughly parallel: with a "trivial" horizontal anisotropy factor ≈ 5 (typical north-south fluctuations are 5 times larger than typical east-west ones). Although, H = 1/2 is the value corresponding to Brownian motion, the actual variability is highly intermittent (spiky), so that unlike the temporal fluctuations, these spatial increments are far from Gaussian; it is *not* Brownian motion. Multifractal analysis indicates that the intermittency parameter (the codimension of the mean) C1 ≈ 0.16 which is very high, reflecting the strong spatial fluctuations as we move from one climate zone to another [Lovejoy and Schertzer, 2013], [Lovejoy, 2018], [Lovejoy, 2019b].

Since the north-south gradients are much stronger than the east-west ones, we can estimate the gradients and Laplacians\* by using the y direction fluctuations: at scale  $\Delta y$ :

$$r_{a,y} = \frac{V \alpha \Delta t}{\Delta y} \left(\frac{\Delta y}{L_{NS}}\right)^{H_{y}}$$
835
$$r_{d,y} = \frac{V \Delta t}{\Delta y} \left(\frac{I_{h}}{\Delta y}\right) \left[ \left(\frac{\Delta y}{L_{NS}}\right)^{2H_{y}} + \left(\frac{\Delta y}{L_{NS}}\right)^{H_{y}} \right]$$

| •( | Formatted: Font: 10 pt |
|----|------------------------|
| -( | Field Code Changed     |
| (  | Formatted: Font: 10 pt |
| ľ  | Formatted: Font: 10 pt |

Field Code Changed

Since  $L_{NS} \approx 3 \times 10^6 \text{m}$ , over most of the range of  $\Delta y$ ,  $r_{d,y} \approx \frac{V \Delta t}{\Delta y} \left(\frac{l_h}{\Delta y}\right) \left(\frac{\Delta y}{L_{NS}}\right)^{H_y}$  so that the ratio of advection to diffusion is  $\frac{r_c}{r_d} \approx \left(\frac{\alpha \Delta y}{l_h}\right)$  so that advection dominates diffusion for  $\Delta y > \frac{l_h}{\alpha}$ . Taking  $\alpha \approx 1$ , it is dominant for  $\Delta y \gg l_h$ .

Using  $l_h \approx 10^4$ m,  $L_{NS} \approx 3 \times 10^6$ m,  $H_y = 1/2$ ,  $V = 10^4$  m/s we find approximately critical length scales that yields unit ratios:

$$\Delta y_{c,a} = 10^{-14} \Delta t^2; \qquad r_a \left( \Delta y_{c,a} \right) = 1$$
$$\Delta y_{c,d} = 10^{-2} \Delta t^{2/3}; \qquad r_d \left( \Delta y_{c,d} \right) = 1$$

840

Where Δt is measured in seconds, Δy in meters. When the typical distances exceed these critical distances (i.e. when Δy>Δyc), we have t<1 so that the temporal derivative terms dominate over the horizontal transport. For Δt = 1 month, we have Δyc,a ≈ 0.1m, and Δyc,d ≈ 200m, so that unless the distances are very small, the temporal (storage) terms are indeed dominant. Even over much longer time scales - e.g. Δt ≈30 years (1010s109s), they dominate for distances greater than ≈Δyc,a ≈ Δyc,d ≈ 10 km.
845 Alternatively, we could estimate the time scales needed so that the critical transport scale is 1000km. From the same equations, we obtain estimates of 300 years (advection), 30,000 years (diffusion). Note however that in the anthropocene, for periods Δt ≈ 10 years, that the temporal fluctuations start to grow (i.e. the empirical relations eqs. 6078, 61–79 will break down); nevertheless, the above scaling relations for the internal variability may hold to much longer times [Lovejoy et al., 2013].

In summary, from eq. 6280, we conclude that for the larger scales >>==10 km, that  $r\ll 1$  and that the HEBE may apply 850 except for time scales  $\gg \tau$ : the only explicit role of  $\kappa_h$ ,  $\kappa_v$ ,  $\rho$ , c is to determine the limits of validity of the HEBE via  $l_h$ ,  $\alpha$ . When the HEBE is valid, only the relaxation time  $\tau$  and the climate sensitivity  $\rho_r$  are relevant.

**Appendix B: The HEBE cross-correlations**

The temperature anomaly cross-correlation function (a matrix when the temperature is discretized on a grid), is commonly used in climate science, notably to determine Empirical Orthogonal Functions (EOFs). These can be determined from the

855 HEBE (or GHEBE if needed) once a forcing model is given. Let us first consider that the climate sensitivities and relaxation times are deterministic characterizations of the local properties at points  $\underline{x}_1, \underline{x}_2$ . In this case, for the HEBE, any correlations between the temperature anomalies at those points will arise because of correlations in the forcing  $F(\underline{x},t)$ . We now consider simple deterministic and stochastic forcings.

| ( | Formatted: Font: 10 pt              |
|---|-------------------------------------|
| ( | Field Code Changed                  |
| ( | Formatted: Font: 10 pt              |
| ( | Formatted: Indent: First line: 1 cm |
| ( | Formatted: Font: Italic             |

(8062)

**a) Deterministic forcing, temporal averaging:**

860 The simplest model is to take complete-spatial correlation correlations obtained by temporally averaging, following with a step function ( $\Theta(t)$ ) forcing at t = 0, but different at each position x:

$$F(\underline{x},t) = F_0(\underline{x})\Theta(t)$$
(8163)

The temporally averaged cross-correlation can be determined by:

865
$$T(\underline{x}_{1},t)T(\underline{x}_{2},t) = \frac{s(\underline{x}_{1})F_{0}(\underline{x}_{1})s(\underline{x}_{2})F_{0}(\underline{x}_{2})}{\tau(\underline{x}_{1})\tau(\underline{x}_{2})} \int_{0}^{t} \int_{0}^{t} G_{\delta,1/2}\left(\frac{t-u_{1}}{\tau(\underline{x}_{1})}\right)G_{\delta,1/2}\left(\frac{t-u_{2}}{\tau(\underline{x}_{2})}\right)du_{1}du_{2}$$
(8264)

Recalling that  $G_{4,1/2}$  (=  $G_{\Theta}$ ) is the step response, is the integral of  $G_{\Theta,1/2}$  (=  $G_{\Theta}$ ) and since  $G_{\Theta,1/2}$  ( $\infty$ ) = 1 we have:

$$\lim_{t_{L}\to\infty} \left[ \frac{1}{t_{L}} \int_{0}^{t_{L}} G_{\Theta,1/2}\left(\frac{t}{\tau(\underline{x}_{1})}\right) G_{\Theta,1/2}\left(\frac{t}{\tau(\underline{x}_{2})}\right) dt \right] = 1$$
(8365)

Hence:

870
$$\overline{T(\underline{x}_{1},t)T(\underline{x}_{2},t)} = s(\underline{x}_{1})F_{0}(\underline{x}_{1})s(\underline{x}_{2})F_{0}(\underline{x}_{2})$$
(8466)

**b) Stochastic forcing:**

A convenient model of pure internal variability, is to assume that the forcing is statistically stationary in time with the following forcing cross-correlations:

$$R_{F}(\underline{x}_{1},\underline{x}_{2},\Delta t) = \langle F(\underline{x}_{1},t)F(\underline{x}_{2},t-\Delta t) \rangle$$
(8567)

875 (the "<>" symbol indicates ensemble, statistical averaging). This The corresponding implies a stationary temperature cross-correlation:

$$R_{T}(\underline{x}_{1},\underline{x}_{2},\Delta t) = \langle T(\underline{x}_{1},t)T(\underline{x}_{2},t-\Delta t) \rangle$$
(8668)

need to determine *R* for
$$\Delta r > 0$$
. For statistically stationary forcing,  $R_T(\underline{x}_1, \underline{x}_2, \Delta t)$  is the anomaly cross-correlation needed -
for example - for constructing Empirical Orthogonal Functions (EOFs).
The easiest way to relate  $R_F$  and  $R_T$  is via their spectra. Let us define the transform pairs:
 $\widehat{T(\omega)} = \int_{-\infty}^{\infty} e^{-i\omega t} T(t) dt;$   $T(t) = \frac{1}{2\pi} \int_{-\infty}^{\infty} e^{i\omega t} \widehat{T(\omega)} d\omega$  (8769)
similarly for the forcing *F* (the circonflex indicates Fourier Transform). Then:
 $\widehat{\left(\frac{d^H T}{dt^H}\right)} = (i\omega)^H \widehat{T}$  (8870)
(this is true for the Weyl fractional derivatives used here, [*Podlubny*, 1999]). So that the impulse response is:

Note the general symmetry property  $R(\underline{x}_1, \underline{x}_2, -\Delta t) = R(\underline{x}_2, \underline{x}_1, \Delta t); R(\underline{x}_1, \underline{x}_2, -\Delta t) = R(\underline{x}_2, \underline{x}_1, \Delta t)$  so that we only

885

$$G_{\delta,1/2}(t) = \frac{1}{2\pi} \int_{-\infty}^{\infty} \frac{e^{i\omega t}}{1 + (i\omega)^{1/2}} d\omega$$
(8971)

The solution to the HEBE at two different points  $\underline{x}_1, \underline{x}_2$  is:

$$\hat{T}(\underline{x}_{1}, \omega_{1}) = s(\underline{x}_{1}) \frac{\hat{F}(\underline{x}_{1}, \omega_{1})}{1 + (i\omega_{1}\tau(\underline{x}_{1}))^{1/2}}$$

$$\hat{T}^{*}(\underline{x}_{2}, \omega_{2}) = s(\underline{x}_{2}) \frac{\hat{F}^{*}(\underline{x}_{2}, \omega_{2})}{1 + (-i\omega_{2}\tau(\underline{x}_{2}))^{1/2}}$$
(9072)

890 Where the asterix indicates complex conjugate. Multiplying and taking ensemble averages and assuming that the forcing and hence responses - are statistical stationary, we obtain:

$$\widehat{\langle T^*(\underline{x}_1,\omega)\hat{T}(\underline{x}_2,\omega')\rangle} = \widehat{R}_T(\underline{x}_1,\underline{x}_2,\omega)\delta(\omega-\omega'); \qquad \widehat{R}_T(\underline{x}_1,\underline{x}_2,\omega) = \widehat{R}_T^*(\underline{x}_2,\underline{x}_1,\omega)$$
(9173)

Where:

880

895
$$R_{T}\left(\underline{x}_{1}, \underline{x}_{2}, \Delta t\right) = \frac{1}{2\pi} \int_{-\infty}^{\infty} e^{i\omega\Delta t} \widehat{R}_{T}\left(\underline{x}_{1}, \underline{x}_{2}, \omega\right) d\omega$$

Therefore:

$$R_{T}(\underline{x}_{1},\underline{x}_{2},\omega) = s(\underline{x}_{1})s(\underline{x}_{2})\hat{G}_{T}(\underline{x}_{1},\underline{x}_{2},\omega)\hat{R}_{F}(\underline{x}_{1},\underline{x}_{2},\omega);$$
$$\hat{G}_{T}(\underline{x}_{1},\underline{x}_{2},\omega) = \frac{1}{\left(1 + \left(-i\omega\tau(\underline{x}_{1})\right)^{1/2}\right)\left(1 + \left(i\omega\tau(\underline{x}_{2})\right)^{1/2}\right)}$$

A special case that is useful later, is when  $\underline{x}_1 = \underline{x}_2 = \underline{x}$ , which yields the spectrum  $E_T$  at the point  $\underline{x}$ :

$$E_{T}(\underline{x},\omega)\delta(\omega-\omega') = \left\langle \hat{T}(\underline{x},\omega)\hat{T}^{*}(\underline{x},\omega')\right\rangle; \qquad E_{T}(\underline{x},\omega) = \hat{R}_{T}(\underline{x},\underline{x},\omega)$$

Using a partial fraction expansion of eq. 7593, we obtain:

$$\widehat{G}_{T}(\underline{x}_{1},\underline{x}_{2},\omega) = \frac{1}{\tau_{1}+\tau_{2}} \left[ \frac{\tau_{1}+i\tau_{g}}{\left(1+\left(-i\omega\tau_{1}\right)^{1/2}\right)} + \frac{\tau_{2}-i\tau_{g}}{\left(1+\left(i\omega\tau_{2}\right)^{1/2}\right)} \right]; \qquad \qquad \tau_{g} = sign(\omega)(\tau_{1}\tau_{2})^{1/2}$$

905 By inverting the Fourier transform, this can be used to determine the real space transfer function  $G_T(\underline{x}_1, \underline{x}_2, \Delta t)$ . Using contour integration, it is convenient to convert the inverse Fourier transforms into Laplace transforms for  $\Delta t > 0$ :

$$G_{T}(\underline{x}_{1},\underline{x}_{2},\Delta t) = \frac{1}{\pi(\tau_{1}+\tau_{2})} \left[ \int_{0}^{\infty} e^{-x(\Delta t/\tau_{2})} \frac{x^{1/2}}{1+x} dx + \left(\frac{\tau_{2}}{\tau_{1}}\right)^{1/2} \int_{0}^{\infty} e^{-x(\Delta t/\tau_{2})} \frac{1}{1+x} dx - \left(\frac{\tau_{1}}{\tau_{2}}\right)^{1/2} \int_{0}^{\infty} e^{-x(\Delta t/\tau_{1})} \frac{1}{1+x^{1/2}} dx \right]$$
(9678)

For  $\Delta t < 0$ , use  $G_T(\underline{x}_1, \underline{x}_2, -\Delta t) = G_T(\underline{x}_2, \underline{x}_1, \Delta t)$ . The spatial cross-correlation, temporal autocorrelation function of the temperature is therefore:

43

Field Code Changed

(9274)

(9375)

(9476)

(9577)

$$R_{T}(\underline{x}_{1},\underline{x}_{2},\Delta t) = s(\underline{x}_{1})s(\underline{x}_{2})G_{T}(\underline{x}_{1},\underline{x}_{2},\Delta t) * R_{F}(\underline{x}_{1},\underline{x}_{2},\Delta t)$$

$$(9779)$$

Where the "\*" indicates convolution.

The basic Laplace transforms in eq. 78.96 can be expressed in terms of higher mathematical functions as follows (all for 100):

$$G_{\delta,1/2}(t) = \frac{1}{\pi} \int_{0}^{\infty} \frac{x^{1/2}}{1+x} e^{-xt} dx = \frac{1}{\sqrt{\pi t}} - e^{t} erfc(\sqrt{t})$$
(9880)

915
$$\frac{1}{\pi} \int_{0}^{\infty} \frac{e^{-xt}}{1+x} dx = \frac{1}{\pi} e^{t} \Gamma(0,t); \qquad \Gamma(0,t) = \int_{t}^{\infty} \frac{e^{-t}}{t} dt$$

$$\frac{1}{\pi}\int_{0}^{\infty}\frac{1}{1+x^{1/2}}e^{-xt}dx = \frac{1}{\sqrt{\pi t}} - e^{-t}erfi\left(\sqrt{t}\right) + \frac{e^{-t}}{\pi}E_{I}(t); \qquad erfi(z) = erf(iz)/i = \operatorname{Im}\left(ercf(-iz)\right);$$

$$E_{I}(t) = -\int_{-t}^{\infty} \frac{e^{-t}}{t} dt = -\Gamma(0, -t) + i\pi$$

The  $i\pi$  comes from integrating half way around the pole at the origin. Note that both the Exponential Integral ( $E_l$ ) and the incomplete Gamma functions have log divergences at the origin. If needed, these formulae can be combined to obtain a complete analytic expression for  $G_T(\underline{x}_1, \underline{x}_2, \Delta t)$ , which can then be used to determine the temperature correlations if the 920

forcing correlations are known:  $R_{T}(\underline{x}_{1}, \underline{x}_{2}, \Delta t) = s(\underline{x}_{1})s(\underline{x}_{2})G_{T}(\underline{x}_{1}, \underline{x}_{2}, \Delta t) * R_{F}(\underline{x}_{1}, \underline{x}_{2}, \Delta t)$  where the asterix is the temporal convolution.

The special case  $\underline{x}_1 = \underline{x}_2$  i.e. with  $\tau_1 = \tau_2 = \tau$ , is a little simpler:

$$G_{T}(\Delta t) = \frac{1}{\tau}g\left(\frac{|\Delta t|}{\tau}\right); \quad g(\Delta t) = \frac{1}{2\pi}\int_{0}^{\infty} e^{-x\Delta t}\left(\frac{x^{1/2}}{1+x} + \frac{1}{1+x} - \frac{1}{1+x^{1/2}}\right)dx; \quad \Delta t > 0$$
(9981)

925 Whose Fourier transform is:

$$\hat{G}_{T}(\underline{x},\underline{x},\omega) = \frac{1}{1+2\operatorname{Re}\left[\left(-i\omega\tau\right)^{1/2}\right] + \omega\tau}$$
(10082)

Evaluating the integral for  $g(\Delta t)$  using the Laplace transform formulae (eq.  $\frac{8098}{2}$ ):

$$g(\Delta t) = \frac{1}{\pi} \left( e^{\Delta t} \Gamma(0, \Delta t) + e^{-\Delta t} \operatorname{Re}\left(\Gamma(0, -\Delta t)\right) \right) - \left( e^{\Delta t} \operatorname{erfc}\sqrt{\Delta t} + e^{-\Delta t} \operatorname{Im}\left(\operatorname{erfc}\left(-i\sqrt{\Delta t}\right)\right) \right)$$
(10183)

( $\Delta t \ge 0$ ). The small scale and asymptotic limits are thus:

$$g\left(\Delta t\right) = -\frac{\log \Delta t}{\pi} - \frac{1}{2} - \frac{\gamma_E}{\pi} + 2\sqrt{\frac{\Delta t}{\pi}} - \frac{t}{2} - \left(\frac{t^2 \log \Delta t}{2\pi}\right) + \dots \qquad \Delta t \ll 1$$

930

$$g(\Delta t) \approx \frac{1}{\Delta t \sqrt{\pi \Delta t}} - \frac{2}{\pi \Delta t^2} + \frac{15}{8 \Delta t^3 \sqrt{\pi \Delta t}} - \dots \qquad \Delta t \gg 1$$
(10284)

Note the small scale log divergence, this is important when the forcing is a white noise, see [Lovejoy, 2019a]. The temporal autocorrelation at the point  $\underline{x}$  is thus:

$$R_{T}(\underline{x},\Delta t) = \frac{\lambda(\underline{x})^{2}}{\tau(\underline{x})}g(\Delta t / \tau(\underline{x})) * R_{F}(\underline{x},\Delta t); \qquad R(\underline{x},\Delta t) = R(\underline{x},\underline{x},\Delta t)$$

$$R_{T}(\underline{x},\Delta t) = \frac{s(\underline{x})^{2}}{\tau(\underline{x})}g(\Delta t / \tau(\underline{x})) * R_{F}(\underline{x},\Delta t); \qquad R(\underline{x},\Delta t) = R(\underline{x},\underline{x},\Delta t) \qquad (10385)$$

However, in general, the Fourier relations are easier to deal with.

**Appendix C: Statistical Space-Time Factorization**

.

At high frequencies (i.e.  $\Delta t < \tau$ ), and empirically over the macroweather regime up to a decade or more ([Lovejoy and de Lima, 2015]), both precipitation and temperature anomalies (at least approximately) respect a space-time symmetry called "spacetime statistical factorization" ("STSF"). For example, for the autocorrelation function *R*, this implies  $R_{gace-time}(\Delta \underline{x}, \Delta t) = R_{gace}(\underline{\Delta x})R_{time}(\Delta t)$ . If obeyed, this factorization implies important simplications in regional macroweather forecasting: it is therefore interesting to investigate the implications HEBE for the STSF hypothesis. The easiest way to approach the STSF is to consider that the forcing and relaxation times  $\tau(\underline{x})$  and sensitivities  $\lambda(\underline{x})$  are stochastic fields that are statistically homogeneous in space so that the correlation functions can be written: . If we assume

945 that the forcing is statistically independent of the temperature, then, taking the high frequency limit of  $\hat{G}_r$  in eq. 75:

$$\hat{G}_{T}(\underline{x},\underline{x}-\underline{\Delta x},\omega) \approx \frac{1}{\left(\tau(\underline{x})\tau(\underline{x}-\underline{\Delta x})\right)^{1/2}\omega}$$

we obtain:

advantages.

$$R_{r}(\underline{\Delta x},\omega) = < \left[\frac{\lambda(\underline{x})\lambda(\underline{x}-\underline{\Delta x})}{\left(\tau(\underline{x})\tau(\underline{x}-\underline{\Delta x})\right)^{1/2}}\right] > \frac{R_{r}(\underline{\Delta x},\omega)}{\omega}$$

(87)

(88)

(86)

950 From this, see that if the forcing factorizes then the temperature autocorrelation function also factorizes:

Where  $\underline{R}_{\lambda \tau^{-1/2}}(\underline{\Delta x})$  is the autocorrelation function of  $\lambda \tau^{-1/2}$  (the term in square brackets in eq. 87). From here, the inverse Fourier transform of and gives the real space version of the STSF symmetry. Notice that at the STSF hinges on the factorization approximation for  $\hat{G}_{\tau}(\underline{x}, \underline{x} - \underline{\Delta x}, \boldsymbol{\omega})$ -and at low  $\omega$ , it breaks down.

**955 Appendix **DC**: Fractional Integration on the sphere**

At long enough time scales, the spatial transport of heat is important and the spherical geometry of the Earth must be taken into account. The standard way (see section 2.3 and e.g. the reviews [North et al., 1981]. [North and Kim, 2017]) is to use spherical harmonics. In Appendix 5D of [Lovejoy and Schertzer, 2013] these were used to define fractional integrals on the sphere, necessary in order to produce the corresponding multifractal cloud and topography models (see also [Landais et 960 al., 2019]). Spherical harmonics are particularly convenient when the heat transport is diffusive, involving fractional Laplacians. In section 3.5.2, these were defined in real space by taking the domain of integration to be a sphere. In this appendix we discuss an alternative method of spherical fractional integration that may have theoretical and practical

The Laplacian on a sphere  $(\nabla_{\Omega}^2)$  is the angular part of the Laplacian in spherical coordinates, it is obtained by expressing the

965 Laplacian in spherical coordinates and setting the radial derivatives to zero:

$$\nabla_{\Omega}^{2} = \left[\frac{\partial}{\partial\mu}\left(1-\mu^{2}\right)\frac{\partial}{\partial\mu} + \frac{1}{\left(1-\mu^{2}\right)}\frac{\partial^{2}}{\partial\phi^{2}}\right]; \quad \mu = \cos\theta$$

(10489)

Field Code Changed

Field Code Changed

where
$$\theta$$
 is the colatitude and  $\phi$  is the longitude. The normalized eigenfunctions of  $\nabla_{\mu,\mu}^{2}$  are the spherical harmonics  $Y_{n,\nu}$ :

$$Y_{n,\mu}(\mu,\phi) = \left[\frac{2n+1}{4\pi} \frac{(n+|m|)}{(n+|m|)!}\right]^{3/2} P_{\mu,\mu}(\mu) e^{im\theta} \left( (-1)^{n}; m \ge 0 \\ 1; m < 0 \right); \mu = \cos\theta; -n \le m \le n$$
(10590)
With  $m, n$  integer,  $n \ge 0$  and  $P_{n,\nu}$  is the associated Legendre polynomial.  $Y_{n,\nu}$  satisfies:

$$-\nabla_{\mu}^{2} Y_{n,\mu}(\mu,\phi) = n(n+1) Y_{n,\mu}(\mu,\phi)$$
(10591)
So that  $n(n+1)$  are the eigenvalues. Since  $|m| \le n$  there are  $2n+1$  degenerate eigenvalues and functions for each  $n$ .
The spherical harmonics form a complete orthogonal basis, so that any function  $f(\mu,\phi)$  on the sphere can be uniquely
975 expressed in terms of a spherical harmonic expansion:

$$f(\mu,\phi) = \sum_{n=0}^{\infty} \sum_{m=0}^{n} \frac{f_{n,0}^{(0)}}{n_{m}^{2} r_{m}}(\mu,\phi); \quad f_{n,m}^{(0)} = \frac{2}{p_{n-1}^{2}} \int_{-1}^{1} Y_{n,m}(\mu,\phi) f(\mu,\phi) d\mu d\phi$$
(10792)
Where  $\lim_{n \to \infty} f_{n,m}^{(0)} x_{n}(\mu,\phi); \quad F_{n,m}^{(0)} = \frac{2}{p_{n-1}^{2}} \int_{-1}^{2} Y_{n,m}(\mu,\phi) f(\mu,\phi) d\mu d\phi$
(10792)
Where  $\lim_{n \to \infty} f_{n,m}^{(0)} x_{n}(\mu,\phi); \quad F_{n,m}^{(0)} = \frac{2}{p_{n-1}^{2}} \int_{-1}^{2} Y_{n,m}(\mu,\phi) f(\mu,\phi) d\mu d\phi$
(10792)
Field Code Changed
Field Code C

|     | $(-2)^{-H/2}$ (1) $\sum_{n=1}^{\infty} \sum_{n=1}^{m} (u)$ (1) (1) $(-1)^{-H/2}$ (0)                                                                                                         | Field Code Changed |
|-----|----------------------------------------------------------------------------------------------------------------------------------------------------------------------------------------------|--------------------|
|     | $\left(-\nabla_{\Omega}^{2}\right) \qquad f\left(\mu,\phi\right) = \sum_{n}\sum_{m} F_{n,m}^{(n)}Y_{n,m}\left(\mu,\phi\right);  F_{n,m}^{(n)} = \lfloor n(n+1) \rfloor \qquad F_{n,m}^{(0)}$ |                    |
| 985 | n=1 m=-n (10994)                                                                                                                                                                             |                    |
|     | i.e. a filter in spherical harmonic space, analogous to the Fourier filter $\underline{ \underline{k} }^{-H}$ for an isotropic fractional integration in Cartesian                           | Field Code Changed |
| 1   | coordinates.                                                                                                                                                                                 |                    |
| ĺ   | The definition of the fractional Laplacian (eq. $93111$ , $94112$ ) is adequate when the horizontal transport coefficients are                                                               |                    |
|     | constant, but in section 3.5, we saw that more generally, the half order divergence operator was written: $l(\mu,\phi)^{-1}(-\nabla_{\Omega}^2)^{-1/2}$                                      | Field Code Changed |
| 990 | i.e. there was an extra multiplication by the spatially varying diffusion length $l(\mu, \phi)$ . In flat (Cartesian) coordinates, such                                                      | Field Code Changed |
| I   | real space multiplications correspond to Fourier space convolutions so that this operator can also be conveniently expressed in                                                              |                    |
|     | Fourier space. However, with spherical harmonics, this simplicity is lost: although isotropic real space convolutions can still                                                              |                    |
|     | be performed by filtering the harmonics, real space multiplications no longer correspond to convolutions of harmonic                                                                         |                    |
|     | coefficients, the closest spherical harmonic equivalent is much more complicated, it involves Clebsch-Gordon coefficients.                                                                   |                    |
| 995 | A method of fractionally integrating the mean $(n = 0)$ component was developed for the purpose of multifractal                                                                              |                    |
|     | modeling in Appendix 5D of [Lovejoy and Schertzer, 2013]. There, a different definition of fractional integrals on the                                                                       |                    |
|     | sphere was proposed: a convolution with the function $\Theta^{-(2-H)}$ , where $\Theta$ is the angle between two points subtended at the center                                              |                    |
|     | of the sphere. The function $\Theta^{(2:H)}/\Gamma(H/2)$ was numerically expanded in spherical harmonics and the convolution was again                                                       |                    |
|     | performed by filtering the coefficients (the constant $\Gamma(H/2)$ is needed so that the normalization is the same as for the definition                                                    |                    |
| 000 | eq. 92107). The main difference between the two definitions is that the latter can be directly applied to fields with nonzero                                                                |                    |
|     | means. With this definition, the $H$ order fractional integral of a constant function on the sphere (representing the nonzero                                                         |                    |
|     | mean), is simply the value multiplied by $2^{-H/2}\sqrt{\pi}/\Gamma(H/2)\int_{0}^{2\pi}s^{-(2-H)}\sin sds$ which for the HEBE $H = 1$ case, reduces to                                       | Field Code Changed |
| 1   | $(1/2)^{1/2}Si(2\pi)$ where Si is the standard sine integral function. However for the coefficients $n \ge 1$ , numerical tests show that the                                                |                    |
|     | two definitions are almost exactly the same; for example with $H = 1$ , the spherical harmonic coefficients of $\Theta^{-(2-H)}$ are within                                                  |                    |
| 005 | 3% for all $n \ge 1$ and the ratio converges rapidly to 1 for large n . The conclusion is that filtering the anomaly by $\left[n(n+1)\right]^{-H/2}$                                  | Field Code Changed |
|     | and then multiplying the mean by the above factor is a practical method of fractionally integrating a function on the sphere.                                                                |                    |

**6. References**

Babenko, Y. I., *Heat and Mass Transfer*, Khimiya: Leningrad (in Russian), 1986. Brunt, D., Notes on radiation in the atmosphere, *Quart. J. Roy. Meterol. Soc.*, 58, 389-420, 1932.

1010 Chenkuan, L., and Clarkson, K., Babenko's Approach to Abel's Integral Equations, *Mathematics 6*, 32 doi: doi:10.3390/math6030032, 2018.

Coffey, W. T., Kalmykov, Y. P., and Titov, S. V., Characteristic times of anomalous diffusion in a potential, in *Fractional Dynamics: Recent Advances*, edited by J. Klafter, S. Lim and R. Metzler, pp. 51-76, World Scientific, 2012.

Del Rio Amador, L., and Lovejoy, S., Predicting the global temperature with the Stochastic Seasonal to Interannual Prediction System (StocSIPS) *Clim. Dyn.* doi: org/10.1007/s00382-019-04791-4, 2019.

Del Rio Amador, L., and Lovejoy, S., Using scaling for seasonal global surface temperature forecasts: StocSIPS Clim. Dyn., under review, 2020a.

Del Rio Amador, L., and Lovejoy, S., Long-range Forecasting as a Past Value Problem: Using Scaling to Untangle Correlations and Causality *Geophys. Res. Lett.*, (submitted, Nov. 2020), 2020b.

Havlin, S., and Ben-Avraham, D., Diffusion in disordered media, Adv. Phys., 36, 695-798, 1987.
 Hebert, R. (2017), A Scaling Model for the Forced Climate Variability in the Anthropocene, MSc thesis, McGill University, Montreal.
 Hébert, R., Lovejoy, S., and Tremblay, B., An Observation-based Scaling Model for Climate Sensitivity Estimates and Global

Projections to 2100, *Climate Dynamics, (in press)*, 2020.

Hilfer, R. (Ed.), Applications of Fractional Calculus in Physics World Scientific, 2000.
Houghton, J. T., Ding, Y., Griggs, D. J., Noguer, M., van der Linden, P. J., Dai, X., Maskell, K., and Johnson, C. A. (Eds.), Climate Change 2001: The Scientific Basis, Contribution of Working Group I to the Third Assessment Report of the Intergovernmental Panel on Climate Change, Cambridge University Press, 2001.
Kobelev, V., and Romanov, E., Fractional Langevin Equation to Describe Anomalous Diffusion Prog. of Theor. Physics Supp., 1030 139, 470-476, 2000.

Kulish, V. V., and Lage, J. L., Fractional-diffusion solutions for transport, local temperature and heat flux. , ASME Journal of Heat Transfer, 122 372-376, 2000.

Landais, F., Schmidt, F., and Lovejoy, S., Topography of (exo)planets, MNRAS 484, 787-793 doi: 10.1093/mnras/sty3253, 2019.

1035 Lionel, R., D., C. M., M., C., S., S., and M., G., Parameter estimation for energy balance models with memory Proc. R. Soc. A, 470 doi: doi.org/10.1098/rspa.2014.0349, 2014.

Lovejoy, S., What is climate?, EOS, 94, (1), 1 January, p1-2, 2013.

Lovejoy, S., How accurately do we know the temperature of the surface of the earth? , *Clim. Dyn.* doi: doi:10.1007/s00382-017-3561-9, 2017.

1040 Lovejoy, S., The spectra, intermittency and extremes of weather, macroweather and climate, *Nature Scientific Reports*, 8, 1-13 doi: 10.1038/s41598-018-30829-4, 2018.

Lovejoy, S., Fractional Relaxation noises, motions and the stochastic fractional relxation equation *Nonlinear Proc. in Geophys. Disc.*, https://doi.org/10.5194/npg-2019-39, 2019a.

Lovejoy, S., Weather, Macroweather and Climate: our random yet predictable atmosphere, 334 pp., Oxford U. Press, 2019b.
 Lovejoy, S., and Schertzer, D., Haar wavelets, fluctuations and structure functions: convenient choices for geophysics, Nonlinear Proc. Geophys., 19, 1-14 doi: 10.5194/npg-19-1-2012, 2012.

Lovejoy, S., and Schertzer, D., *The Weather and Climate: Emergent Laws and Multifractal Cascades*, 496 pp., Cambridge University Press, 2013.

Lovejoy, S., and de Lima, M. I. P., The joint space-time statistics of macroweather precipitation, space-time statistical factorization and macroweather models, *Chaos 25*, 075410 doi: 10.1063/1.4927223., 2015.

Lovejoy, S., Schertzer, D., and Silas, P., Diffusion on one dimensional multifractals, *Water Res. Res.*, 34, 3283-3291, 1998. Lovejoy, S., Schertzer, D., and Varon, D., Do GCM's predict the climate... or macroweather?, *Earth Syst. Dynam.*, 4, 1–16 doi: 10.5194/esd-4-1-2013, 2013.

Lovejoy, S., Del Rio Amador, L., and Hébert, R., Harnessing butterflies: theory and practice of the Stochastic Seasonal to Interannual Prediction System (StocSIPS), , in *Nonlinear Advances in Geosciences*, , edited by A. A. Tsonis, pp. 305-355, Springer Nature, 2017.

Lovejoy, S., del Rio Amador, L., and Hébert, R., The ScaLIng Macroweather Model (SLIMM): using scaling to forecast global-scale macroweather from months to Decades, *Earth Syst. Dynam.*, *6*, 1–22 doi: www.earth-syst-dynam.net/6/1/2015/, doi:10.5194/esd-6-1-2015, 2015.

1060 Lovejoy, S., Procyk, R., Hébert, R., and del Rio Amador, L., The Fractional Energy Balance Equation, *Quart. J. Roy. Met. Soc.*, (under revision), 2020.

Magin, R., Sagher, Y., and Boregowda, S., Application of fractional calculus in modeling and solving the bioheat equation, in *Design and Nature II*, edited by M. W. C. C. A. Brebbia, pp. 207-216, WIT Press, 2004.

Márquez, J. M. A., Bohórquez, M. A. M., and Melgar, S. G., Ground Thermal Diffusivity Calculation by Direct Soil
 Temperature Measurement. Application to very Low Enthalpy Geothermal Energy Systems, *Sensors (Basel)*, 16, 306 doi: 10.3390/s16030306, 2016.

Meerschaert, M. M., and Sikorskii, A., Stochastic Models for Fractional Calculus, 2012.

Miller, K. S., and Ross, B., An introduction to the fractional calculus and fractional differential equations, 366 pp., John Wiley and Sons, 1993.

1070 North, G. R., Cahalan, R. F., and Coakley, J., J. A., Energy balance climate models, *Rev. Geophysics Space Phy.*, 19, 91-121, 1981.

North, R. G., and Kim, K. Y., *Energy Balance Climate Models*, 369 pp., Wiley-VCH, 2017.

Podlubny, I., Fractional Differential Equations, 340 pp., Academic Press, 1999.

Procyk, R., Lovejoy, S., and Hébert, R., The Fractional Energy Balance Equation for Climate projections through 2100, *Earth* 5/95. *Dyn. Disc., under review* doi: org/10.5194/esd-2020-48 2020.

Schertzer, D., and Lovejoy, S., The dimension and intermittency of atmospheric dynamics, in *Turbulent Shear Flow*, edited by L. J. S. Bradbury, F. Durst, B.E. Launder, F.W. Schmidt, J.H. Whitelaw, pp. 7-33, Springer-Verlag, 1985.

Schertzer, D., and Lovejoy, S., Physical modeling and Analysis of Rain and Clouds by Anisotropic Scaling of Multiplicative Processes, *Journal of Geophysical Research*, *92*, 9693-9714, 1987.

080 Sellers, W. D., A global climate model based on the energy balance of the earth-atmosphere system, J. Appl. Meteorol., 8, 392-400, 1969.

Trenberth, K. E., Fasullo, J. T., and Kiehl, J., Earth's global energy budget, *Bull. Amer. Met.Soc.*,, 311-323 doi: DOI:10.1175/2008BAMS2634.1, 2009.

Warren, S. G., and Schneider, S. H., Seasonal simu-lation as a test for uncertainties in the parameterizations of a Budyko-085 Sellers zonal climate model, *J. Atmos. Sci.*, *36*, 1377-1391, 1979.

Weissman, H., S. Havlin, Dynamics in multiplicative processes, *Phys. Rev. B*, *37*, 5994-5996, 1988.
West, B. J., Bologna, M., and Grigolini, P., *Physics of Fractal Operators*, 354 pp., Springer, 2003.
Zhuang, K., North, G. R., and Stevens, M. J., A NetCDF version of the two-dimensional energy balance model based on the full multigrid algorithm, *SoftwareX*, *6*, 198–202 doi: <a href="https://doi.org/doi.org/10.1016/j.softx.2017.07.003">https://doi.org/doi.org/10.1016/j.softx.2017.07.003</a>, 2017.

1090 Ziegler, E., and Rehfeld, K., TransEBM v. 1.0: Description, tuning, and validation of a transient model of the Earth's energy balance in two dimensions, *Geosci. Model Devel. Disc.* doi: https://doi.org/10.5194/gmd-2020-237, 2020.

Fig. 1: The surface impulse response function  $(G_{\delta}(t,r;0))$ , eq. 12, i.e. Dirac in time and Dirac in space) as a function of nondimensional time (*t*) for nondimensional distance from the source increasing from r = 0 (top) to r = 1 in steps of 0.2 (top to bottom).